# MuSR: Testing the Limits of Chain-of-thought with Multistep Soft Reasoning

**Zayne Sprague, Xi Ye, Kaj Bostrom, Swarat Chaudhuri, Greg Durrett**
Department of Computer Science
The University of Texas in Austin
`zayne@utexas.edu`

## Abstract

While large language models (LLMs) equipped with techniques like chain-of-thought prompting have demonstrated impressive capabilities, they still fall short in their ability to reason robustly in complex settings. However, evaluating LLM reasoning is challenging because system capabilities continue to grow while benchmark datasets for tasks like logical deduction have remained static. We introduce MuSR, a dataset for evaluating language models on multistep soft reasoning tasks specified in a natural language narrative. This dataset has two crucial features. First, it is created through a novel neurosymbolic synthetic-to-natural generation algorithm, enabling the construction of complex reasoning instances that challenge GPT-4 (e.g., murder mysteries roughly 1000 words in length) and which can be scaled further as more capable LLMs are released. Second, our dataset instances are free text narratives corresponding to real-world domains of reasoning; this makes it simultaneously much more challenging than other synthetically-crafted benchmarks while remaining realistic and tractable for human annotators to solve with high accuracy. We evaluate a range of LLMs and prompting techniques on this dataset and characterize the gaps that remain for techniques like chain-of-thought to perform robust reasoning.[1]

## 1 Introduction

A great remaining challenge for large language models (LLMs) is the ability to do reasoning and planning (Valmeekam et al., 2023; Tang et al., 2023; Dziri et al., 2023) Numerous methods have been proposed to augment models' capabilities on this front, including prompting strategies like chain-of-thought (Wei et al., 2022), integration with tools (Schick et al., 2023; Lyu et al., 2023; Ye et al., 2023), and embedding models in search loops (Bostrom et al., 2022; Creswell et al., 2023).

Do these approaches suitably address the shortcomings of LLMs? This is difficult to measure. Math reasoning tasks can be approached in a two-stage fashion (Gao et al., 2022; Ye et al., 2023): the LLM translates the problem into a formal specification which is then solved with conventional tools. Other datasets like RuleTakers (Clark et al., 2020) and CLUTRR (Sinha et al., 2019) are solvable with rule-based systems (Kazemi et al., 2023a; Ye et al., 2023; Poesia et al., 2023). Finally, datasets like SocialIQA (Sap et al., 2019) or StrategyQA (Geva et al., 2021) that involve more nuanced commonsense are often structurally simple (i.e., only involve 1-2 steps of reasoning). What is lacking is a benchmark involving **both** sophisticated natural language and sophisticated reasoning.

In this work, we present MuSR: Multistep Soft Reasoning, a dataset focused on tasks involving reasoning based on text narratives. The narratives in our dataset are hundreds of words long and present evidence in ways that require commonsense knowledge to unpack. Then, when all of the evidence is assessed, coming to a final answer requires "System 2"-style deliberation, which takes a different form for each domain of interest. The domains we address here, murder mysteries, object placement, and team assignment, involve commonsense reasoning about physical (Bisk et al., 2020) and social situations (Sap et al., 2019), theory-of-mind, and more. Crucially, these types of reasoning *arise naturally* from text descriptions of each problem.

---

[1] Project website can be found at `https://github.com/Zayne-Sprague/MuSR`

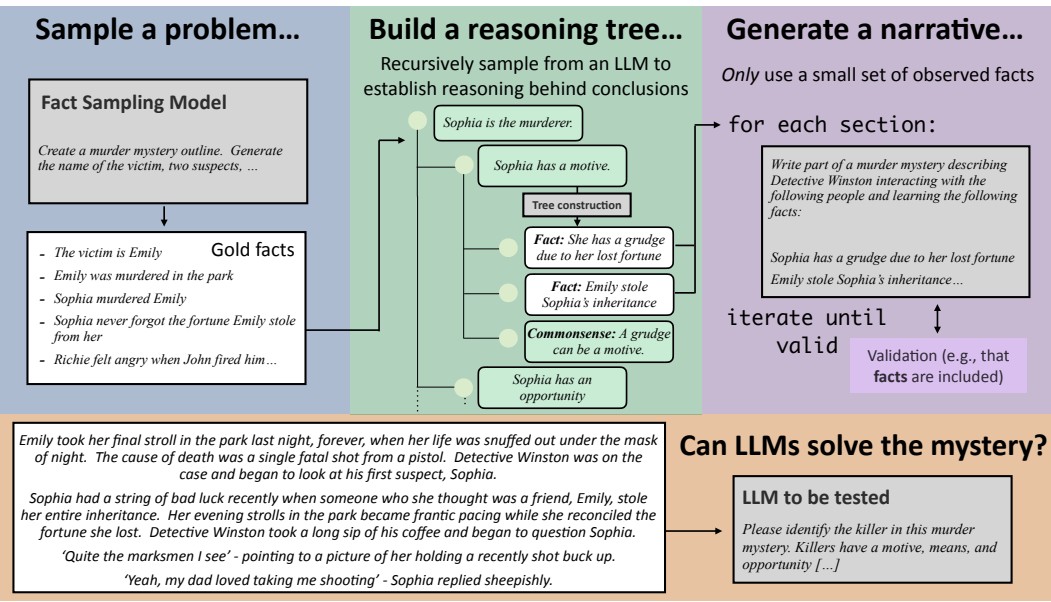

Figure 1: Dataset construction process for MuSR. First, we generate gold facts that are used to deduce the correct answer (the murderer in this case). Then, using an LLM, we create a reasoning tree leading to those deductions from facts in a story combined with commonsense. Finally, we iteratively generate a narrative one chunk at a time using the facts generated in step 2, validating the generations for fact consistency and recall.

The congruence between the reasoning and the text itself allows us to generate these datasets automatically with the aid of LLMs, using supporting logic to elicit examples that *the LLMs themselves* cannot reliably solve. Our novel neurosymbolic dataset generation procedure is shown in Figure 1. Recovering the reasoning from the final narrative itself is a hard problem, solvable by humans but not by GPT-4 using any of a number of prompting strategies and neurosymbolic approaches we tried. Notably, these properties do not hold when creating narratives with more basic prompting strategies: asking GPT-4 to define and write a murder mystery in a single shot leads to unnatural, homogeneous stories that may include inconsistencies, as we show later.

Our contributions are as follows: (1) We introduce a new reasoning benchmark, MuSR, consisting of 756 total examples across three domains that challenge state-of-the-art models such as GPT-4, Llama 2, and Vicuna. (2) We propose an algorithm for generating natural language narratives grounded in reasoning trees. (3) We analyze the performance of LLMs on our dataset, focusing on variants of chain-of-thought prompting and existing neurosymbolic approaches for these problems.

## 2 BACKGROUND AND MOTIVATION

We survey past dataset efforts in Table 1, using our analysis to establish the need for a new textual reasoning benchmark. First, a number of prior benchmarks do not have **natural text**. Others do not blend **commonsense** and **multistep** reasoning. Finally, we want a dataset that contains ground-truth **intermediate structure** and which is not **solvable with rules**.

Many past datasets are simply too artificial, including bAbI (Weston et al., 2016), BigTOM (Gandhi et al., 2023), ToMi (Le et al., 2019), RuleTakers (Clark et al., 2020), ProntoQA (Saparov & He, 2023; Saparov et al., 2023), and CLUTRR (Sinha et al., 2019). These datasets are generally designed to test some aspect of language model reasoning, but they are only challenging for "pure" LLM approaches; many are solvable with rule-based methods. Furthermore, many of these do not involve any commonsense reasoning, a key feature of reasoning from text.

EntailmentBank (Dalvi et al., 2021), Everyday Norms: Why Not (Sprague et al., 2022), and BoardgameQA (Kazemi et al., 2023b) present somewhat more challenging multistep settings, but consist of isolated collections of facts, not grounded in complex narratives. LLMs can solve the former two datasets quite easily even without consulting ground truth facts. As these datasets are

Table 1: Recent reasoning datasets used for benchmarking LLMs and neurosymbolic systems compared across various dataset qualities. To the best of our knowledge, no previous dataset encompasses all of these qualities. The $\sim$ symbol denotes datasets that partially qualify for the property. More details about how we define and classify these features can be found in Appendix B.

| Dataset | Properties | | | | |
| --- | --- | --- | --- | --- | --- |
| | Natural Text | Commonsense | Multistep | Intermediate structure | Not solvable w/rules |
| bAbI | ✗ | ✗ | ✓ | ✓ | ✗ |
| BigTOM | ~ | ~ | ~ | ✗ | ✓ |
| ToMi | ✗ | ✗ | ~ | ✓ | ~ |
| RuleTakers | ✗ | ✗ | ✓ | ✓ | ✗ |
| ProntoQA | ✗ | ✗ | ✓ | ✓ | ~ |
| CLUTRR | ✗ | ~ | ~ | ✓ | ~ |
| BoardgameQA | ~ | ✓ | ✓ | ✓ | ✓ |
| EntailmentBank | ~ | ✗ | ✓ | ✓ | ✓ |
| ENWN | ~ | ✗ | ✓ | ✓ | ✓ |
| SocialIQA | ~ | ✓ | ✗ | ✗ | ✓ |
| True Detective | ✓ | ✓ | ✓ | ✗ | ✓ |
| MuSR | ✓ | ✓ | ✓ | ✓ | ✓ |

designed to be solved using explicit step-by-step deduction, they tend to avoid softer kinds of inferences prevalent in commonsense reasoning. Past commonsense datasets (Sap et al., 2019; Talmor et al., 2019; 2021), conversely, often do not involve multistep reasoning.

**Techniques** Many reasoning systems have been built to handle specific axes of reasoning that we list in Table 1, but cannot handle a dataset which exhibits them all. Several past systems employ an LLM in a search loop that enumerates a list of facts, generating conclusions deductively or abductively until a goal is reached (Bostrom et al., 2022; Creswell et al., 2023; Kazemi et al., 2023a; Sprague et al., 2022; Hong et al., 2022). However, these systems do not handle natural contexts where facts can be distributed among sentences across a long narrative. Other systems involve tools or neurosymbolic algorithms to help solve reasoning problems (Sclar et al., 2023; Gao et al., 2022; Ye et al., 2023); however, these are often run on artificial datasets that can be easily translated into formal specifications, and have limited ability to handle soft reasoning types like commonsense.

One versatile technique is prompting, including various chain-of-thought strategies (Wei et al., 2022; Yao et al., 2023) and techniques to measure consistency (Wang et al., 2023; Jung et al., 2022). Using these approaches to solve reasoning problems end-to-end has shown to be challenging (Ye & Durrett, 2022; Zhang et al., 2023; Xue et al., 2023; Valmeekam et al., 2023). Our dataset is ideally suited to test the limits of these approaches: a system must extract facts from our stories, apply appropriate commonsense to interpret those facts, and finally use multistep reasoning to arrive at an answer.

**Why a synthetic benchmark** One alternative to the approach we describe could be to use human authoring. Our murder mystery domain is represented in the recent True Detective (Del & Fishel, 2022) dataset, which collects human-authored murder mysteries from `5minutemystery.com`. We argue that a synthetic benchmark is preferable for two reasons. First, it is scalable and can be renewed as more capable LLMs are produced. For example, if the mysteries on the aforementioned website are solved by future LLMs, it will be costly and challenging to collect a new dataset, whereas a synthetic benchmark can be refreshed with more complex reasoning and longer narratives. The disentangling of logical reasoning and text generation gives us a reusable lever for producing instances more complex than what systems themselves can solve. Second, because our dataset can be regenerated, issues with dataset leakage and exposure to test data become less of a concern. Finally, note that while our benchmark involves GPT-4 generated narratives, the scaffolding of the construction process and the hidden facts involved mean that the final generated outputs are not trivially solvable with GPT-4. As long as the underlying information is faithfully preserved in the narrative, we believe our data instances are valid test cases for any well-behaved reasoning system, which we verify by measuring human performance.

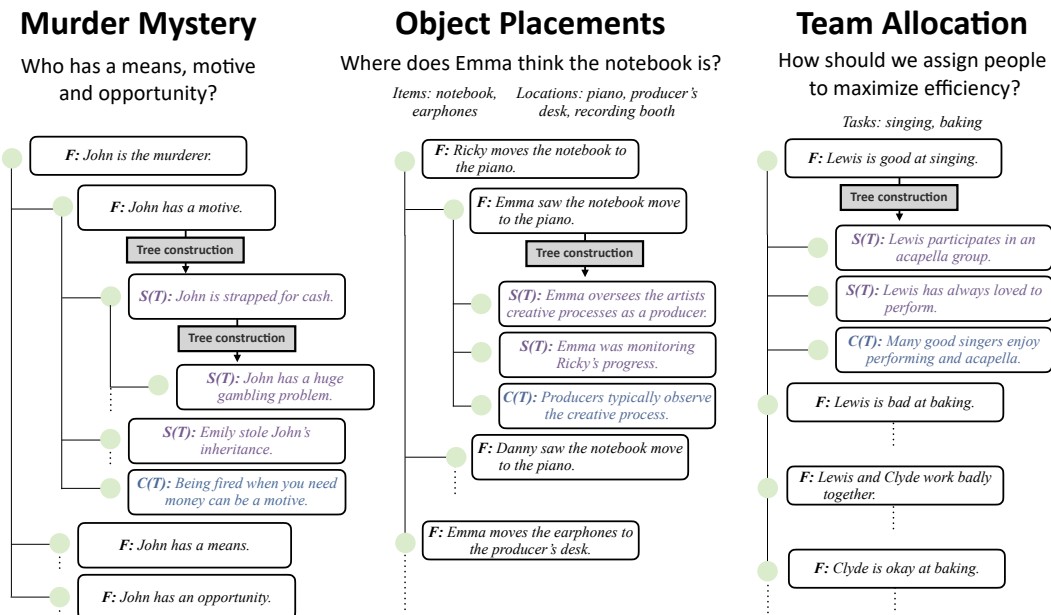

Figure 2: Partial reasoning trees showing gold facts $F$, story facts $S(T)$, and commonsense facts $C(T)$ for each of our three domains. Dotted lines indicate incomplete trees. Each deduction sampled from an LLM will yield two scenario facts and one commonsense fact in our setup.

## 3 CREATING MuSR

MuSR is composed of multi-step reasoning problems, each rooted in a specific domain with a unique logical structure to its solutions. To generate these problems, we have a construction method with three stages: **Tree Template Construction**, responsible for the reasoning strategy and initial gold fact set $F$; **Reasoning Tree Completion**, which generates a tree $T$ of intermediate reasoning steps expanding upon $F$; and **Story Generation**, which embeds the facts generated from the trees into a natural narrative $\mathbf{x}$. This process is described here and is represented in Figure 1.

The construction algorithm finally yields tuples $(F, T, \mathbf{x}, \{q_1, \ldots, q_n\}, \{a_1, \ldots, a_n\})$. Formally, the reasoning task is to predict answers $a_i$ given the narrative $\mathbf{x}$ and the question to answer $q_i$. The gold fact set $F$ and reasoning tree $T$ are used as part of the generation process but are generally *not* provided to an LLM at test time. Throughout the process, we use Prompt to denote using a prompted LLM to sample an output conditioned on another variable.

### 3.1 TREE TEMPLATE CONSTRUCTION

Each of our domains starts with a high-level fact set $F$, and a set of question-answer pairs $((q_1, a_1), \ldots, (q_n, a_n))$. For example, in our murder mysteries, the only question is "who is the murderer?" and $F$ contains ground-truth information about each suspect (*John is the murderer*, *John has an opportunity*). Information for each domain is in Section 4, with facts shown in Figure 2.

More formally, $F$ is a structured object with the requirement that there exists some program $\Phi$ such that $\Phi(F, q_i) = a_i$ for all $i$. $F$ can also be represented in natural language through templated expansion. Our questions $q$ and answers $a$ are templated. At this stage, we also generate additional facts $G$ to increase diversity and help expand the story later. This is done by sampling from curated lists or by sampling from LLMs (e.g., when a coherent set of objects needs to be generated for object placement). These facts differ from those in $F$ in that they are not templated but instead actual facts that must be included in the narrative. The output of this stage is a tuple $(F, G)$ which contains the core facts used to answer the question and a set of diversity facts used to give each question unique storylines.

## 3.2 REASONING TREE COMPLETION

Once the collection of facts $F$ has been constructed, we produce reasoning trees for each individual fact, $f_i$, in the set $F$. A reasoning tree $T = (\mathbf{s}, T_1, \ldots, T_m)$ is a recursive data structure representing a statement $\mathbf{s}$ supported by a collection of other statements: it must be the case that $\mathbf{s}$ is logically entailed by $\mathbf{s}_{T_1}, \ldots, \mathbf{s}_{T_m}$. The root of each tree is a fact $s_{T_1} = f_i$ where $f_i \in F$. We include the facts from $G$ while prompting the language model so that the generated facts include diverse information and ultimately help create interesting stories in the later stages.

These trees are automatically produced from root facts $f_i$ via recursively sampling from an LLM, in our case GPT-4. This process is shown in Algorithm 1 in Appendix G.1. We repeat this process to a specified depth, ultimately producing a set of leaf facts that deductively yield the original fact $f_i$ but require multi-step reasoning to do so. These facts are divided into two types: scenario-specific facts, which must be included in the ultimate narrative, and commonsense facts, which will not be stated but should be facts that most people would agree are true. We denote these sets of scenario facts and commonsense facts by $S(T)$ and $C(T)$, respectively, as shown in Figure 2.

Our generation process involves controlling the depth and shape of trees generated. We also want to ensure that there are no vacuous transitions in our trees (e.g., the fact $f_i$ being explicitly stated in a leaf node) or reasoning "shortcuts." To ensure this, we use a collection of **tree validators**, $V = (v_1, \ldots, v_k)$, per domain. These are often simple keyword lookups that prevent the keywords from appearing in the deduction, for example, preventing "motive" from appearing in a lower-level deduction in the murder mystery domain so that the reader is forced to deduce a motive. For more details on validators for each domain, see Appendix G.

At each step in the tree, for a node with text $\mathbf{s}$ we sample $T_1, \ldots, T_m \sim \mathrm{PromptLM}(T_1, \ldots, T_m \mid \mathbf{s})$. We then filter this output against the validators $V$. We retry this prompt up to three times, and if we are not able to draw a valid sample, prune the branch of the reasoning tree, making the current deduction a leaf node. We repeat this process until the tree is at the target depth. Figure 2 shows an example of the resulting trees.

## 3.3 STORY GENERATION

In the last stage, we use the scenario-specific leaf facts $S(T)$ from the reasoning tree. Our goal is to generate a narrative by sampling $\hat{\mathbf{x}} \sim \mathrm{Prompt}(\mathbf{x} \mid S(T))$ from an LLM with an appropriate prompt. However, for a long and complex narrative, $S(T)$ is not always reflected accurately in $\mathbf{x}$; some facts may be dropped as the model produces more of a summary of the situation, for example.

To address this, we instead divide $S(T)$ into chunks relating to a specific answer choice (e.g., the information related to a specific possible murderer). We can then use this subset to prompt GPT4 for a "chapter" with a smaller list of facts. Once every chapter has been created, we concatenate all the chapters together into one narrative. Some domains use additional prompts to "smooth" the final narrative when necessary. Because our narratives do not need to be produced by one LLM call, they can scale to be over 1000 words (in this dataset) and theoretically to be even longer. We refer this process as **chaptering**; the overall process is broadly inspired by Yang et al. (2022).

## 4 MuSR DOMAINS

### 4.1 MURDER MYSTERIES: SOCIAL AND PHYSICAL DEDUCTIVE REASONING

Murder mysteries are a classic domain requiring a variety of reasoning types. They have been explored in the context of LLM reasoning before (Frermann et al., 2018; Del & Fishel, 2022); however, ours is the first work to approach the scale of human-written murder mysteries with synthetic challenge data. Murder mysteries elicit physical and social reasoning in the fact sets $S(T)$ and $C(T)$. Specifically, unique and complex social scenarios arise naturally in murder mysteries that lead to motives for murder and can require understanding social norms. Solving a murder mystery also requires temporal reasoning about a person having an opportunity to commit the crime.

In this domain, $\Phi(F, q_i)$ is defined as an algorithm that can find the suspect with three facts in $F$. Specifically, the murderer and answer $a_i$ is the suspect with the facts "*x has a means*", "*x has a motive*", and "*x has an opportunity*". To construct $F$ such that $\Phi(F, q_i)$ will produce the correct

answer $a_i$, we create two suspects and randomly assign one as the murderer. We then populate the set $F$ with the three facts proving a means, motive, and opportunity. For the innocent suspect, we randomly chose two of the facts used to prove guilt, then add these and one additional "suspicious fact" to the set $F$, creating a set that does not establish guilt. A suspicious fact has no impact on $\Phi(F, q_i)$ and should not add any additional information relevant to the murder; for example, "*x is affiliated with a gang, and this is suspicious*".

In total, $F$ is composed of three facts per suspect that are passed to the reasoning tree creation stage. The reasoning tree will expand upon the descriptions for these facts, $G$, such as the fact that someone could have had an opportunity to murder a victim in their study by having a key to the study, which can be recursively expanded to describe *why* they had a key to the study. More details about the construction can be found in Appendix G.2.

## 4.2 Object Placements: Observational and Theory Of Mind Reasoning

Inspired by theory-of-mind datasets (Le et al., 2019; Gandhi et al., 2023) we chose a domain that focuses on a group of people moving items around to different locations. Other people in the story either see each item move or not for various reasons. The reader is then asked where a person would look for an item if asked to search for it, where the last move they saw is the most likely place they'd begin to search. Because of this, Object Placements requires spatial and physical reasoning as well as reasoning about people's awareness in $S(T)$ and $C(T)$. The reader is tested further by having to determine the observations of a specific person, modeling their belief state of where an item is. Notably, our dataset features longer narratives and more sophisticated scenarios than past theory-of-mind datasets.

In this domain, $q_i$ asks where a person believes an item to be in the story. The answer, $a_i$, is then the last location the person saw the item move in the story, or where the item was originally if they never saw the item move. $\Phi(F, q_i)$ is backed by a set of sequential moves $F$, where each move is a collection of observations denoting whether each person in the story saw the move or not. A move is denoted as a fact "$P$ moves $I$ to $L$" where $P$ is a person, $I$ is an item, and $L$ is a location, respectively. For every move, each person other than the one moving the item is given a chance $c$ (set to 0.33 for our experiments) to see the move, which will add either "$P'$ saw $I$ move to $L$." or "$P'$ did not see $I$ move to $L$." to $F$.

The reasoning trees then focus on explaining why someone may or may not have observed a move. This integrates commonsense reasoning: for example, a barista was busy doing latte art for a customer and didn't observe the manager moving an item from the fridge to the storage room. More details can be found in Appendix G.3.

## 4.3 Team Allocation: Social and Constraint Reasoning

Team Allocation takes inspiration from assignment and MAX-SAT problems (Pan et al., 2023). In this domain, the reader must determine the most optimal assignment of people to tasks where each person can be assigned to one task. Because there are three people and two tasks, two people must work together, adding a social dynamic to the assignment. $S(T)$ and $C(T)$ often involve inferring about past experiences and personal preferences of an individual as to why they do or do not perform a skill well. They also include reasoning over the strength of a relationship between two people in a workplace setting, which requires social reasoning.

$F$ represents these relations through numeric scores corresponding to each person's skill for a task and numerical teamwork score. Specifically, three people are each assigned score values for task capabilities (0, 1, or 2) and for their pairwise relationships. To solve a Team Allocation question, $\Phi(F, q_i)$ can enumerate the assignments adding the skill level and teamwork scores as a score for the overall assignment and then take the assignment that maximizes this score, $a_i = \Phi(F, q_i) = \max_{a_i \in A} \text{skill}(a_i) + \text{teamwork}(a_i)$. We found that using a small number of values for skills translated well into soft natural language statements where the decision of human annotators respects the hard underlying reasoning process. We further enforce that the optimal assignment outperforms all other assignments by a score of at least 2.

The reasoning tree then describes factors that contribute to these skills and relationships. More details can be found in Appendix G.4.

# 5 EXPERIMENTS

## 5.1 DATASET VALIDATION

We generate our three datasets comprising MuSR using GPT-4 following the procedure outlined in the previous sections. See Appendix C for a discussion of using models other than GPT-4. Table 2 describes the statistics of our generated datasets. We provide examples from each dataset in Appendix F.

We do not aim to formally evaluate fluency or coherence of our generated stories. GPT-4 generates stories that are, based on our inspection, very strong according to these attributes. We also do not evaluate intrinsic "sensibility" of commonsense, which we also found to be very high; we opt instead to evaluate this in an end-to-end fashion based on whether humans can correctly reason about the right answer.

Table 2: Dataset statistics for MuSR, including the number of instances, number of steps, number of commonsense facts, and performance of a rule-based system on the domain.

|  | Size | # Steps | # CS | Rule-based |
|---|---|---|---|---|
| Murder Mystery | 250 | 10 | 9 | 50.0 |
| Object Placements | 256 | 11 | 6 | 35.9 |
| Team Allocations | 250 | 10 | 9 | - |

**Rule-based Performance**  Table 2 also shows the performance of two rule-based systems we developed to sanity-check our datasets. The Murder Mystery rule baseline looks for which suspect has the longest chapter in the context. Object Placements looks for the location that is mentioned the most. We find that each of these is near random chance (reported in Table 5).

**Human performance**  To validate that the answers derived from $F$ actually match what can be derived from our narratives, we conducted a human evaluation of dataset quality. A total of 7 annotators were used, 4 of whom were authors of the paper and 3 of whom were hired undergraduate students not familiar with the datasets. Annotators were given the exact "chain-of-thought+" prompt that we evaluated the LLMs with, described in the next section.

We triply-annotated between 34 instances for Murder Mystery and Team Allocation and 40 instances for Object Placements. Table 3 displays the annotators' scores broken down by the lowest, highest, and average scores for each annotator; the average is based on all (instance, annotator) pairs, over 100 for each domain. Our best annotator across all domains was

Table 3: A granular view of the human annotation scores for each domain including the lowest score, highest score, average score, and the majority vote score. No model or prompt variant scores higher than any of our annotators.

|  | Lowest | Highest | Average | Majority |
|---|---|---|---|---|
| Murder Mystery | 88.2 | 94.1 | 92.1 | 94.1 |
| Object Placements | 85.0 | 95.0 | 90.0 | 95.0 |
| Team Allocations | 91.1 | 100.0 | 95.1 | 100.0 |

one of the undergraduate students not familiar with the dataset construction procedure. We also display the majority annotation. Crucially, the majority is higher than the average annotator, showing that many annotator errors are simply due to inattentiveness and our panel of three annotators is able to collectively arrive at the right answer via voting. Overall, we believe this majority is reflective of the ceiling for human performance on these datasets, demonstrating that it is very high.

**Ablating our creation process**  Finally, we aim to establish that the procedure we have presented so far in this paper is in fact necessary to produce high-quality data. Table 4 shows a set of ablations on our construction procedure on 25 examples per domain, measured with several metrics. First, we track the length (**Len**) and diversity (**Div**) of the context, measured by self-BLEU of a sentence from one narrative compared with all other narratives. We also compute Fact Recall (R), which is a percentage of the number of facts entailed in the context from the gold reasoning trees leaf nodes, using GPT-4 to check for entailment of each fact. Finally, we evaluate GPT-4's performance. Although our goal is to make a challenging dataset, in the context of this table, low GPT-4 performance usually means that the examples are ambiguous or unsolvable.

Basic prompting for any domain yields extremely short stories (often ten sentences in length). They are also usually very similar. The GPT-4 performance is quite low; anecdotally, we found these stories to have spurious solutions. Using diversity sampling to improve the reasoning gives a better

Table 4: Variations of our dataset creation process. We compare against a simple one-shot prompting approach and an approach using seed facts $G$ to add diversity, which produce simple and poor-quality narratives. We then ablate chaptering and tree validators, showing that these lower length, fact recall in the narrative, and accuracy; the latter usually indicates inconsistent narratives.

| | Murder Mysteries | | | | Object Placements | | | | Team Allocation | | | |
|---|---|---|---|---|---|---|---|---|---|---|---|---|
| Ablation | Len | Div | R | Acc | Len | Div | R | Acc | Len | Div | R | Acc |
| Prompt Only | 280 | 0.30 | - | 76 | 200 | 0.26 | - | 64 | 172 | 0.34 | - | 80 |
| Diversity Sampling | 422 | 0.25 | - | 60 | 404 | 0.24 | - | 39 | 448 | 0.26 | - | 84 |
| MuSR − chapt − validators | 428 | 0.24 | 67 | 60 | 380 | 0.27 | 83 | 78 | - | - | - | - |
| MuSR − validators | 924 | 0.24 | 93 | 60 | 793 | 0.25 | 82 | 65 | - | - | - | - |
| MuSR | 900 | 0.25 | 95 | 84 | 777 | 0.25 | 87 | 58 | 503 | 0.25 | 81 | 68 |

set of reasoning examples including a minor boost in length, but again, the problems are not always solvable, nor are the solutions always consistent with the underlying ground truth.

When we introduce reasoning trees (the three MuSR variants), we can see GPT-4's performance still remains low. This is because prompting GPT-4 to generate a story from all the facts often leads to shorter stories and can elide facts: only 62% of the facts from the original reasoning trees are entailed in the resulting story for murder mysteries. By introducing "chaptering," we can see that fact recall increases and the story length nearly doubles in size while maintaining high diversity. Finally, the added tree validators to ensure the reasoning tree is constructed according to a set of rules (like not mentioning key items in deductions) fact recall increases slightly and GPT-4's performance increases substantially for Murder Mystery. Team Allocation did not require chaptering or validators to create good examples and thus has no ablations for these components.

## 5.2 BENCHMARKING WITH MUSR

We now evaluate a series of LLMs (Brown et al., 2020; OpenAI, 2023; Touvron et al.; Chiang et al., 2023) with multiple prompting strategies. Specifically, we compare single-shot prompting, chain-of-thought (Wei et al., 2022, CoT), and a variant of chain-of-thought we call "CoT+". CoT+ uses an engineered textual description of the domain's reasoning strategy described in Section 3. Prompts for CoT+ can be seen in Appendix I.1 Finally, we test multiple neurosymbolic algorithms on domains that best match the settings those algorithms were designed for.

Table 5: Scores for LLMs on each domain in MuSR as well as the human evaluation using the CoT+ strategy.

| | MM | OP | TA |
|---|---|---|---|
| random | 50.0 | 24.6 | 33.3 |
| GPT-4 | 80.4 | 60.9 | 68.4 |
| GPT-3.5 | 61.6 | 46.9 | 40.4 |
| Llama2 70b Chat | 48.8 | 42.2 | 44.8 |
| Llama2 7b Chat | 50.8 | 29.3 | 36.8 |
| Vicuna 7b v1.5 | 48.4 | 29.7 | 26.4 |
| Vicuna 13b v1.5 | 50.8 | 34.4 | 32.0 |
| Vicuna 33b v1.3 | 49.6 | 31.2 | 30.0 |
| Human Eval | 94.1 | 95.0 | 100.0 |

**Zero-shot results on LLMs** We first focus on the ability of large language models to solve this dataset zero-shot, given only the prompt. We constructed the dataset with this scenario in mind, but also evaluate a 1-shot prompt in Table 7.

Table 5 shows results over our LLMs with the CoT+ prompt as well as human performance. Llama 2 and Vicuna-based language models are able to get above chance for each domain but only slightly. Although these models are often compared to GPT variants, they are unable to surpass GPT-3.5 on two out of the three domains, with Team Allocation being the only one where the Vicuna models outperform slightly. GPT-4 performs the best out of all the models we tested, but still underperforms compared to humans. Although GPT-4 was instrumental in creating this dataset, it does not have the reasoning capabilities to solve it end-to-end. A small qualitative analysis of some of the error classes exhibited by the GPT-3.5-turbo and GPT-4 are discussed in Appendix D.

**Results on Prompt Variants** Table 7 shows GPT-3.5 and GPT-4, the two models that did best on MuSR, evaluated with different prompting strategies. Overall, the best performance is seen when the model is given a single-shot example with the "1-shot CoT+" or "Few-shot CoT+" prompt variants. However, adding more examples is not always better. Despite significant jumps in performance on some domains, the models still underperform compared to the human majority.

Table 7: Evaluations of different popular prompting strategies for GPT-3.5 and GPT-4, our strongest models. "Regular" supplies only the context and question. "CoT" asks the model to think step-by-step. "CoT+" includes a textual description of the reasoning strategy, and "1-Shot CoT+" includes a solved example. "Few-Shot CoT+" extends "1-Shot CoT+" with 3 examples (3 examples hits the token limit for GPT-4)

|  | Murder Mystery | | Object Placements | | Team Allocation | |
|---|---|---|---|---|---|---|
|  | GPT-3.5 | GPT-4 | GPT-3.5 | GPT-4 | GPT-3.5 | GPT-4 |
| Regular | 59.2 | 64.8 | 44.5 | 43.0 | 41.2 | 64.0 |
| CoT | 56.0 | 65.6 | 48.4 | 41.8 | 46.4 | 64.4 |
| CoT+ | 61.6 | 80.4 | 46.9 | 60.9 | 40.4 | 68.4 |
| 1-Shot CoT+ | 70.0 | 86.0 | 56.2 | 72.3 | 50.4 | 88.4 |
| Few-Shot CoT+ | 68.4 | 84.8 | 58.2 | 71.5 | 78.8 | 89.6 |

**Results on Neurosymbolic Approaches** We believe that this dataset is an ideal testbed for different neurosymbolic approaches. Besides basic chain-of-thought, we are not aware of a single approach that naturally handles all the reasoning types in our dataset and scales to examples of the difficulty we present. As a result, we present three different methods in Table 6 each tailored to one domain and evaluated in that domain. We describe these approaches here and in Appendix E

In the Murder Mystery domain, we implement a variation of Decomposed Prompting (Khot et al., 2023) by manually imposing the breakdown of motive, means, and opportunity and prompt GPT-4 to decide on each suspect for each fact. We then decide the murderer based on who has the most facts proving guilt, with random selection in case of ties. Despite aligning well with the reasoning strategy, the accuracy is lower than prompting GPT-4 end-to-end.

Table 6: Scores for a selection of reasoning systems on the domain that best fit their capabilities.

| Murder mysteries | |
|---|---|
| GPT-4 CoT+ | 80.4 |
| Decomposed Prompting | 77.6 |
| Decomposed Prompting 1-Shot | 86.0 |
| Object Placements | |
| GPT-4 CoT+ | 60.9 |
| SymbolicTOM | 23.8 |
| Team Allocation | |
| GPT-4 CoT+ | 68.4 |
| PAL | 77.2 |
| PAL 1-Shot | 87.2 |

Next we used SymbolicTOM (Sclar et al., 2023) on the Object Placements domain with minor adjustments. Specifically, we use GPT-3.5 to produce the resulting state of a sentence that is then used in the graph creation algorithm. The low accuracy of SymbolicTOM is mostly attributed to selecting key entities from sentences that are not as templated as the original dataset (Le et al., 2019). Because the contexts are more natural, entities' actions and observations can span multiple paragraphs rather than be isolated in one sentence. This introduces a new level of complexity for these neurosymbolic methods, and past approaches on ToM cannot generalize here.

Finally, we run a variant of Program-Aided Language Models (Gao et al., 2022) on the Team Allocation domain. From the story, PAL must deduce numerical values for the skill and teamwork levels of each person and pair. Once this is done, we give it a description of the reasoning strategy for Team Allocation, which it implements in a program and solves returning the assignment with the highest score. We find that this solution pairs quite well with the domain, outperforming the end-to-end models on both zero and single-shot settings, but falling short of aggregate human performance.

## 6    CONCLUSION

In this paper, we introduced Multistep Soft Reasoning (MuSR) a reasoning dataset written with natural narratives presenting complex reasoning scenarios involving various reasoning strategies. We presented a neurosymbolic dataset generation method for constructing instances of our dataset, which can be scaled in complexity as more powerful models emerge. Human evaluation and other intrinsic validations shows that the construction method is sound for sufficiently large models. Our results show that LLMs are currently unable to match human performance on specific types of reasoning like multi-step and commonsense in our three domains. This dataset presents a challenge for both the largest and smaller language models: we believe it can serve as (1) a benchmark for LLMs; (2) a benchmark for general neurosymbolic approaches over language; (3) a general construction procedure for generating challenging datasets as models improve.

# 7 REPRODUCIBILITY OF MUSR

To aid in reproducing the datasets for each domain in MuSR, we've included high-level details of the construction procedure in Sections 3 and 4. We further detail each domain's reasoning strategy and algorithms as well as give the prompts verbatim for all parts of the construction procedure in Appendix G. Implementation details, including hyperparameters and model design choices, can be found in Appendix H. For our neurosymoblic evaluations, we provide relevant details on their implementations in Section 5.2 and with further detail in Appendix E. Finally, all data will be publicly available, including the code used to generate and evaluate the dataset in future versions of this paper.

## ACKNOWLEDGMENTS

This work was supported by NSF CAREER Award IIS-2145280, a grant from Open Philanthropy, and by the NSF AI Institute for Foundations of Machine Learning (IFML). This material is also based on research that is in part supported by the Air Force Research Laboratory (AFRL), DARPA, for the KAIROS program under agreement number FA8750-19-2-1003. Thanks to Kathryn Kazanas and Keziah Reina for providing human judgments on MuSR. Thanks to Juan Diego Rodriguez and members of the UT TAUR lab for helpful discussion and feedback.

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

# A MuSR Limitations

## A.1 Tree Construction

Although our method can create reasoning trees of varying depths, we found that shallower trees (of depth one or two) provide the best level of detail for creating a narrative. GPT-4 often failed to create complex enough facts that could be broken down recursively to a larger depth. We believe that prompting and better LLMs may increase the depth of acceptable deductions and is an important area of future work for our method.

## A.2 Human Evaluation

We experimented with validation on Amazon Mechanical Turk, but found that many workers performed very badly in qualification rounds. When we collected justifications to try to improve the quality of their judgments, we found many justifications which we suspected were written by Chat-GPT.

# B Dataset Features Explained

In this section, we elaborate on the features employed to evaluate the datasets as illustrated in Table 1.

**Natural Text:** This denotes datasets containing organically constructed text, not created by templates. For instance, bAbI generates text by filling in predefined templates. True Detective is human-authored. Datasets like EntailmentBank and ENWN, while incorporating natural text, present them in specialized "premises" rather than contexts, hence the notation $\sim$. SocialIQA uses ATOMIC to create templates for the questions, and while it aims to test commonsense reasoning in social situations, the questions and answers might not always reflect natural English. MuSR produces natural contexts with language prompted from an LLM without any templating.

**Commonsense:** Refers to datasets that require commonsense knowledge to answer questions. EntailmentBank and RuleTakers supply all the necessary facts to answer a question in the input, requiring no commonsense. BigTOM is harder to classify as nearly all the facts required to answer the question are given, but understanding what it means for someone to have a "belief" could require non-trivial commonsense from the reader. MuSR intentionally omits certain commonsense facts during its construction, compelling users to draw on their inherent knowledge for problem-solving.

**Multistep:** Denotes datasets requiring a layered reasoning approach to deduce answers. Each reasoning layer involves merging multiple premises to generate interim conclusions, which then contribute to the final inference. SocialIQA is not designed to require such intermediate conclusions. In contrast, MuSR, through its design, compels users to recreate the reasoning tree used in the questions creation (or something similar to it) for comprehensive understanding.

**Intermediate Structures:** This captures datasets with underlying structure (chains of facts, etc.) that can potentially assist in deducing answers. True Detective was written by humans and thus lacks an intermediate structure. MuSR has the reasoning trees used to create each example given for every question.

**Not solvable with rules:** This category represents datasets resilient to systematic, rule-based solutions without the need for a language model. Datasets like Babi and ToMi, given their textual templates, may reveal patterns that can be reverse-engineered to facilitate solutions. Contrarily, datasets such as MuSR, True Detective, and SocialIQA lack easily identifiable patterns, safeguarding them against oversimplified, template-based resolutions.

Table 8: Results of prompting LLMs to solve the 50 murder mysteries created by GPT-3.5-turbo. The 1-shot example was taken from a murder mystery example created by GPT-4 and solved by a human annotator.

|  | Murder Mystery | |
| --- | --- | --- |
|  | GPT-3.5 | GPT-4 |
| CoT+ | 62.0 | 80.0 |
| 1-Shot CoT+ | 64.0 | 76.0 |

## C    CREATING MuSR WITH OTHER MODELS

Quality MuSR examples require a language model that can follow prompt instructions. A model with more limited reasoning ability may not be able to perform the subtasks needed to construct MuSR examples adequately. We highlight this by exploring GPT-3.5-turbo, Llama2-70B-Chat (Touvron et al.), and Mixtral (Jiang et al., 2024) to create MuSR examples.

For weaker models to create examples approaching the quality of GPT-4, we had to minorly edit the prompts, introduce more detailed self-refine prompts, increase the temperature iteratively per retry, and increase the total number of retries.

We created 50 murder mysteries using GPT-3.5-turbo. The results of the CoT+ and 1-Shot CoT+ systems running on these are shown in Table 8. The overall performance of LLMs on them is similar to our main MuSR dataset.

Despite being able to use the same workflow as GPT-4 and producing some examples of high quality, many still include major flaws in the reasoning tree that do not stand up to our inspection. For example, some exhibit common reasoning errors seen in smaller language models like hallucination. Examples of invalid deductions from GPT-3.5-turbo that were erroneously considered "valid" in the workflow are shown in Listing 1.

Listing 1: Murder Mystery Logic Tree from GPT-3.5-turbo

*Penelope is a chemist. | Deduced Fact*

> *Penelope has studied toxicology. | Fact From Story*

> *Penelope has a background in chemistry. | Fact From Story*

> *If someone has a background in chemistry and has studied toxicology, they have knowledge of toxic gases and their effects, providing them with the means to create and use poisonous gas as a murder weapon. | Commonsense Knowledge*

*...*

*Tessa and Melody were alone in the kitchen at the time of the murder. | Deduced Fact*

> *Tessa had a life insurance policy worth a significant amount of money. | Fact From Story*

> *Melody is a police officer. | Fact From Story*

> *When someone is alone with a victim and the victim has a life insurance policy, it provides an opportunity for murder. | Commonsense Knowledge*

Llama2-70B-Chat and Mixtral were incapable of following the prompt instructions and could not complete the deductions required to create MuSR examples at all, despite significant edits to the format of the construction prompts to match their expected formats. Two examples of Llama2-70B-Chat failing to understand the prompt and produce a deduction are shown in Listing 2

Listing 2: Llama2-70b-Chat failure outputs for creating reasoning tree deductions

*Please provide the next two explicit facts to continue the deduction.*

> ...
>
> *has a motive.* | *Deduced Fact*
> *> > Penelope wants to protect a secret.* | *Fact From Story*
> *> > Protecting a secret can be a strong motive for murder.* | *Commonsense Knowledge*
> *> Penelope has an opportunity.* | *Deduced Fact*
> *> > Penelope was in the victim's home.* | *Fact From Story*
> *> > Being in the victim's home provides an opportunity to release the poisonous gas, which could lead to the victim's death.* | *Commonsense Knowledge*
>
> *Deduction to complete:*
> *Penelope has a means.*
> *> ?*
> *> ?*
> *> Commonsense Knowledge*

While further prompt engineering or using stronger models for validation (as opposed to generation) could lead to more effective generation for these smaller models, we believe that using the strongest LLMs is the best practice for creating a dataset like this.

## D  QUALITATIVE ERROR ANALYSIS

Five examples from each domain were selected and used to generate chains-of-thought from GPT-4 and GPT-3.5-turbo. Each of those chains-of-thought were then examined by hand and annotated for errors. Below are the major categories of errors we saw in the GPT models when solving MuSR problems.

**Pre-committing to an answer:**  A large portion of the answers suffer from giving an answer before the reasoning, which biases the subsequent reasoning. A more subtle error but still frequent in the responses is "silently" pre-committing to an answer where the model will produce only relevant evidence for one answer and not the other, leaving out important and relevant facts that should be included.

**Ignoring instructions:**  Many responses included reasoning and logic that went against the prompt details, i.e., selecting a murderer on a strong motive solely because the strength of that motive rather than the definition of a murderer being someone with a motive, means, and opportunity. Team allocation also has multiple examples of the model asserting that two people can work together despite being assigned to different jobs, contradicting the prompt.

**Hallucination and invalid logic:**  Hallucinations appear in various ways within the chains of thought, including assumptions of characters' locations or actions as well as confusing pronouns and entity relations, i.e., confusing what one person did with another person. Furthermore, invalid claims or reasoning, i.e., stating unsound deductions or stating deductions with no evidence, are frequent as well. A few times these hallucinations and invalid claims would lead the model to contradict their previous logic resulting in contradicting answers that did not align with the reasoning trace.

## E  STRUCTURED BASELINE IMPLEMENTATION DETAILS

### E.1  DECOMPOSED PROMPTING

We test Decomposed Prompting (Khot et al., 2023) on the Murder Mystery domain. As each murder mystery can be solved using the same high-level logical procedure (determining which suspect has or is most likely to have a means, motive, and opportunity), we omit the Decomposer Prompt stage, and use a fixed set of three sub-task prompts, each of which are specialized to identify one of means, motive, or opportunity. The results of these sub-task prompts are then aggregated to determine the system's overall answer; in the event that both suspects are predicted to satisfy the same number of criteria, the system guesses at random between them.

### E.2 SYMBOLICTOM

We test SymbolicTOM (Sclar et al., 2023) on the Object Placements domain. We use the model as implemented by the authors with two changes. First, the resulting states that are used in creating the graph are not precomputed and are instead queried on-the-fly using GPT-3.5. Second, if the model abstains from answering, we randomly sample one of the answer choices. The data format originally intended for SymbolicTOM was the format for ToMI (Le et al., 2019); however, our data is easily translated into the format for SymbolicTOM through simple manipulation.

### E.3 PAL

We test program-aided language models (Gao et al., 2022) on the Team Allocation domain. Given a question, we first prompt LLMs to generate a Python program, and then execute the program using the Python interpreter to obtain the final answer. We test PAL in both zero-shot setting and one-shot settings.

For the zero-shot setting, we provide detailed instructions on how the program should be organized for solving a question. As shown in Listing 32, the prompt asks for a program containing three steps: (1) assign a value to the variables representing each person's skill level on one of the two tasks; (2) assign a value to the variables representing how well two people work together, and (3) compute the scores for each of the options by adding up the scores for each person's skill level and the teamwork score. For the one-shot setting, we provide one demonstration of the question-program pair in addition to the detailed instructions, which leads to better performance compared to the zero-shot setting.

## F DATASET EXAMPLES

### F.1 MURDER MYSTERIES

Listing 3: Murder Mystery Example 1

> *In an adrenaline inducing bungee jumping site, Mack's thrill−seeking adventure came to a gruesome end by a nunchaku; now, it's up to Detective Winston to unravel the deadly secrets between Mackenzie and Ana.*
>
> *Winston took a gulp of his black coffee, staring at the notes sprawled across his desk. A murder case at a bungee jumping site was definitely out of the ordinary. Today's victim was a young man named Mack, loud mouthed and cocky by all accounts.*
>
> *Mack was bungee jumping the day he was killed. Oddly enough, according to the records, no one else was documented at the bungee jumping site that day, making this case even more peculiar. The first stop for the day was to visit one of Mack's housemates, a woman named Ana. They were seen leaving in the same vehicle from their shared housing complex the morning of the murder, and it was time for Winston to dig deeper.*
>
> *As he pulled into the shared housing driveway, a nondescript car came into sight. He learned from neighbours that it was frequently used by multiple residents, but Ana had a peculiar interest in it. She would insist on driving whenever with a group of friends, later meticulously cleaning the car after each use. An idiosyncrasy of hers maybe, but a part of the puzzle nonetheless.*
>
> *Winston knocked on the door, Ana opened it warily, twiddling a cleaning cloth and spray in her hands and greeted him with a nervous nod. Ana gets nervous and fidgets with the cleaner and cloth when questioned. Winston could sense palpable unease as he started asking her questions.*
>
> *"Ana, did you not join Mack and the others for bungee jumping today?" Winston questioned, to which she responded, "I signed up to jump. But I didn't end up going through with it."*
>
> *"Any particular reason you didn't join the others, Ana?" Winston proceeded.*
>
> *Ana took a deep breath, "Well sir, my faith doesn't really permit bungee jumping. Truth be told, I was persuaded strongly by Mack. I had even signed up out of peer pressure but couldn't push myself."*

*It was true − Mack was insisting that everyone in the group should bungee jump. Mack had reportedly also been vocal about ridiculing Ana's faith, even encouraging others to join him in doing so. It was a significant factor in their relationship.*

*"Ana, did you and Mack leave in the same car for the bungee jumping event this morning?" Winston gently pushed further.*

*"Yes. Yes, we did. We always carpool." She responded while anxiously using the cleaner and cloth on her car's dashboard. Her eyes flickered nervously back to Winston, expecting the next question.*

*Winston took a deep breath, standing up to leave, "Alright Ana, that should cover everything for now. We'll be in touch."*

*Ana nervously nodded without looking up from her cleaning, wringing the cloth repeatedly as Winston walked away, left again with another piece to the enigmatic puzzle of Mack's murder.*

*The day was getting older and Winston was getting more tired, but the case was fresh, and he wasn't one to back down. He tugged on his coat as he approached the bashful teen waiting for him by the police station.*

*"Mackenzie, it is?" he asked, extending his hand.*

*"Yeah, that's right." The slight lisp, overlaid with blanket anxiety, confirmed what the school reports suggested.*

*"You were at the site when Mack... erm... you know," Winston's voice was methodical, calm −− almost robotic. The suspicion on Mackenzie was not unfounded − the security cameras showed him buying nunchaku a week before.*

*Mackenzie shifted on his feet, looking away before answering, "Yeah, I was there."*

*Winston pulled out a small notebook, "What were you doing there, Mackenzie?"*

*"Bungee jumping, like Mack... Then I left. I didn't... I didn't do anything..." Mackenzie replied.*

*Internally, Winston sighed at the never−ending waterfall of teenage angst this case was turning into.*

*"Martial arts, huh?" Winston segued, gesturing to a bruise on Mackenzie's knuckles. "Nunchaku particularly, I see? Training does include the use of those, correct?"*

*The change in Mackenzie's demeanor mirrored the bitterness in the last month's weather − dark eyes replaced with ice−cold ones. "Yeah," he admitted, shrinking slightly.*

*Mackenzie always took pride in being the best at everything. So when Mack got everything he wanted − the promotion to team captain, the respect, the attention − it was a hard pill for Mackenzie to swallow. Winston remembered the team talk, Mackenzie was indeed the top candidate but it had gone to Mack instead.*

*What clinched it was Mackenzie's remarks about Mack, echoing whispers of dispute and bickering, lost in the crowded lunchroom. There were also multiple witness reports of the two seen arguing at the bungee jumping site previously. Mackenzie had indeed said disparaging, almost emotional things about Mack − all stemming from a potent brew of jealousy, Winston inferred.*

*Shifting later through the detritus of Mackenzie's life, Winston discovered the nunchaku that matched the forensics report. They were tucked away, but the layer of dust suggested they weren't a favored possession anymore. It wasn't hidden, it was misplaced − discarded in the throes of developing maturity.*

*As the sun started to set, Winston could see witnesses, scattered across the park, repeatedly pointing to the bungee jumping scaffolding. It occurred to him, then, the narrative of the past days. Mackenzie, jealous and wronged, over and over, at the same sight. It was quite a sight.*

*Winston, shuffling back to the station, was left with one thought − Looks like Mackenzie had quite an eventful week.*

*Who is the most likely murderer?*

*Pick one of the following choices:*

*1 − Mackenzie*
*2 − Ana*

*You must pick one option. Before selecting a choice, explain your reasoning step by step. The murderer needs to have a means (access to weapon), motive (reason to kill the victim), and opportunity (access to crime scene ) in order to have killed the victim. Innocent suspects may have two of these proven, but not all three. An innocent suspect may be suspicious for some other reason, but they will not have all of motive, means, and opportunity established.*

*If you believe that both suspects have motive, means, and opportunity, you should make an educated guess pick the one for whom these are best established. If you believe that neither suspect has all three established, then choose the suspect where these are most clearly established. Explain your reasoning step by step before you answer. Finally, the last thing you generate should be "ANSWER: (your answer here, including the choice number)"*

Listing 4: Murder Mystery Example 1 Reasoning Tree

*Mackenzie is the murderer. | Deduced Root Conclusion*
*> Mackenzie has a means. | Deduced Fact*
*> > Mackenzie is highly skilled with nunchaku. | Deduced Fact*
*> > > Martial arts training includes nunchaku techniques. | Fact From Story*
*> > > Mackenzie practices martial arts, including nunchaku. | Fact From Story*
*> > > A person who practices martial arts, specifically nunchaku, can become highly skilled at using it. | Commonsense Knowledge*
*> > Mackenzie owns nunchaku. | Deduced Fact*
*> > > Nunchaku were found in Mackenzie's house. | Fact From Story*
*> > > Mackenzie was seen purchasing nunchaku. | Fact From Story*
*> > > When a person is seen purchasing a weapon and the same weapon is found in their possession, they own the weapon. | Commonsense Knowledge*
*> > If a person owns and is proficient in using nunchaku, they have means to use it as a weapon in a crime. | Commonsense Knowledge*
*> Mackenzie has an opportunity. | Deduced Fact*
*> > Mackenzie had an altercation with Mack at the bungee jumping site in the past. | Deduced Fact*
*> > > There are witnesses that saw Mackenzie and Mack arguing at the bungee jumping site previously. | Fact From Story*
*> > > Mackenzie was seen at the bungee jumping site on the day of the murder. | Fact From Story*
*> > > Being at the location of a crime when it is committed and having a history of confrontation with the victim at that location provides the opportunity to commit a crime. | Commonsense Knowledge*
*> > Mackenzie was at the bungee jumping site at the time of the murder. | Deduced Fact*
*> > > Mackenzie admitted being at the site during the murder. | Fact From Story*
*> > > Witnesses saw Mackenzie at the site when the murder happened. | Fact From Story*
*> > > If someone is seen at the scene of a crime when it happens and they admit to it, they were present during the crime, which provides opportunity. | Commonsense Knowledge*
*> > Someone with a history of altercations at a specific location who is also present at the time of a crime in that location potentially has the opportunity to commit the crime. | Commonsense Knowledge*
*> Mackenzie has a motive. | Deduced Fact*
*> > Mack got the promotion that Mackenzie always wanted. | Deduced Fact*
*> > > The promotion went to Mack instead of Mackenzie. | Fact From Story*
*> > > Mackenzie was the top candidate for the promotion. | Fact From Story*
*> > > Losing a much−anticipated promotion to a rival can lead to extreme resentment and provide a motive for harmful actions. | Commonsense Knowledge*
*> > Mackenzie was jealous of Mack. | Deduced Fact*
*> > > Mack has things that Mackenzie has always wanted. | Fact From Story*
*> > > Mackenzie was overheard saying disparaging and jealous things about Mack. | Fact From Story*
*> > > Extreme jealousy can motivate someone to eliminate what they see as the cause of their unhappiness, thereby providing a motive for murder. | Commonsense Knowledge*
*> > Intense jealousy can motivate someone to eliminate their rival. | Commonsense Knowledge*
*Ana is the murderer. | Deduced Root Conclusion*
*> Ana has an opportunity. | Deduced Fact*
*> > Ana was traveling in the same car as Mack to the bungee jumping site. | Deduced Fact*
*> > > No one else was documented at the bungee jumping site that day. | Fact From Story*
*> > > Ana and Mack were seen leaving in the same vehicle from their shared housing complex the morning of the murder. | Fact From Story*
*> > > Being the only two people at a location gives you the opportunity to commit a crime there without witnesses. | Commonsense Knowledge*

> > Ana was also at the bungee jumping site the same day as Mack. | *Deduced Fact*
> > > Ana had also signed up for bungee jumping that day. | *Fact From Story*
> > > Mack was bungee jumping the day he was killed. | *Fact From Story*
> > > Signing up for the same activity at the same location as the victim gives you the opportunity to be at the crime scene. | *Commonsense Knowledge*
> > Travelling together to the same location gives you the opportunity to commit a crime there. | *Commonsense Knowledge*
> Ana has a motive. | *Deduced Fact*
> > Ana frequently argued with Mack over their differing views on religion. | *Deduced Fact*
> > > Mack was insisting that everyone in the group should bungee jump. | *Fact From Story*
> > > Ana's religion forbids bungee jumping. | *Fact From Story*
> > > A deep commitment to a religious doctrine that considers certain actions immoral or sinful can lead to strong emotional reactions, including violence, when defiance of that doctrine is insisted upon. | *Commonsense Knowledge*
> > Mack was a vocal critic of Ana's religious beliefs. | *Deduced Fact*
> > > Mack encouraged others to ridicule Ana's religious faith. | *Fact From Story*
> > > Mack repeatedly ridiculed Ana's religious faith. | *Fact From Story*
> > > Persistent ridicule and disrespect of one's closely held beliefs could lead to extreme reactions, including violence. | *Commonsense Knowledge*
> > Strong religious beliefs can lead to extreme actions, such as harm or murder, when those beliefs are threatened or disrespected. | *Commonsense Knowledge*
> Keeps their car obsessively clean. And this is suspicious. | *Deduced Fact*
> > Ana constantly keeps a bottle of car cleaner and cloth in their pocket. | *Deduced Fact*
> > > Ana uses the cleaner immediately after anyone else uses the car. | *Fact From Story*
> > > Ana gets nervous and fidgets with the cleaner and cloth when questioned. | *Fact From Story*
> > > People who are unusually careful about a specific object often have an emotional or secretive connection to that object. | *Commonsense Knowledge*
> > Ana cleans the car immediately after every use. | *Deduced Fact*
> > > Ana insists on driving whenever with a group of friends. | *Fact From Story*
> > > Ana's car is used by multiple people very frequently. | *Fact From Story*
> > > People often clean things used by many individuals to ensure it remains in their control and to remove any trace of others. | *Commonsense Knowledge*
> > People who obsessively clean something usually have something to hide or they are trying to remove evidence. | *Commonsense Knowledge*

Listing 5: Murder Mystery Example 2

In the haze of neon lights and the serving of a silent hand of fate, Timothy lies dead in a casino, a sai his cruel end, leaving the unruffled Detective Winston to interrogate suspects, Harry and Rosemary.

It had been a long day for Winston. The air was heavy with the scent of fresh coffee and the clamour of a bustling restaurant kitchen. His eyes fell on a seasoned chef, Rosemary, as she deftly wielded her bladed tools − knives, cleavers, graters − with calm precision. Watching her, it came as no surprise that Rosemary had clocked several years in this industry.

Something in the room changed. Shouting ensued, then a loud crash that rang out above the normal kitchen discord. Rosemary had hurled a metal pot across the room. The assistant, who stood close by, looked shocked but unharmed. Winston decided it was his cue to intervene.

"Rosemary, care to explain what just happened?" Winston asked, stepping closer to the irate chef.

She gave him a guarded look before deliberately changing the subject, "Did you know Timothy was a fan of my stir fry? Ironic, isn't it?"

Winston frowned slightly at the statement but decided to push forward. He knew how to dance around subjects, but Rosemary seemed skilled at the bucolic ballet of the restaurant business.

"I've heard some disturbing claims, Rosemary," Winston brought out his notebook, "about the threats you've been issuing to Timothy, and your hostility towards people of his nationality."

At Winston's words, Rosemary ran a weary hand over her face and sighed. "Seems word gets around."

"A public event, not long ago. You spoke openly about your, um−" Winston glanced down at his notes, "−' distaste' for Chinese folks," he pressed on, "and you've been caught on tape making similar remarks towards Timothy."

*"Is that a crime, detective?" Rosemary challenged.*

*"I'm just here to piece the puzzle together. I understand you take a particular interest in Asian culture − antique Asian weapons in particular. I've seen your collection, Rosemary. Sais, even?" he prodded, hoping for a reaction.*

*Rosemary's gaze sharpened as she turned her back on him, busily cleaning her array of kitchen knives. She didn't confirm nor deny his observation. Noting her silence, Winston thanked her for her time and walked out onto the casino floor, a maelstrom of thoughts whirling around his mind. He felt like he was leaving with more questions than when he had entered.*

*Winston took a good look at the crime scene, a corner of the bustling casino, cordoned off by the police tape. Something felt grimly out of place among the bright lights and incessant chatter of the casino. He carefully sifted through the conflicting information and people's statements spinning in his head.*

*Time to get some answers, Winston thought, and made his way to his interviewee.*

*It was late in the day when he finally knocked on Harry's door. A man in his early thirties, with a life− hardened face glanced out at him skeptically.*

*"Harry, correct?" Winston asked.*

*"And who's asking?" came the guarded reply.*

*"Detective Winston," he flashed his badge, "I'm here to ask you a few questions about Timothy."*

*Harry's eyes flashed, "I'm not surprised," he grumbled. "Come on in then."*

*As Winston made his way inside, he noticed the place bore a striking resemblance to traditional dojo settings. A pair of sai swords caught his eye, arranged carefully on a display holder. A typical weapon of the martial arts form Harry used to instruct.*

*"Nice collection." Winston gestured towards the sai. "You instruct?"*

*Harry looked back at the sai, "Used to."*

*Harry's manner was gruff, but he seemed at home sharing his old days as a martial arts instructor. They talked about martial arts, how Harry won several competitions, his daily training routine, which apparently included practicing with the sai regularly. Harry's days as a horse trainer surfaced later in the conversation.*

*"Got dealt a bad hand?" Winston inquired casually, nodding at the pile of losing horse race betting slips on Harry's coffee table.*

*Harry grunted, "Yeah, you could say that."*

*Winston knew Harry only had income from betting on races, and recently he had lost quite a few. Harry had a deep gambling debt with Timothy over his betting habits. Photography was not Winston's hobby, but he recalled Harry's face distinctly in the casino cameras' footage from before the murder took place. There were rumors that Timothy was planning to expose Harry's debt to the other horse owners, and the situation got tough.*

*"Got into any recent arguments?" Winston asked.*

*Harry frowned and averted his eyes, "Maybe...just one with Timothy at the casino."*

*Winston nodded, keeping his expression neutral. The timing was unfortunate, he thought. And that debt wasn't going anywhere, especially with Harry having recently lost his job at the stables.*

*"Heard you were giving out loans?' Winston asked.*

*Harry's face stiffened, "He needed money", he replied, explaining that Timothy had lent him a large sum of money specifically for his betting habit, a haunted expression crossing his face.*

*Winston stood up, concluding his visit, "Just one last thing, Harry," Winston queried, "The VIP lounge, in the casino? You're familiar with it, aren't you?"*

*Harry met Winston's gaze, resignation in his eyes, "Used to spend a lot of time there."*

*As Winston exited the apartment, he couldn't shake off the heavy feeling hanging in the air, leaving him with more questions than answers. Good thing he was in a questioning mood.*

*Who is the most likely murderer?*

*Pick one of the following choices:*
*1 − Harry*
*2 − Rosemary*

*You must pick one option. Before selecting a choice, explain your reasoning step by step. The murderer needs to have a means (access to weapon), motive (reason to kill the victim), and opportunity (access to crime scene) in order to have killed the victim. Innocent suspects may have two of these proven, but not all three. An innocent suspect may be suspicious for some other reason, but they will not have all of motive, means, and opportunity established.*

*If you believe that both suspects have motive, means, and opportunity, you should make an educated guess pick the one for whom these are best established. If you believe that neither suspect has all three established, then choose the suspect where these are most clearly established. Explain your reasoning step by step before you answer. Finally, the last thing you generate should be "ANSWER: (your answer here, including the choice number)"*

Listing 6: Murder Mystery Example 2 Reasoning Tree

*Harry is the murderer. | Deduced Root Conclusion*
*> Harry has a means. | Deduced Fact*
*> > The martial art Harry practices uses the sai as a weapon. | Deduced Fact*
*> > > Sai is a commonly used weapon in the martial art that Harry used to instruct. | Fact From Story*
*> > > Harry used to be a martial arts instructor. | Fact From Story*
*> > > If someone used to instruct a martial art that commonly uses a certain weapon, they would know how to use that weapon effectively. | Commonsense Knowledge*
*> > Harry is a skilled martial artist. | Deduced Fact*
*> > > Harry's training routine includes daily practice with the sai. | Fact From Story*
*> > > Harry has won several martial arts competitions. | Fact From Story*
*> > > Proficiency in a martial art and regular practice with a specific weapon implies skill in using that weapon, which could provide the means for murder. | Commonsense Knowledge*
*> > Someone skilled in a martial art that uses a particular type of weapon will know how to use the weapon effectively, thus providing a means for murder. | Commonsense Knowledge*
*> Harry has an opportunity. | Deduced Fact*
*> > Harry is a frequent visitor to the casino due to his love for gambling. | Deduced Fact*
*> > > The murder took place in a secluded area of the casino that Harry is familiar with. | Fact From Story*
*> > > Harry was seen entering the casino before the murder took place. | Fact From Story*
*> > > If someone is familiar with the secluded areas of a building and was present before a crime was committed, it is possible they had the opportunity to commit the crime. | Commonsense Knowledge*
*> > Harry was at the casino at the time of the murder. | Deduced Fact*
*> > > Harry was seen at the casino arguing with Timothy earlier that night. | Fact From Story*
*> > > Harry has a deep gambling debt with Timothy. | Fact From Story*
*> > > A person with a motive, opportunity, means and caught in the proximity of the crime scene at the time has a likelihood of being involved in the crime. | Commonsense Knowledge*
*> > Regular visitors to a location will likely know the layout, the routines and patterns of the location, providing them with the perfect opportunity to commit a crime. | Commonsense Knowledge*
*> Harry has a motive. | Deduced Fact*
*> > Harry was unable to repay this debt. | Deduced Fact*
*> > > His only income was from betting on races, and he had lost his recent bets. | Fact From Story*
*> > > Harry lost his job at the stable. | Fact From Story*
*> > > Losing one's job and dependance on an unstable income, like betting, can lead to inability to repay debts. | Commonsense Knowledge*
*> > Harry owed a substantial amount of money to Timothy. | Deduced Fact*
*> > > Timothy was planning to expose Harry's debt to other race horse owners. | Fact From Story*
*> > > Timothy had lent a large sum of money to Harry for his betting habit. | Fact From Story*

> > > *Exposure of a person's substantial debt could damage their reputation and livelihood, especially if their professional community became aware. This could provide sufficient reason to commit murder to prevent the debt's exposure.* | *Commonsense Knowledge*
> > *A desperate need to escape financial debt can drive a person to commit extreme acts, such as murder.* | *Commonsense Knowledge*
*Rosemary is the murderer.* | *Deduced Root Conclusion*
> *Rosemary has a motive.* | *Deduced Fact*
> > *Timothy's ethnicity is the same as the ethnicity that Rosemary has expressed disdain for.* | *Deduced Fact*
> > > *During a public event, Rosemary verbally expressed her hatred for Chinese people.* | *Fact From Story*
> > > *Timothy was of Chinese heritage.* | *Fact From Story*
> > > *Stereotypes or prejudices against a certain ethnicity can often lead individuals to commit harmful acts towards individuals of that ethnicity.* | *Commonsense Knowledge*
> > *Rosemary has publically made derogatory remarks about Timothy's ethnicity.* | *Deduced Fact*
> > > *Timothy has received threats from Rosemary in the past.* | *Fact From Story*
> > > *During a conversation caught on tape, Rosemary publicly stated her dislike for Timothy due to his ethnicity.* | *Fact From Story*
> > > *People who make public derogatory remarks and threats towards another's ethnicity may be compelled to commit harmful actions, such as murder, against individuals of that ethnicity.* | *Commonsense Knowledge*
> > *Discrimination against a certain ethnicity can lead one to commit harmful actions towards individuals of that ethnicity, providing a possible motive for murder.* | *Commonsense Knowledge*
> *Rosemary has a means.* | *Deduced Fact*
> > *Rosemary has access to bladed weapons like a sai.* | *Deduced Fact*
> > > *Rosemary is fond of Asian culture and collects antique Asian weapons, including sais.* | *Fact From Story*
> > > *Rosemary works in a high−end kitchen that uses various bladed utensils.* | *Fact From Story*
> > > *If a person works with similar tools and collects antiques in line with the murder weapon, they could potentially have access to it.* | *Commonsense Knowledge*
> > *Rosemary is proficient in using bladed instruments due to her role as a chef.* | *Deduced Fact*
> > > *Training and working as a chef involves the use of various bladed tools.* | *Fact From Story*
> > > *Rosemary has years of experience as a chef.* | *Fact From Story*
> > > *Having years of experience using bladed tools provides the skills needed to use a bladed weapon like a sai.* | *Commonsense Knowledge*
> > *Proficiency and access to bladed weapons imply the ability to use a sai effectively, providing the means for murder.* | *Commonsense Knowledge*
> *Has unexplained and sudden mood swings. And this is suspicious.* | *Deduced Fact*
> > *When asked, Rosemary does not explain her anger.* | *Deduced Fact*
> > > *Rosemary refuses to answer and changes the subject.* | *Fact From Story*
> > > *Rosemary was asked about her sudden anger.* | *Fact From Story*
> > > *People who refuse to answer direct questions often have something to hide.* | *Commonsense Knowledge*
> > *Rosemary suddenly snaps at a kitchen assistant.* | *Deduced Fact*
> > > *The assistant did not provoke this reaction.* | *Fact From Story*
> > > *Rosemary threw a pot across the kitchen.* | *Fact From Story*
> > > *People usually don't snap and exhibit violent behavior without provocation unless there are deeper issues.* | *Commonsense Knowledge*
> > *Unexplicable mood swings often raise suspicions about a person's behavior and intentions.* | *Commonsense Knowledge*

## F.2 OBJECT PLACEMENTS

### Listing 7: Object Placement Example 1

*Across the room, Danny, the diligent studio assistant, was doing his due diligence, keeping the earphones nestled in the recording booth. His aim was to ensure an optimized and meticulous environment for recording, a testament to his commitment to their shared mission. They were all aware of the arrangement − the notebook on the producer's desk, the earphones in the recording booth. Their shared consciousness of these items only intensified the anticipation; they were eager to turn the contents of a weathered notebook into a world−class album.*

*Ricky, with his weathered notebook of potent lyrics in hand, gently places it onto the piano. An air of creativity and anticipation lingers in the room, everyone aware that this was the first instrumental step in the creation of their masterpiece. In sync with the palpable creative energy, Ricky was engrossed in perfecting the*

*rhythm of his song, preparing himself for an intense day ahead. Not too far away, Emma was sincerely engrossed in her role of musically steering the session. She was focussed on Ricky's progress, her eyes constantly monitoring him and her mind alive with ideas to enhance the music.*

*Meanwhile, Danny was diligently covering every corner of the studio. He was making his rounds, ensuring that the studio was prim and proper for Ricky's crucial session. As part of his tasks, he passed by Ricky several times, always careful not to interrupt the artist's flow.*

*Emma, engrossed in her thoughts, deftly moves the earphones to the producer's desk. She is preparing to tweak the sound settings, pre−empting Ricky's need for perfect audio in his performance. Diverting from his rounds, Danny found himself in the midst of a stirring conversation with a visiting sound engineer. Knowledge flowed between them, illuminating the studio's atmosphere, the engineer's insight bringing a new perspective into Danny's role. Ricky, ensconced in his own world, was in deep discussion with the blank page before him. The daunting silence of the empty studio buzzed with his focus, as he honed his lyrics to perfection in a space separate from the producer's. The visitor, oblivious to the careful choreography of the studio session, stood blocking Danny's general overview of the studio space.*

*Delicately lifting Ricky's notebook, Danny orchestrates its move to the producer's desk. At the desk, he glimpses a pair of earphones indirectly drawing his attention amidst his routine of tidying up. Emma, from the isolated interior of a sound−proofed booth, lent her ears diligently to already recorded tracks, pouring over them for any room for improvement. Being lost in the music was her way of paying homage to her craft − an unspoken ritual she followed each time she embarked on a music production journey. The entirety of her focus was consumed by the musical notes and rhythm filtering through the studio speakers.*

*Concurrently, Ricky was absorbed in the act of playing his guitar. His fingers navigated deftly over the strings , lost in an intimate dance with the instrument. As he played, the melodic strums reverberated throughout the studio, filling it with an infectious pulse that hinted at the birth of yet another musical masterpiece. Despite the flurry of activity around him, Ricky was lost in a world of his own, operating on a singular vision of delivering his best performance.*

*In the meantime, Danny was continuing his cautious management of the studio, ensuring that everything fell into place for the optimum recording session. His watchful eyes were scanning every corner, taking stock of the minor details that could impact the session. However, the design of the studio didn't allow for an unrestricted view into all the corners. The sound booth, where Emma was engrossed in her work, was out of his visual range. The seclusion provided by the booth, although crucial for immersive work, also acted as a barrier for Danny's comprehensive vigilance.*

*As the day progressed, the studio was entwined in a concerted symphony of dedication and workmanship, the trio, each engrossed in their pursuit, working together to create the best version of Ricky's impending album. As the final note of the day rang through the studio, each person revelled in the satisfaction of another day done right, another step closer towards the realization of Ricky's artistic vision.*

*Within the dynamic dance of the day's events, the relationships of the trio sang a compelling tune. Each individual played their crucial part in the creation of the impending masterpiece − Ricky with his raw talent, Emma with her passion for perfection, and Danny with his meticulous eye for detail. And as the lights faded on another day of creation, they could sense the beginning of an important chapter in their artistry, a silence collecting the scattered notes of the day, signing off on another critical step in the journey of Ricky's upcoming album.*

*Based on this story, we want to identify where someone believes that a certain object is at the end of the story. In order to do that, you need to read the story and keep track of where they think the object is at each point. When an object is moved, the person may observe its new location if they saw it move.*

*To see where an object ends up, they must be able to see the location that it moves to and not be too distracted by what they are doing. If they do not observe the object moving, then they will still believe it to be in the last location where they observed it.*

*Which location is the most likely place Danny would look to find the earphones given the story?*

*Pick one of the following choices:*
*1 − piano*
*2 − producer's desk*
*3 − recording booth*

*You must pick one option. Explain your reasoning step by step before you answer. Finally, the last thing you generate should be "ANSWER: (your answer here, including the choice number)"*

Listing 8: Object Placement Example 1 Reasoning Tree

*A worn, leather−bound notebook contains all of Ricky's song lyrics, a crucial piece for his upcoming studio session. | Deduced Root Conclusion*
*> opening scene | Deduced Fact*
*> > Danny sees the notebook at the producer's desk. | Fact From Story*
*> > Danny sees the earphones at the recording booth. | Fact From Story*
*> > Emma sees the notebook at the producer's desk. | Fact From Story*
*> > Emma sees the earphones at the recording booth. | Fact From Story*
*> > Ricky sees the notebook at the producer's desk. | Fact From Story*
*> > Ricky sees the earphones at the recording booth. | Fact From Story*
*> Ricky moves the notebook to the piano. Because, Ricky needs his lyrics close while he works on the piano to compose his melodies. | Deduced Fact*
*> > Danny saw the notebook move to the piano. | Deduced Fact*
*> > > Ricky was in his line of sight during this tidying session. | Fact From Story*
*> > > Danny was doing his usual rounds in the studio to tidy up. | Fact From Story*
*> > > Being in someone's line of sight means they can see what you are doing. | Commonsense Knowledge*
*> > Emma saw the notebook move to the piano. | Deduced Fact*
*> > > Emma's role as a producer involves overseeing the artist's creative process. | Fact From Story*
*> > > Emma was monitoring Ricky's progress with his current song. | Fact From Story*
*> > > Performers are typically observed by their producers during a creative process. | Commonsense Knowledge*
*> Emma moves the earphones to the producer's desk. Because, Emma decided to adjust some sound settings and needed the earphones at her desk for that. | Deduced Fact*
*> > Danny did not see the earphones move to the producer's desk. | Deduced Fact*
*> > > The visitor was standing in a spot blocking Danny's view. | Fact From Story*
*> > > Danny was engrossed in a conversation with a visiting sound engineer. | Fact From Story*
*> > > When someone is distracted by a conversation and their view is blocked, they can't perceive actions occurring beyond their line of sight. | Commonsense Knowledge*
*> > Ricky did not see the earphones move to the producer's desk. | Deduced Fact*
*> > > The lyric composing session took place in a different area than where Emma was. | Fact From Story*
*> > > Ricky was absorbed in a lyric composing session. | Fact From Story*
*> > > When someone is focused on a task in a different area, they are usually unable to observe actions happening outside of that area. | Commonsense Knowledge*
*> Danny moves the notebook to the producer's desk. Because, Danny is tidying up the studio, and he believes the notebook would be safer at the producer's desk. | Deduced Fact*
*> > Danny saw the earphones at the producer's desk when moving the notebook. | Fact From Story*
*> > Emma did not see the notebook move to the producer's desk. | Deduced Fact*
*> > > The sound booth has no visual access to other areas of the studio. | Fact From Story*
*> > > Emma was reviewing some audio recordings inside a sound−proofed booth. | Fact From Story*
*> > > If a person is inside a closed area without visual access to other parts of their environment, they can't see what actions transpire in those other parts. | Commonsense Knowledge*
*> > Ricky did not see the notebook move to the producer's desk. | Deduced Fact*
*> > > His eyes were closed as he focused on feeling the music. | Fact From Story*
*> > > Ricky was engrossed in playing guitar. | Fact From Story*
*> > > When someone's eyes are closed, they cannot see what is transpiring in their surroundings. | Commonsense Knowledge*

Listing 9: Object Placement Example 2

*Richard, ever the diligent pilot, keeps his eye on the horizon and his flight manual. He keeps it conveniently placed in the cockpit, within arm's reach. Lisa, with the same dedication to the job, ensures that the safety booklet is tucked away in storage for quick access. Tom, the copilot, is always ready to assist Richard, familiar with the careful locations of the flight manual and safety booklet. Their tireless commitment to safety and preparedness was evident; everyone was aware, ready, and knew exactly where the crucial objects were located.*

*With a disciplined stride, Richard carries the flight manual to his office. Placing it down, he feels a sense of satisfaction, knowing he can review and improve his protocol knowledge at his leisure. Despite the din of commotion around her, flight attendant, Lisa, was caught up in instructing a fresh recruit on the necessity of excellent beverage service, ensuring that passenger comfort was meticulously addressed. In tandem with this,*

*pilot Richard left the vicinity, clutching something tightly as he intrepidly ventured forth. With a show of respect for his partner's goal of constant preparation, Tom, the reliable copilot was closely following Richard, heading in the same direction. All actions undeniably affirmed their unwavering commitment to safety, readiness, and flawless execution of cabin operations.*

*Slipping the flight manual under his arm, Tom headed straight toward the cockpit. His determined footsteps echoed his intent − another successful and incident−free flight. Whilst Richard found himself deeply engrossed in a task elsewhere, Lisa was indulging a passenger in pleasant banter, discussing their travel experiences. The hums of the conversation did little to fill the vast distance that separated Lisa and the engaged passenger from Tom and Richard. Lisa's laughter, dancing on the edge of the lively chatter within the aircraft, signaled her absorption in the conversation.*

*Simultaneously, Tom navigated the plane, making his move amid the quiet of lesser trodden areas of the aircraft. His path, charted away from the watchful gaze of Richard, led him back to the heart of operation − the cockpit.*

*Unbroken strides took Lisa towards the passengers seating area, a bundle of safety booklets firmly clutched against her chest. The leak of anticipation curled up around her lips as she began resupplying each seat, ready to welcome new passengers onboard. At the same time, Lisa, with her trademark charm, was diligently restocking the passenger seating area. Her hands swiftly moved in rhythm, ensuring that all was in order and ready for the hopeful passengers about to embark on their journey. Meanwhile, Richard, consistent with his role as the meticulous pilot, was thoroughly engrossed in the pre−flight checks located in another section of the plane. Despite not being in the same vicinity, Lisa and Richard's dedication to duty created a seamless link between the front and back of the aircraft.*

*Elsewhere, Tom, the faithful copilot, was discussing painstaking flight procedures with Richard. Their commitment to precise execution was evident in the quiet confidence that reverberated along with their diligent pace. Their work was choreographed like an unobserved ballet, an underpinning rhythm of safety and reliability in the background. As the trio ventured forth in their tasks, an unseen thread of unwavering readiness connected them, even with the distance that separated them physically. Their concentrated efforts in different sectors of the plane echoed a well−tuned rhythm of safety that reverberated throughout. Together, their individual tasks interwove to create a strong fabric of confidence, preparing the plane and its occupants for the journey ahead.*

*In conclusion, the meticulously choreographed routine of Richard, Lisa, and Tom painted a picture of steadfast dedication and commitment. Their collective endeavor towards precision and safety lays the foundation for a journey where safety and comfort were harmoniously entwined. Despite their varying roles or positions within the aircraft, the trio's dedication is a testament to the unwavering commitment to air travel's highest standards.*

*Based on this story, we want to identify where someone believes that a certain object is at the end of the story. In order to do that, you need to read the story and keep track of where they think the object is at each point. When an object is moved, the person may observe its new location if they saw it move.*

*To see where an object ends up, they must be able to see the location that it moves to and not be too distracted by what they are doing. If they do not observe the object moving, then they will still believe it to be in the last location where they observed it.*

*Which location is the most likely place Lisa would look to find the flight manual given the story?*

*Pick one of the following choices:*
*1 − cockpit*
*2 − office*
*3 − passenger seating area*
*4 − storage*

*You must pick one option. Explain your reasoning step by step before you answer. Finally, the last thing you generate should be "ANSWER: (your answer here, including the choice number)"*

Listing 10: Object Placement Example 2 Reasoning Tree

*An airline pilot, Richard, has his flight manual with him all the time; flying without it is against safety protocols. | Deduced Root Conclusion*

> *opening scene | Deduced Fact*
> > *Lisa sees the flight manual at the cockpit. | Fact From Story*
> > *Lisa sees the safety booklet at the storage. | Fact From Story*
> > *Richard sees the flight manual at the cockpit. | Fact From Story*
> > *Richard sees the safety booklet at the storage. | Fact From Story*
> > *Tom sees the flight manual at the cockpit. | Fact From Story*
> > *Tom sees the safety booklet at the storage. | Fact From Story*
> *Richard moves the flight manual to the office. Because, After finishing his shift, Richard took the manual with him to his office to review some protocols. | Deduced Fact*
> > *Lisa did not see the flight manual move to the office. | Deduced Fact*
> > > *Richard left the area while Lisa was engrossed in her teachings. | Fact From Story*
> > > *Lisa was instructing a new flight attendant about beverage service. | Fact From Story*
> > > *When someone is concentrated on a task, they are often unaware of the surrounding activities. | Commonsense Knowledge*
> > *Tom saw the flight manual move to the office. | Deduced Fact*
> > > *Richard was carrying something when they were walking. | Fact From Story*
> > > *Tom was walking in the same direction as Richard. | Fact From Story*
> > > *When you walk in the same direction as another person, you are usually able to see what they are carrying. | Commonsense Knowledge*
> *Tom moves the flight manual to the cockpit. Because, As part of his role, Tom made sure the manual was back in the cockpit before their next flight. | Deduced Fact*
> > *Lisa did not see the flight manual move to the cockpit. | Deduced Fact*
> > > *Lisa and the passenger were conversing at a considerable distance from Tom and Richard. | Fact From Story*
> > > *Lisa was absorbed in a conversation with a passenger about their travel experience. | Fact From Story*
> > > *Typically, you do not notice activity outside of your immediate conversation when you are fully immersed in it. | Commonsense Knowledge*
> > *Richard did not see the flight manual move to the cockpit. | Deduced Fact*
> > > *Tom moved while Richard was not in a position to observe him. | Fact From Story*
> > > *Richard was in a different area performing an essential task. | Fact From Story*
> > > *If someone is not present in the same location, they cannot witness the actions taking place there. | Commonsense Knowledge*
> *Lisa moves the safety booklet to the passenger seating area. Because, Lisa was restocking the safety booklets in the passenger seating area for the incoming passengers. | Deduced Fact*
> > *Richard did not see the safety booklet move to the passenger seating area. | Deduced Fact*
> > > *Richard was in another section of the airplane working on the pre−flight dossier. | Fact From Story*
> > > *Lisa was restocking when Richard was busy with document checks. | Fact From Story*
> > > *Busy individuals, especially when focused on a separate area and task, are unlikely to spot any movements outside their area. | Commonsense Knowledge*
> > *Tom did not see the safety booklet move to the passenger seating area. | Deduced Fact*
> > > *Lisa and Tom were not in the same location at the time. | Fact From Story*
> > > *Tom was engaged in a review of procedures with Richard. | Fact From Story*
> > > *If two people are not in the same place at the same time, they cannot witness each other's activities. | Commonsense Knowledge*

### F.3    TEAM ALLOCATION

#### Listing 11: Team Allocation Example 1

*Amidst the vibrant chaos of the Redwood Zoo, nestled in the heart of the city's sprawling jungle, the task of assigning roles was a crucial cog in the machinery of its operation. As the manager, the responsibility of allocating Olivia, Alex, and Mia to the positions of Animal Caretaker and Exhibit Cleaner presented an intriguing conundrum. Each individual, with their distinct personalities and skill sets, added a layer of complexity to this assignment puzzle.*

*Let's begin with Alex, the tall lad with bright eyes, whose history with the mighty beast of the animal kingdom, lacked a certain comfort. The lad, known to express an almost innate unease around animals larger than him, fell short of the prerequisites for an Animal Caretaker. His comfort zone extended to the four−legged companions in our homes, a sentiment I withheld from the petting zoo section of our park. Yet, his association and collaboration with Mia had seen quite the successes in their high school club's fundraising initiatives.*

*However, his relationship with the gentle Olivia was not as seamless. Alex often mentioned feeling ostracized due to Olivia's tendency to maintain her distance. This seemingly innocent avoidance stirred disquiet within*

*our hushed ranks. And all this, stemming from a disagreement rooted in their previous shared workplaces. Unresolved perhaps, but a factor nonetheless.*

*Then, there was Mia, the determined bright spark, whose affinity for cleanliness would often bemuse us. She would spend her spare time in her immaculate home cleaning and reorganizing, while her enthusiasm for a spotless Exhibit could not be underestimated. However, her overly thorough methods would invariably result in clashes with Olivia, who criticized her for crossing some form of unspoken boundary.*

*Mia too had her phobias, the gravelly roars of the zoo's majestic lion had once left her shaken and worried. Loud noises had a similar effect leaving her in a state of nervous terror, much like that of the petite animals held within our barriers. Yet, she was all smiles and peasant conversation around Alex during lunch breaks, sharing a sense of humor that lightened the mood of our everyday grind.*

*Finally, subdued Olivia, a soul strangled with allergies, and a deep−seated fear for wild animals. An incident with a chimp in her past wove tales of nightmarish betrayal, enough to send her away from the animal exhibits during her zoo visits. Potent elements of dust and pollen resulted in uncontrolled sneezing fits, a remainder from her days at the school as a custodian, responsible for the cleanliness and maintenance.*

*Three souls; Animals to be cared for, Exhibits to be cleaned. Assigning them was always going to be an enigma for anyone navigating the zoological labyrinth. Love for animals, discomfort, alliances, conflicts; each factor extraordinarily crucial in shaping not just the overall productivity but also the personal growth of each of these individuals at the Redwood Zoo.*

*Given the story, how would you uniquely allocate each person to make sure both tasks are accomplished efficiently?*

*Pick one of the following choices:*
*1 − Animal Caretaker: Alex, Exhibit Cleaner: Mia and Olivia*
*2 − Animal Caretaker: Olivia, Exhibit Cleaner: Alex and Mia*
*3 − Animal Caretaker: Mia, Exhibit Cleaner: Alex and Olivia*

*You must pick one option. The story should allow you to determine how good each person is at a skill. Roughly, each person is either great, acceptable, or bad at a task. We want to find an optimal assignment of people to tasks that uses their skills as well as possible. In addition, one task will have to have two people assigned to it. The effectiveness of their teamwork (great team, acceptable team, or bad team) also impacts the overall quality of the assignment.*

*When two people need to work on a task and one is bad at it, they don't necessarily benefit from the other person being good, unless they work well together.*

*With different strengths, weaknesses, and interpersonal dynamics at play, you should allocate your team to find the single assignment to ensure that the tasks overall are completed as effectively as possible.*

*Explain your reasoning step by step before you answer. Finally, the last thing you generate should be " ANSWER: (your answer here, including the choice number)"*

Listing 12: Team Allocation Example 1 Reasoning Tree

*A zoo requires caretakers for different animals and keepers for cleaning the exhibits. | Deduced Root Conclusion*
*> Olivia is bad at exhibit cleaner. Because we find out in the story that, | Deduced Fact*
*> > Olivia experienced sneezing fits and allergies during her custodial job at a school where cleaning was part of her task. | Fact From Story*
*> > Olivia mentioned her allergy to dust and pollen. | Fact From Story*
*> > Someone who is allergic to dust and pollen would have a hard time performing tasks like cleaning an exhibit, which often involves dusting and exposure to pollen. | Commonsense Knowledge*
*> Olivia is bad at animal caretaker. Because we find out in the story that, | Deduced Fact*
*> > Olivia often avoids the animal exhibits during her visits to the zoo. | Fact From Story*
*> > Olivia has admitted her fear of wild animals due to a past traumatic experience with a chimp. | Fact From Story*
*> > If someone admits fear of wild animals and purposely avoids them, they're unlikely to do a good job as an animal caretaker, which requires close and frequent interactions with animals. | Commonsense Knowledge*
*> Mia is bad at animal caretaker. Because we find out in the story that, | Deduced Fact*
*> > Mia mentioned a discomforting encounter with a roaring lion that left her shaken. | Fact From Story*
*> > Mia gets nervous and excessively worried at loud noises. | Fact From Story*

> > If someone gets unnerved by loud animal noises and had a bad experience with them in the past, they're unlikely to be a good animal caretaker as zoos usually have animals that can be loud. | Commonsense Knowledge
> Mia is good at exhibit cleaner. Because we find out in the story that, | Deduced Fact
> > Mia showed enthusiasm when she found out about the cleaning tasks at the zoo, expressing her belief that the exhibitors need to mirror the animals' natural habitats as closely as possible. | Fact From Story
> > Mia insists on a spotless living environment, often spending free time in her own home cleaning and organizing. | Fact From Story
> > Somebody who prioritizes cleanliness in their own life is likely to be meticulous in cleaning tasks at work, especially if they express enthusiasm for the task. | Commonsense Knowledge
> Alex is okay at exhibit cleaner. Because we find out in the story that, | Deduced Fact
> > Alex has shown mild interest in keeping his surroundings neat, but he doesn't go out of his way to tidy up . | Fact From Story
> > Alex sometimes voluntarily helped with exhibit cleaning when he was a volunteer at a cat shelter. | Fact From Story
> > If someone voluntarily cleans up in past experiences and has a moderate interest in it, he or she could probably do okay in a cleaning job, even if they do not excel. | Commonsense Knowledge
> Alex is bad at animal caretaker. Because we find out in the story that, | Deduced Fact
> > Alex expressed in the past no desire to pursue furthering his knowledge of animals outside of pets. | Fact From Story
> > Alex admitted that he feels uncomfortable around animals larger than him. | Fact From Story
> > If someone is uncomfortable around large animals and has no interest in expanding his knowledge of animals, they probably won't be good at a job that involves taking care of a variety of animals, some of which can be large. | Commonsense Knowledge
> Alex and Mia work okay together. Because we find out in the story that, | Deduced Fact
> > At lunch breaks, Alex and Mia engage in friendly conversations and share a similar sense of humor. | Fact From Story
> > Alex and Mia used to cooperate well in the same high school club, often collaborating on fundraising initiatives. | Fact From Story
> > If two people have cooperated well in the past and have good social interactions, they are likely to work okay together. | Commonsense Knowledge
> Olivia and Alex work badly together. Because we find out in the story that, | Deduced Fact
> > Alex expressed his discomfort around Olivia, mentioning how her avoidance makes him feel ostracized. | Fact From Story
> > Olivia avoids Alex during their shared shifts because of an old workplace disagreement. | Fact From Story
> > If two coworkers actively avoid each other due to past conflicts, they likely can't work together effectively. | Commonsense Knowledge
> Olivia and Mia work badly together. Because we find out in the story that, | Deduced Fact
> > Mia finds Olivia too passive and non−confrontational, which results in stewing resentment and lack of open communication. | Fact From Story
> > Olivia has explicit disagreements with Mia's work habits, often criticizing her for cleaning the exhibits too thoroughly. | Fact From Story
> > If individuals have fundamental disagreements about work ethics and habits and lack of open communication, it is unlikely they will work well together. | Commonsense Knowledge

Listing 13: Team Allocation Example 2

As the overseer of the local Poetry Palace, I am privileged to know my poets and judges not just as employees, but also as friends. Today, we found ourselves in the throes of preparing for an upcoming poetry event. A challenging puzzle presented itself: the roles of recitation and scoring needed to be allocated among my dedicated trio: Rachel, David, and Lily.

Rachel, a spirited woman with a wide grin, had always been a passionate poet. However, her work habits could be called into question, according to David. She tended to be more laid back and unstructured, which David considered a flaw. Lily too, had tangled with Rachel in the past, when she had offered some critiques on Rachel's poetry − critiques that were not well−received, leading to a heated argument and a grudge that still lingered between them. Rachel's reaction reflected her struggle with accepting feedback from others. Her tendency to judge poetry personally over objectively, even letting her opinion of a poet color her scores, was also an issue.

David, on the other hand, was a connoisseur of the poetic word. He boasted a deep understanding and appreciation for a wide spectrum of poetry styles, which revealed itself when he shared comprehensive and incisive feedback with poets. Yet, David had flaws of his own. He was known for his sarcasm, a trait particularly hurtful to Lily due to remarks about her mild stutter. Tensions between them had escalated into a

*silent disagreement. Moreover, while David's knowledge of poetry was vast, his voice was not the musical instrument required for an engaging recitation. His monotone delivery and self−conscious fear of boring people made him shy away from recitations.*

*Lastly, there was Lily. Although her speech bore the unique quirk of a mild stutter which became emphasized when she was nervous or faced a large crowd, she was an ardent poetry enthusiast. At home, she had a routine of reading and analyzing poems, propelling her understanding of poetry. She had earned her stripes by taking literature classes in college, including a course dedicated entirely to poetry. However, David's sarcastic remarks about her stutter had marred her morale, and she was already apprehensive about performing in front of large crowds.*

*As I watched them, my thoughts spun with the complexity of their dynamic− the strengths and weaknesses of each individual, the silent feuds, and shared enthusiasm for poetry. Balancing it all was a tough job, but as the manager, the responsibility sat squarely on my shoulders. The event was quickly approaching and I had to decide who would recite and who would score− a decision that, I hoped, would inspire personal growth, heal strained relationships, and ultimately make the event a success.*

*Given the story, how would you uniquely allocate each person to make sure both tasks are accomplished efficiently?*

*Pick one of the following choices:*
*1 − Recitation: David, Scoring: Lily and Rachel*
*2 − Recitation: Lily, Scoring: David and Rachel*
*3 − Recitation: Rachel, Scoring: David and Lily*

*You must pick one option. The story should allow you to determine how good each person is at a skill. Roughly, each person is either great, acceptable, or bad at a task. We want to find an optimal assignment of people to tasks that uses their skills as well as possible. In addition, one task will have to have two people assigned to it. The effectiveness of their teamwork (great team, acceptable team, or bad team) also impacts the overall quality of the assignment.*

*When two people need to work on a task and one is bad at it, they don't necessarily benefit from the other person being good, unless they work well together.*

*With different strengths, weaknesses, and interpersonal dynamics at play, you should allocate your team to find the single assignment to ensure that the tasks overall are completed as effectively as possible.*

*Explain your reasoning step by step before you answer. Finally, the last thing you generate should be " ANSWER: (your answer here, including the choice number)"*

Listing 14: Team Allocation Example 2 Reasoning Tree

*A poetry event needs poets to recite and judges to score. | Deduced Root Conclusion*
*> Lily is okay at scoring. Because we find out in the story that, | Deduced Fact*
*> > Lily often reads poems at home and analyzes them in her free time. | Fact From Story*
*> > Lily took a few literature classes in college including Poetry and Literature 101. | Fact From Story*
*> > If a person has studied a similar subject and practices it often, they may have at least some competence in it. | Commonsense Knowledge*
*> Lily is bad at recitation. Because we find out in the story that, | Deduced Fact*
*> > Lily has a mild stutter that becomes more pronounced when she is nervous. | Fact From Story*
*> > Lily often gets stage fright when speaking in front of a large crowd. | Fact From Story*
*> > If a person gets nervous in front of a crowd and has a speech issue, like stuttering, they're likely to be bad at reciting poetry in front of an audience. | Commonsense Knowledge*
*> Rachel is bad at recitation. Because we find out in the story that, | Deduced Fact*
*> > When Rachel gets feedback, she struggles to take it to heart and improve. | Fact From Story*
*> > Rachel has been told before that her rhythm and pacing when reading poetry is off. | Fact From Story*
*> > If a person cannot maintain rhythm and pacing while reciting poetry, and they do not take feedback well, they will likely be poor at recitation. | Commonsense Knowledge*
*> Rachel and Lily work badly together. Because we find out in the story that, | Deduced Fact*
*> > Rachel holds a grudge towards Lily after this incident. | Fact From Story*
*> > Lily gave Rachel critique on her poetry once which resulted in an argument. | Fact From Story*
*> > If two people have held a grudge over a work−related incident, they'll likely struggle to work effectively together. | Commonsense Knowledge*
*> Rachel and David work badly together. Because we find out in the story that, | Deduced Fact*
*> > David finds Rachel's work habits to be too laid back and unstructured. | Fact From Story*

> > *Rachel feels that David doesn't respect her opinions and ideas. | Fact From Story*
> > *When two people have differing work habits and do not respect each other's methods, they typically don't work well together. | Commonsense Knowledge*
> *David and Lily work badly together. Because we find out in the story that, | Deduced Fact*
> > *Lily has overheard David making these remarks and it has led to a silent disagreement between them. | Fact From Story*
> > *David makes sarcastic remarks about Lily's stutter which hurts her feelings. | Fact From Story*
> > *If workers have unresolved disagreements or conflicts, they usually have trouble cooperating and working well together. | Commonsense Knowledge*
> *David is bad at recitation. Because we find out in the story that, | Deduced Fact*
> > *David never practiced recitation fearing he would bore people. | Fact From Story*
> > *David has a monotone voice that lacks the inflection needed for engaging poetry recitation. | Fact From Story*
> > *A monotone voice and lack of practice in recitation can make a person bad at reciting poetry. | Commonsense Knowledge*
> *David is good at scoring. Because we find out in the story that, | Deduced Fact*
> > *David has a deep understanding and appreciation for a variety of poetry styles. | Fact From Story*
> > *David shares comprehensive and constructive feedback with poets. | Fact From Story*
> > *If someone is well−versed in poetry and has a knack for providing constructive criticism, they would likely be a good judge at a poetry event. | Commonsense Knowledge*
> *Rachel is bad at scoring. Because we find out in the story that, | Deduced Fact*
> > *Rachel admitted to having difficulty separating her personal biases from her judgement of the poetry. | Fact From Story*
> > *Rachel often makes scores based on her personal feelings about the poet, not the poem's content or structure. | Fact From Story*
> > *If a person admits they struggle to judge work objectively due to their feelings towards the individuals involved, they will likely be bad at judging a competition objectively. | Commonsense Knowledge*

## G  BUILDING THE DOMAINS

### G.1  TREE CONSTRUCTION ALGORITHM

---
**Algorithm 1** The recursive reasoning tree expansion algorithm.

---
**Input:** The scenario $\mathbf{s}$. The current reasoning tree $T$. **Output:** Tree is expanded in-place.
**function** CREATEDEDUCTION($T$, node, max_depth, depth)
    **for** child **in** node.children **do**
        child.children $\leftarrow$ PromptLM($T_1, \ldots, T_m \mid \mathbf{s}$, child, $G$)
        **if** VALIDATE(child.children) **and** depth $<$ max_depth **then**
            CREATEDEDUCTION($T$, child, max_depth, depth $+ 1$)

---

### G.2  MURDER MYSTERY

Tree construction prompt (per deduction)

Listing 15: Murder Mystery Deduction Prompt for the Tree Completion stage

*Your task is to generate a logic tree for a story, as shown in the example. In this tree, each fact should be deduced from its immediate children. If a deduced fact already has a name, do not overwrite it.*

*Type of story:*

*We are creating a murder mystery. A murder mystery needs to have a complex web of evidence that leads to a means, motive, and opportunity for a suspect, which will make them a likely murderer. When writing a murder mystery, the story should take the point of view of the detective. Evidence should be collected through investigation, including things like interrogation, hearing conversations, reading past criminal records, looking at mail or trash, and other normal modes of detecting evidence.*

*1. Each fact in the tree must follow via logical deduction from its children.*
*2. All Fact From Story nodes and the Commonsense Knowledge node must be relevant to the deduction they yield.*
*3. Each root fact is labeled with a source (Fact from Story or Commonsense Knowledge).*

*4. A Fact From Story should be a statement about a character, place, or object in the story.*
*5. Commonsense Knowledge should be a fact that most people will know and agree with. It should not explicitly reference any characters in the story.*
*6. Commonsense Knowledge should be used as a deduction rule that when the sibling facts are applied to it they yield the parent deduced fact.*
*7. The tree you generate must match the structure of the tree I give you.*

*A suspect has a means when they have access to the murder weapon.*
*A suspect has a motive when the suspect has reason to kill the victim.*
*A suspect has an opportunity when they were at the crime scene.*

*Here's an example.*

*Scenario:*

*Victim: Victoria*
*Crime scene: Home*
*Murder weapon: Gun*
*Suspect: James*
*Suspect's role in story: Brother*
*Suspect's motive: Financial gain*

*Current Tree:*
*James is a murderer. | Deduced Root Conclusion*
*> James has a means | Deduced Fact*
*> > James has practiced shooting guns. | Fact From Story*
*> > James owns guns | Fact From Story*
*> > If you both own and practice using guns then you have the ability to murder someone. | Commonsense Knowledge*
*> James has a motive. | Deduced Fact*
*> > James was violently desperate for cash. | Fact From Story*
*> > James was violently desperate for Victoria's cash | Fact From Story*
*> > When someone is violently desperate they may go to extreme measures to accomplish a task, including murderer. | Commonsense Knowledge*
*> James has a opportunity. | Fact From Story*

*Entailment Step to complete:*
*James has a opportunity.*
*> Fact From Story*
*> Commonsense Knowledge*

*Output:*
*James has a opportunity.*
*> James has access to Victoria's house. | Fact From Story*
*> Having access to someones house gives you the opportunity to murder them. | Commonsense Knowledge*
*Here is another example.*

*Scenario:*

*Story Information:*
*Victim: Harry*
*Crime scene: Racetrack*
*Murder weapon: Shovel*
*Suspect: Claire*
*Suspect's role in story: Running buddy*
*Suspects motive: To prevent someone else harm*

*Current Tree:*
*Claire is a murderer. | Deduced Root Conclusion*
*> Claire has a means. | Deduced Fact*
*> > Claire is a farmer | Fact From Story*
*> > Farmers typically use gardening tools like shovels in their work. | Commonsense Knowledge*

> *Claire has a motive.* | *Fact From Story*
> *Claire has an opportunity* | *Fact From Story*

*Entailment Step to complete:*
*Claire has a motive.*
> *Fact From Story*
> *Fact From Story*
> *Commonsense Knowledge*

*Output:*
*Claire has a motive.*
> *Claire loves Brian deeply.* | *Fact From Story*
> *Harry threatened Brian.* | *Fact From Story*
> *Deep and passionate love can push people to do extreme things like murder when that loved one is threatened.* | *Commonsense Knowledge*
*Here is another example.*

*Scenario:*

*Victim: Jared*
*Crime scene: Public park bench*
*Murder weapon: Heroin overdose*
*Suspect: Jose*
*Suspect's role in story: Drug user*
*Suspects motive: Public humiliation*

*Current Tree:*
*Jose is a murderer.* | *Deduced Root Conclusion*
> *Jose has a means.* | *Fact From Story*
> *Jose has a motive* | *Fact From Story*
> *Jose has an opportunity.* | *Fact From Story*

*Entailment Step to complete:*
*Jose has a means.*
> *Fact From Story*
> *Fact From Story*
> *Commonsense Knowledge*

*Output:*
*Jose has a means.*
> *Jose has access to heroin.* | *Fact From Story*
> *Jose knows how much heroin is needed for an overdose.* | *Fact From Story*
> *Having access to heroin and knowing how much heroin is required to overdose implies you could have intentionally given the victim a dose of lethal heroin providing a means for murder.* | *Commonsense Knowledge*

*Your Turn.*

*Scenario: Victim: Tessa*
*Crime Scene: kitchen*
*Murder Weapon: poisonous gas*
*Suspect: Penelope*
*Role in story: Tarot Reader*
*The suspect's motive: To protect a secret*

*Current Tree:*
*Penelope is the murderer.* | *Deduced Root Conclusion*
> *Penelope has a means.* | *Deduced Fact*
> *Penelope has an opportunity.* | *Deduced Fact*
> *Penelope has a motive.* | *Deduced Fact*

*Entailment Step to Complete:*
*Penelope is the murderer.*
> *Penelope has a means. Because,*

```
> > Fact From Story
> > Fact From Story
> > Commonsense Knowledge

Output:
```

Validator prompt for GPT-3 and GPT-4

Listing 16: Murder Mystery Deduction Validation Prompt for the Tree Completion stage

*We are writing a murder mystery and to do so we are creating a narrative guide of evidence. We are proving a means (access to the murder weapon) right now. Does this deduction in any way prove or help to prove a motive (reason to kill) or an opportunity (access to the crime scene) given the description of the mystery below.*

*Victim: Tessa*
*Crime Scene: kitchen*
*Murder Weapon: poisonous gas*
*Suspect: Penelope*
*Role in story: Tarot Reader*
*The suspect's motive: To protect a secret*

*The Deduction:*
*Penelope has a means.*
*> Penelope was witnessed purchasing an unusual quantity of chemicals the day prior. | Fact From Story*
*> Records show Penelope studied chemistry and alchemy extensively. | Fact From Story*
*> One can use their knowledge of chemistry and alchemy to create a poisonous gas if they possess the necessary chemicals. | Commonsense Knowledge*

*Write your answer in the following format:*
*ANSWER: (yes/no)*

Intro prompt (introduce victim and suspects)

Listing 17: Murder Mystery Intro Creation for Story Generation

*Create an intro for this murder mystery. It should only be 1 or 2 sentences. Only write the intro nothing else.*

*Scenario:*
*Tessa was killed with a poisonous gas at a kitchen. Detective Winston is on the case, interviewing suspects. The suspects are Penelope, Melody.*

*Output:*

Chapter prompts

Listing 18: Murder Mystery Chapter Prompt for the Story Generation stage

*We are creating a murder mystery. A murder mystery needs to have a complex web of evidence for a suspect. When writing a murder mystery, the story should take the point of view of the detective. Evidence should be collected through investigation, including things like interrogation, hearing conversations, reading past criminal records, looking at mail or trash, and other normal modes of detecting evidence.*

*We will give you a list of facts to use when writing your story. You must include all the facts in the list. Never state deduced facts or conclusions. The story should stick to the fact list pretty closely.*

*You will write a chapter for the murder mystery. This should not introduce the murder or victim, but strictly follow a detective slowly interviewing and detecting the suspect. Use the list below as a guide.*

*Rules:*
*1. Only write the contents of the chapter, do not write a title or number it. This text must be easily placed in a larger story.*
*2. Never say a suspect has a means*
*3. Never say a suspect has a motive*
*4. Never say a suspect has an opportunity*

*5. Never hint at a suspect having or not having a means, motive, or opportunity. 3. Never hint at or explicitly say a deduced fact.*
*6. Never say a suspect is a murderer. It's a puzzle that the reader is supposed to guess and deduce!*
*7. Write the story from the perspective of a detective uncovering clues through various methods that normal detectives use (interrogation, notes, stake−outs, etc.)*
*8. Never make mental notes, point out suspicious facts, nor make connections. Let the reader do this.*

*Include as many sentences as needed to write each fact from the list of facts. Also include up to 10 sentences of dialogue.*

*Here is an example:*

*Suspect and crime information*
*Victim: Dora*
*Crime Scene: Remote forest*
*Murder Weapon: Knife*
*Suspect: Willard*
*Role in story: Grounds keeper*
*The suspect's motive: Religious sacrifice*

*You are detective Winston.*
*Facts you must include:*
*− A witness saw someone with a spaghetti face and green ears*
*− Willard is a groundskeeper for a local school*
*− Willard also provides services to nearby residents like house painting, lawn care, etc.*
*− Willard painted a nearby home green.*
*− Willard was in a horrible fire when he was a child.*
*− Willard's family has been in the local area for multiple generations*
*− Long ago, the local area had religious extremists around them, all participating in occult activities.*
*− Willard and his immediate family were all handymen of some sort.*
*− Willard believes in respecting his elders and ancestral history.*
*− Dora had written about joining a new church nearby.*
*− A friend of Dora mentioned worrying about Dora's involvement with a new cult−like group of friends.*

*Output:*

*Winston took a long drag from his cigarette while reviewing the crime scene photos. He was hardened to common grotesqueries of his line of work, but the murder of Dora sparked something in him... a calling into the void.*

*The only hard evidence he had in this case was an eyewitness of a monster... a spaghetti face, green−eared monster. That, and the fact that Dora had been exploring some new church... or perhaps a cult, depending on who you asked. Cults had a history of leaving a bad taste in people's mouths here... having some very dark pasts in the local area.*

*Winston put his cigarette out, placed the photos down, and set out for his following suspect interview with Willard.*

*The smell of fresh−cut grass washed over Winston as the local groundskeeper of the local elementary shut off his mower.*

*"Howdy, Mister," − Willard said with a rich southern twang.*

*"Howdy..." Winston said carefully, trying to match the southern pleasantries he wasn't all too familiar with.*

*"You, Willard?" Winston inquired.*

*"Well, sure I am!" He chuckled, "Been him for a while now, I suppose."*

*"You do a lot of work for the school here?"*

*"Heh, yes, sir, I do. My family and I have been helping out the local schools and locals with standard chores ." noticing Winston motioning for him to continue, Willard explained further, "You know, like painting houses, cutting people's lawn... well heck, just a couple days ago I painted that green house over yonder." Willard said, pointing out across the schoolyard.*

*Winston couldn't help but notice that Willard had a severe burn scar across the bottom half of his face, which had previously been hidden by sweat and grass shavings.*

*∗Ring∗ ∗Ring∗ − Winston was getting called away, back to the office, so he had to make a quick excuse to get away.*

*"You know the way back to Highway 45?" Winston asked. "Sure," Willard replied with a grin, "You take two lefts here, then about 12 miles down, you'll hit it."*

*Winston smiled, "You sure do know your way around here," he said. "Hah, well, sir, my families have been here for... for a long, long time. I take great pride in my ancestry and family history here..."*

*With that, Winston got in his car and left.*

*−−−*

*Your turn.*

*Suspect and crime information*
*Victim: Tessa*
*Crime Scene: kitchen*
*Murder Weapon: poisonous gas*
*Suspect: Penelope*
*Role in story: Tarot Reader*
*The suspect's motive: To protect a secret*

*You are detective Winston.*

*Facts you must include:*
*− Tessa found incriminating letters connecting Penelope to past crimes*
*− Penelope was seen leaving Tessa's place immediately before the time of the murder.*
*− Records show Penelope studied chemistry and alchemy extensively.*
*− Penelope was seen arriving at Tessa's house before the murder took place.*
*− Penelope was witnessed purchasing an unusual quantity of chemicals the day prior.*
*− Penelope reacted strongly with fear when Tessa confronted her about the letters*
*− Penelope was seen reading books about poisonous gases*
*− Penelope was alone in Tessa's kitchen during the tarot reading session.*
*− No one saw Penelope leave Tessa's place before the discovery of murder.*
*− Penelope was cryptically speaking about the perfect crime that leaves no traces*
*− Tessa had confronted Penelope about those past crimes.*
*− Tessa knew about Penelope's past crimes.*

*Output:*

During the Tree Template Construction stage, we sample the victim's name, the murder weapon, and the crime scene. For every suspect, we also sample the suspect's name, motive, a suspicious fact, and their role in the story (usually a job or relation to the victim). We can use the sampled information from the Tree Template Construction stage to create the gold fact set $F$.

Once we have all these sample, we choose one suspect to be the murderer; this suspect will have the facts: "[suspect] has a means", "suspect has a motive", and "suspect has an opportunity" to the gold set of facts $F$. All other suspects have two of the "MMO" facts (the ones listed above); the third fact for innocent suspects is suspicious facts that do not contribute to solving the murder (for example, "[suspect] enjoys reading murder mystery novels.")

Once each suspect has three facts in the set $F$, we can begin the tree construction process. However, in the prompt, we ensure that the details of the suspect (like their motive) are included so that when we build a tree for the fact "[suspect] has a motive," the reasons for having a motive are still linked to the sampled motive from the Tree Template Construction stage.

Because many of the trees are already made for each suspect, we will create a "contrastive example" for each mystery where we simply set the murderer to another suspect. We can then reuse most of the trees from the previous mystery and only have to prove two new ones and then generate the story.

We use validators during the reasoning tree construction process ensure that the deductions being generated are correct and meaningful. When proving a means, we check the generation from GPT-4 for any mention of "motive" or "opportunity" including key items in the story that might relate to them (like the crime scene). If these are mentioned, we re-prompt GPT-4 for another deduction. We repeat this process for motive and opportunity.

To further increase the validity of the murder mysteries generated, we also implemented a story validator during the narrative generation. The story validator prompts GPT-4 to check the entailment of each scenario-specific fact $S(T)$ given the story. If any fact is not entailed by the story, we prompt GPT-4 to rewrite the story and ensure that fact is entailed.

**Suspicious Facts** The suspicious facts were useful to control for a length bias where the murderer was described in longer and more explicit terms than the non-murderer. We also considered negations of facts like "*Sophia has no motive to kill Emily*", but these often led to easy mysteries where GPT-4 would explicitly rule out a suspect. Suspicious facts blend in with the narrative, provide balance, and do not modify the ground-truth reasoning.

## G.3    OBJECT PLACEMENTS

Listing 19: OP scenario creation for generating items, people, roles, etc. in the Domain Injection stage

*Use the given scenario description to create a list of objects, people, and locations as well as where some items are currently located and the correct items that people want. The idea is that someone really needs something in this scenario.*

*Include one item and location that are relevant to the scene but aren't necessarily relevant to the scenario. For example, coffee beans might be relevant to a coffee shop, but not so relevant for the scenario of someone looking for the milk.*

*In the correct assignment and outcome, the person who gets an item in the If clause must be different than the person in the Then clause. See the example.*

*Rules:*
*1) You must use names of real people, describe their role in the story after the comma. For example, "Luis, customer" or "Adam, lawyer" etc. always use real names like "Berry" or "Cynthia".*
*2) You must use tangible small items, do not use ideas "a performance" for example, do not use large items like "a tv". Instead use small, easily moved items like "Iphone", "Notebook", "Laptop" etc.*
*3) The locations you pick must be able to house the items you made. So if you said "Golf club" was an item, all locations must be able to fit a golf club in them, you would not say "coat pocket" for example.*

*Here's an example*

*Description: Sarah is making coffee at her work, she wants to use almond milk for her customer.*

*Output:*

*Items: almond milk; regular milk; almond milk coffee; regular milk coffee; coffee beans*
*People: Sarah, a barista; Luis, a customer; John, a customer*
*Locations: Fridge; Shelf; Counter; Storage Closet*

*Located: Almond Milk −> Fridge*
*Located: Regular Milk −> Counter*

*CORRECT −− format (If: someone −> (gets) Something. Then: Another person −> (gets) something they want.)*
*If: Sara −> Almond Milk*
*Then: Luis −> Almond Milk Coffee*

*Your turn!*

*Description: Mason, a magician, has his magic deck of cards which are essential for his performance.*

*Output:*

Listing 20: OP Prompt for generating plausible moves in the Domain Injection stage.

*You are going to continue our story that we have written by writing a short description of this event that will happen next. Only write about the move, do not add any additional information.*

*Never say "someone didn't see something" or infer someones ability to infer where something is. Never say " Unbeknownst" or anything like this!*
*Here is an example.*

*Only write one or two sentences. It should be a very short continuation.*

*Description: Timmy was angry at Bob for cheating his way into the job Timmy deserved! So he started throwing away Bobs possessions.*

*Character:*
*Name: Timmy*
*Role in story: A recent graduate who is sharing an apartment.*
*Motivation in story: Timmy is angry because he interviewed for a job that his roommate got, but only because he cheated.*

*Event:*
*− Timmy moves the car keys to the trash bin. Because, Timmy was angry with Bob and wanted to throw away his keys.*
*− Timmy saw the iphone at the trash bin when moving the car keys.*

*Output: With an angry thrust, the keys clanked against the tin trash bin. An unexpected ∗smack∗ followed though... curiosity overtaking his anger, Timmy looked in the trash and saw the iphone in there as well.*

*Here is another example.*

*Description: Carol had just moved into her new apartment, but, the previous tenant made a huge mess! The landlord wouldn't do anything, so it looks like she has to clean it all up herself.*

*Character:*
*Name: Carol*
*Role in story: Just moved into a new messy apartment.*
*Motivation in story: Carol wants to clean her new apartment that was left a mess by the previous tenant and has exactly no help from management.*

*Event:*
*− Carol moves the noodles to the pantry. Because, Carol was excited to have a clean apartment finally, and the noodles were the last step!*

*Output: Carol excitingly places the noodles back into the pantry. What was once thought of as a never ending onslaught of trash and random items finally concluded and the apartment was finally clean again!*

*Your turn.*

*Description: Ricky, with his weathered notebook in hand, was ready for a full day's studio session. His aim: perfect his upcoming album, for which his notebook was crucial. Emma, his producer and confidante, eager to bring Ricky's lyrics to life, shared Ricky's enthusiasm for the project. Meanwhile, Danny − the diligent studio assistant − was making sure all elements of the studio were meticulously managed for the recording session.*

*Character:*
*Name: Emma*
*Role in story: Music Producer*
*Motivation in story: Emma wants to support Ricky's creative process and record outstanding music to advance both Ricky's career and her own reputation as a successful music producer.*

*Event:*
*− Emma moves the earphones to the producer's desk. Because, Emma decided to adjust some sound settings and needed the earphones at her desk for that.*

*Output:*
*In the heart of the bustling studio, Ricky, Emma, and Danny readied themselves for a day of creating magic. Ricky, holding the helm as the gifted singer−songwriter, was poised for perfection, his precious notebook of lyrics awaiting its call to duty on the producer's desk. Emma, their dutiful and talented producer, was just as eager to breathe life into Ricky's lyrics. She was cognizant of the notebook's place at her desk, awaiting the melodies they would cultivate together.*

*Across the room, Danny, the diligent studio assistant, was doing his due diligence, keeping the earphones nestled in the recording booth. His aim was to ensure an optimized and meticulous environment for recording, a testament to his commitment to their shared mission. They were all aware of the arrangement − the notebook on the producer's desk, the earphones in the recording booth. Their shared consciousness of these items only intensified the anticipation; they were eager to turn the contents of a weathered notebook into a world−class album.*

*Ricky, with his weathered notebook of potent lyrics in hand, gently places it onto the piano. An air of creativity and anticipation lingers in the room, everyone aware that this was the first instrumental step in the creation of their masterpiece. In sync with the palpable creative energy, Ricky was engrossed in perfecting the rhythm of his song, preparing himself for an intense day ahead. Not too far away, Emma was sincerely engrossed in her role of musically steering the session. She was focussed on Ricky's progress, her eyes constantly monitoring him and her mind alive with ideas to enhance the music.*

*Meanwhile, Danny was diligently covering every corner of the studio. He was making his rounds, ensuring that the studio was prim and proper for Ricky's crucial session. As part of his tasks, he passed by Ricky several times, always careful not to interrupt the artist's flow.*

Listing 21: OP deduction prompt for the Entailment Tree Construction stage.

*We are creating a story where people are going to move a lot of items many times. The goal of the story is to be interesting and unique but also have a clear tracking of where objects are so we can quiz readers and language models later on about the world state and about who knows what.*

*To make this story, we've created a tree structure that outlines the narrative and updates to the world. Your job is to fill out the entailment trees that prove a person saw or did not see an event happen.*

*An entailment tree is a tree structure where intermediate nodes are entailed by their children. They create a natural language reasoning proof for some collection of facts.*

*To fill out this tree we need to complete an entailment. Completing an entailment is akin to filling out one subtree of the entailment tree. To fill in this step, you must follow the structure of the step.*

*Facts From Story are facts that will be explicitly stated when we write the story.*
*Commonsense Knowledge are facts that most people would agree are true and don't need to be explicitly said .*
*Complex Facts are facts that will be entailed by simpler facts from the story (they will be filled in later through a recursive call back to you!)*

*All facts for the step must combine to entail the root parent fact.*

*No facts may contradict the current structure tree.*

*Do not include facts about other people, focus on the facts for the person who is seeing or not seeing something move.*

*Always match the exact structure of the entailment step I give you. Give the same number of Facts From Story and Commonsense Knowledge facts. Give them in the same order as well.*

*Never explicitly say someone didn't see something or did see something. Your job is to provide facts that suggest this. For example, if May saw Greg move something, we might say "May was watching Greg while doing her chores", and that "by watching someone, you are seeing what they are doing". Notice we describe the physics of seeing something, but we don't outright say that someone saw something.*

*Never mention an item being moved or reuse an item. For example, if theres a fact like "Greg didn't see his iphone" and you are proving why "Joel didn't see the apple move", never reuse Greg's iphone in your facts.*

*Our program strictly controls where items are placed, we don't want you introducing item placements we haven't accounted for.*

*Each fact should be crucial to the deduction. Intentionally leave out details so that the other facts can account for them. If one fact is missing, the conclusion should not be entailed. Try not to reuse the same facts.*

*Always use the persons name instead of a pronoun like "He" or "She", if you know someones name, use the name.*

*Only perform one deduction at a time. Your deduction should match the "Entailment Step to Complete" template exactly so we can parse it later on.*

*Here's an example.*

*Scenario:*
*Paul and Alice are at a karaoke bar.*

*Current Tree:*
*Paul and Alice are at a karaoke bar. | Deduced Root Conclusion*
*> Opening Scene | Deduced Fact*
*> > Paul sees the microphone at the stage. | Fact From Story*
*> > Alice sees the microphone at the stage. | Fact From Story*
*> > Paul sees the beer at the bar. | Fact From Story*
*> > Alice sees the beer at the bar. | Fact From Story*
*> Paul moves the beer to the table. | Deduced Fact*
*> > Alice did not see the beer move to the table. | Deduced Fact*
*> > > Alice was facing away from the table. | Deduced Fact*
*> > > > Alice was talking to another patron. | Fact From Story*
*> > > > The other patron was facing the table. | Fact From Story*
*> > > > Usually people talk to each other while facing each other, so if one person is looking in one direction the other person is looking in the opposite direction. | Commonsense Knowledge*
*> > > If someone is facing away from something else, they cannot see things transpire near that something else. | Commonsense Knowledge*
*> Alice moves the microphone to the table | Deduced Fact*
*> > Alice saw the beer at the table when moving the microphone. | Fact From Story*
*> > Paul saw the microphone move to the table. | Deduced Fact*
*> > > Paul was drinking the beer at the table. | Commonsense Knowledge*
*> > > When something happens where a person is at, they usually see things that are happening there. | Fact From Story*
*> Alice moves the beer to the trash can. | Deduced Fact*
*> > Paul did not see the beer move to the trash can. | Fact From Story*

*Entailment Step to complete:*
*Paul and Alice are at a karaoke bar.*
*> Alice moves the beer to the trash can. Because in the story we find out,*
*> > Paul did not see the beer move to the trash can. Because in the story we find out,*
*> > > Fact From Story*
*> > > Fact From Story*
*> > > Commonsense Knowledge*

*Output:*
*Paul and Alice are at a karaoke bar.*
*> Alice moves the beer to the trash can. Because in the story we find out,*
*> > Paul did not see the beer move to the trash can. Because in the story we find out,*
*> > > Alice tricked Paul into looking "over there". | Fact From Story*
*> > > Alice pointed in the opposite direction of the trash can to Paul. | Fact From Story*
*> > > If you trick someone into looking elsewhere, they cannot see what happens in the other direction. | Commonsense Knowledge*

*Your Turn.*

*Scenario: Mason, a magician, has his magic deck of cards which are essential for his performance.*

*People in the story:*
*− Mason, a magician*

*– Adam, an audience member*
*– Patricia, an assistant*

*Respect the peoples roles when creating deductions and try to create motivations for them in the story that would influence if they see or do not see an event.*

*Current Tree:*
*Mason, a magician, has his magic deck of cards which are essential for his performance. | Deduced Root Conclusion*
*> opening scene | Deduced Fact*
*> > mason sees the top hat at the props table. | Fact From Story*
*> > mason sees the magic deck of cards at the backstage room. | Fact From Story*
*> > adam sees the top hat at the props table. | Fact From Story*
*> > adam sees the magic deck of cards at the backstage room. | Fact From Story*
*> > patricia sees the top hat at the props table. | Fact From Story*
*> > patricia sees the magic deck of cards at the backstage room. | Fact From Story*
*> mason moves the top hat to the stage. | Deduced Fact*
*> > adam did not see the top hat move to the stage. | Deduced Fact*
*> > patricia did not see the top hat move to the stage. | Deduced Fact*
*> patricia moves the magic deck of cards to the props table. | Deduced Fact*
*> > mason saw the magic deck of cards move to the props table. | Deduced Fact*
*> > adam saw the magic deck of cards move to the props table. | Deduced Fact*
*> mason moves the magic deck of cards to the stage. | Deduced Fact*
*> > adam did not see the magic deck of cards move to the stage. | Deduced Fact*
*> > patricia saw the magic deck of cards move to the stage. | Deduced Fact*

*Entailment Step to Complete:*
*Mason, a magician, has his magic deck of cards which are essential for his performance.*
*> mason moves the top hat to the stage. Because in the story we find out,*
*> > adam did not see the top hat move to the stage. Because in the story we find out,*
*> > > Complex Fact*
*> > > Complex Fact*
*> > > Commonsense Knowledge*

*Output:*

Listing 22: OP Intro generation prompt for the Story Generation stage

*Create an opening scene description for a story. It will be short. Only write about the objects we list out and their location. Your story MUST include each item and their location from the list. Your story also MUST indicate that all the people we give you saw the location of all these items!*

*You may use the description to infer the correct scenery to describe, but are only allowed to talk about the facts presented in the list.*

*You must state that everyone knows where everything is, "They were all aware of each items location" or something like that is a safe way to ensure this condition is met. Try to make this coherent with the story though. For example, if someone doesn't know the exact location you could say "Everyone was aware that the item was somewhere in the location, and they definitely all knew that the other item was in the other location", or something like this.*

*Here is an example.*

*Description: Alex is making Ramen and needs the noodles to cook.*

*Items and Locations:*
*– The pans are at the stove.*
*– The noodles are at the fridge.*
*– The kitchen knife is at the table.*

*Character 1:*
*Name: Alex*
*Role in story: A want to be chef*
*Motivation in story: To make a bowl of ramen.*

*Character 2:*
*Name: Carol*
*Role in story: the roommate*
*Motivation in story: Hanging out with Alex, she is also hungry.*

*Character 3:*
*Name: Joshua*
*Role in story: A random visitor*
*Motivation in story: Joshua was a friend of Alex but showed up unannounced and hungry.*

*Output: Alex and Carol were having a peaceful evening hanging out. Both getting a bit peckish, they decided to eat some ramen, which Alex had been practicing making for awhile now. Everyone knew Alex was the "Chef Friend", meaning that he was always cooking something delicious up. In fact, that's why a hungry friend, who showed up unannounced, Joshua decided to show up. All three of them noticed that the pans were already on the stove, perfect for ramen making! The kitchen knife was on the table, and the noodles were in the fridge.*

*Your turn.*

*Description: Ricky, with his weathered notebook in hand, was ready for a full day's studio session. His aim: perfect his upcoming album, for which his notebook was crucial. Emma, his producer and confidante, eager to bring Ricky's lyrics to life, shared Ricky's enthusiasm for the project. Meanwhile, Danny − the diligent studio assistant − was making sure all elements of the studio were meticulously managed for the recording session.*

*Items and Locations:*
*− the notebook is at the producer's desk*
*− the earphones are at the recording booth*

*Character 1:*
*Name: Ricky*
*Role in story: Singer−Songwriter*
*Motivation in story: Ricky is motivated to create his best music using his trusty notebook filled with personally crafted lyrics.*

*Character 2:*
*Name: Emma*
*Role in story: Music Producer*
*Motivation in story: Emma wants to support Ricky's creative process and record outstanding music to advance both Ricky's career and her own reputation as a successful music producer.*

*Character 3:*
*Name: Danny*
*Role in story: Music Studio assistant*
*Motivation in story: Danny's aspiration is to maintain an organized and conducive environment for Ricky's recording session.*

*Output:*

Listing 23: OP Prompt for generating a short description of why and how someone moved an item.

*Write a short description of this event happening. Only write about what is happening.*

*You will never add additional information! Never say "someone didn't see something" or infer someones ability to infer where something is. Never say "Unbeknownst" or anything like this!*
*Here is an example.*

*Facts:*
*− Timmy moves the car keys to the trash bin.*
*− Timmy saw the iphone at the trash bin when moving the car keys.*

*Output: Being ever clever and funny... in his own way... Timmy moved the keys to the trash bin. While he was doing that, he couldn't help but notice someone had put the iphone in the trash can too.*

*Here is another example.*

*Facts:*
*− Carol moves the noodles to the pantry.*

*Output: Carol, wanting to be cleanly, moved the noodles into the pantry.*

*Your turn.*

*Facts:*
*− mason moves the top hat to the stage.*

*Output:*

Listing 24: OP prompt for creating a paragraph on people's observations.

*Continue the story we have written so far by writing about the observational facts below. Only write about the facts and do not add new information. Never say "Someone saw" or "Did not notice" and never indicate if someone sees something, unless the only fact you have is "someone sees X".*

*Stick to the facts, there will be more information about the story that you can use to set the tone, but you should always use the facts as the main guide for the story.*

*Never mention the key items in the story:*
*− earphones*
*− notebook*

*There will be another event after this paragraph, so end this paragraph abruptly sticking only with the facts. Make no general statements. The last sentence should be something about the facts we listed out only. It should be a complete sentence.*

*Your story should take place during the most recent move. So while this is happening:*

*"Emma, engrossed in her thoughts, deftly moves the earphones to the producer's desk. She is preparing to tweak the sound settings, pre−empting Ricky's need for perfect audio in his performance."*

*the facts you will be writing about are happening at the same time.*

*Here is an example.*

*Description: Jerry, Marry and Timmy are getting ready for the day. Jerry has a huge meeting that he needs to prep for. Marry is excited to help Jerry for his meeting and to hear about it later that day. Timmy was getting ready for his test, but is being a bit inconsiderate to his dad, Jerry, with respect to his big meeting.*

*Character 1:*
*Name: Jerry*
*Role in story: the husband*
*Motivation in story: Jerry had a huge meeting coming up, one that could decide the fate of his career.*

*Character 2:*
*Name: Marry*
*Role in story: the wife*
*Motivation in story: Marry is always super supportive and wants the best for her family.*

*Character 3:*
*Name: Timmy*
*Role in story: the son*
*Motivation in story: Timmy has a huge test coming up in his school which he is nervous for and accidentally makes him a bit inconsiderate to everyone else.*

*Observational Facts:*
*− Jerry is cooking breakfast*
*− The trash bin is not in the kitchen.*
*− Marry is outside watering her garden.*
*− Marry has a window into the room with the trash bin.*

*Output: Jerry was hungry before he starts his day, so he was cooking his breakfast. The kitchen turned out to not have the trash bin though. Marry, always with her green thumb, was watering her garden and could see the trash bin through a nearby window.*

*Your turn.*

*Description: Ricky, with his weathered notebook in hand, was ready for a full day's studio session. His aim: perfect his upcoming album, for which his notebook was crucial. Emma, his producer and confidante, eager to bring Ricky's lyrics to life, shared Ricky's enthusiasm for the project. Meanwhile, Danny − the diligent studio assistant − was making sure all elements of the studio were meticulously managed for the recording session.*

*Character 1:*
*Name: Ricky*
*Role in story: Singer−Songwriter*
*Motivation in story: Ricky is motivated to create his best music using his trusty notebook filled with personally crafted lyrics.*

*Character 2:*
*Name: Emma*
*Role in story: Music Producer*
*Motivation in story: Emma wants to support Ricky's creative process and record outstanding music to advance both Ricky's career and her own reputation as a successful music producer.*

*Character 3:*
*Name: Danny*
*Role in story: Music Studio assistant*
*Motivation in story: Danny's aspiration is to maintain an organized and conducive environment for Ricky's recording session.*

*Observational Facts:*
*− Danny was engrossed in a conversation with a visiting sound engineer.*
*− Ricky was absorbed in a lyric composing session.*
*− The lyric composing session took place in a different area than where Emma was.*
*− The visitor was standing in a spot blocking Danny's view.*

*Output:*
*In the heart of the bustling studio, Ricky, Emma, and Danny readied themselves for a day of creating magic. Ricky, holding the helm as the gifted singer−songwriter, was poised for perfection, his precious notebook of lyrics awaiting its call to duty on the producer's desk. Emma, their dutiful and talented producer, was just as eager to breathe life into Ricky's lyrics. She was cognizant of the notebook's place at her desk, awaiting the melodies they would cultivate together.*

*Across the room, Danny, the diligent studio assistant, was doing his due diligence, keeping the earphones nestled in the recording booth. His aim was to ensure an optimized and meticulous environment for recording, a testament to his commitment to their shared mission. They were all aware of the arrangement − the notebook on the producer's desk, the earphones in the recording booth. Their shared consciousness of these items only intensified the anticipation; they were eager to turn the contents of a weathered notebook into a world−class album.*

*Ricky, with his weathered notebook of potent lyrics in hand, gently places it onto the piano. An air of creativity and anticipation lingers in the room, everyone aware that this was the first instrumental step in the creation of their masterpiece. In sync with the palpable creative energy, Ricky was engrossed in perfecting the rhythm of his song, preparing himself for an intense day ahead. Not too far away, Emma was sincerely engrossed in her role of musically steering the session. She was focussed on Ricky's progress, her eyes constantly monitoring him and her mind alive with ideas to enhance the music.*

*Meanwhile, Danny was diligently covering every corner of the studio. He was making his rounds, ensuring that the studio was prim and proper for Ricky's crucial session. As part of his tasks, he passed by Ricky several times, always careful not to interrupt the artist's flow.*

*Emma, engrossed in her thoughts, deftly moves the earphones to the producer's desk. She is preparing to tweak the sound settings, pre−empting Ricky's need for perfect audio in his performance.*

To generate the fact set $F$, we sample a collection of people, items, and locations. Only two items can be used and we incorporate three moves which will force one item to be moved twice. We then sample three plausible moves using GPT-4 that take the form of "x moves [item] to [location]". For everyone else not moving the item, they have a chance to see the move, "y sees [item] move to [location]" or to not see the move, "y does not see [item] move to [location]". We set the chance of seeing a move to 0.33. These facts are grouped into the move and added to the fact set $F$.

During the reasoning tree construction, we apply a keyword lookup validator to ensure the deductions being created are correct and meaningful. Specifically, we disallow any deductions that mention key locations or key items in the story. This ensures the deductions are not vacuous and state facts that could trivially solve the problem.

### G.4 TEAM ALLOCATION

Listing 25: Constructing the scenario for Team Allocation

*DESCRIPTION: A zoo requires caretakers for different animals and keepers for cleaning the exhibits.*
*Use the given scenario description to create a list of three people, two tasks, and two skills. Each skill should be associated with one of the tasks, and it should be that each skill is unique and orthogonal to the other and it's assigned task.*

*Rules*
*1) Never indicate in someones name their title or that they are good at any particular skill or task. For example, never say "Dr. Bob" as this implies they should be in charge of medical tasks.*

*Here's an example*

*Description: A heavy flux of customers walk into the coffee bar, you have to assign workers to the register and others to be baristas to handle the flow.*

*Output:*

*People: Sarah; Luis; John;*
*Tasks: Barista; Cashier*
*Skills: Can make coffee; Can handle customers*

*Your turn!*

*Description: A zoo requires caretakers for different animals and keepers for cleaning the exhibits.*

*Output:*

Listing 26: Team Allocation Deduction prompt for the Tree Construction stage

*We are creating a story about people being assigned to do certain jobs by a manager (you). You have to look into the personal details of your worker, their past experiences, their likes and dislikes, and their social interactions to determine what they are good and bad at doing.*

*To make this story, we've created a tree structure that outlines the narrative. Your job is to fill out the entailment trees that prove a set of facts on how well someone does at a particular job or how well two people work together.*

*For the facts involving skills and jobs, make the facts story driven focusing on past experience and personal details about the character. For example, if the fact is that "George is bad at tennis" you might say that "George was never athletic growing up", "George actively avoids sporting events now" and "If someone isn't athletic and avoids sporting events, they probably aren't good at sports like tennis".*

*For the facts involving teamwork, focus on the social element and how the two individuals have interacted in the past. For example, if the fact is that "George and Bill work well together" then you might say "George and Bill grab lunch from time to time.", "George and Bill were able to get a job done in average time working as a team last week", "If two people spend time together and are able to do adequate work together, then they probably work well together."*

*Most importantly, the facts should be interesting and personal; do not make them bland.*

*An entailment tree is a tree structure where intermediate nodes are entailed by their children. They create a natural language reasoning proof for some collection of facts.*

*To fill out this tree we need to complete an entailment. Completing an entailment is akin to filling out one subtree of the entailment tree. To fill in this step, you must follow the structure of the step.*

*Facts From Story are facts that will be explicitly stated when we write the story.*
*Commonsense Knowledge are facts that most people would agree are true and don't need to be explicitly said .*

*Complex Facts are facts that will be entailed by simpler facts from the story (they will be filled in later through a recursive call back to you!)*

*All facts for the step must combine to entail the root parent fact.*

*No facts may contradict the current structure tree.*

*Always match the exact structure of the entailment step I give you. Give the same number of Facts From Story and Commonsense Knowledge facts. Give them in the same order as well.*

*Each fact should be crucial to the deduction. Intentionally leave out details so that the other facts can account for them. If one fact is missing, the conclusion should not be entailed. Try not to reuse the same facts.*

*Always use the persons name instead of a pronoun like "He" or "She", if you know someones name, use the name.*

*Here's an example.*

*Scenario:*
*Paul and Alice are at a karaoke bar.*

*Current Tree:*
*Paul and Alice are at a karaoke bar. | Deduced Root Conclusion*
*> Opening Scene | Deduced Fact*
*> > Paul sees the microphone at the stage. | Fact From Story*
*> > Alice sees the microphone at the stage. | Fact From Story*
*> > Paul sees the beer at the bar. | Fact From Story*
*> > Alice sees the beer at the bar. | Fact From Story*
*> Paul moves the beer to the table. | Deduced Fact*
*> > Alice did not see the beer move to the table. | Deduced Fact*
*> > > Alice was facing away from the table. | Deduced Fact*
*> > > > Alice was talking to another patron. | Fact From Story*
*> > > > The other patron was facing the table. | Fact From Story*
*> > > > Usually people talk to each other while facing each other, so if one person is looking in one direction the other person is looking in the opposite direction. | Commonsense Knowledge*
*> > > If someone is facing away from something else, they cannot see things transpire near that something else. | Commonsense Knowledge*
*> Alice moves the microphone to the table | Deduced Fact*
*> > Alice saw the beer at the table when moving the microphone. | Fact From Story*
*> > Paul saw the microphone move to the table. | Deduced Fact*
*> > > Paul was drinking the beer at the table. | Commonsense Knowledge*
*> > > When something happens where a person is at, they usually see things that are happening there. | Fact From Story*
*> Alice moves the beer to the trash can. | Deduced Fact*
*> > Paul did not see the beer move to the trash can. | Fact From Story*

*Entailment Step to complete:*
*Paul and Alice are at a karaoke bar. Because,*
*> Alice moves the beer to the trash can. Because,*
*> > Paul did not see the beer move to the trash can. Because,*
*> > > Fact From Story*
*> > > Fact From Story*
*> > > Commonsense Knowledge*

*Output:*
*Paul and Alice are at a karaoke bar. Because,*
*> Alice moves the beer to the trash can. Because,*
*> > Paul did not see the beer move to the trash can. Because,*
*> > > Alice tricked Paul into looking "over there". | Fact From Story*
*> > > Alice pointed in the opposite direction of the trash can to Paul. | Fact From Story*
*> > > If you trick someone into looking else where, they cannot see what happens in the other direction. | Commonsense Knowledge*
*Here is another example.*

*Scenario:*
*Your dog has just pooped on the neighbours yard. The neighbour glares in your direction and comes forward ... he says "Hey you! What do you think you are letting your dog do on my nice yard right here!"*

*Current Tree:*
*Punch the neighbour square in the nose. | Deduced Root Conclusion*
*> It'll look cool and this is a pro. | Deduced Fact*
*> > People think fighting is cool where you live. | Fact From Story*
*> > You would be fighting. | Fact From Story*
*> > Doing something people think is cool will make you cool too. | Commonsense Knowledge*
*> It'll look cool unless... | Fact From Story*

*Entailment Step to complete:*
*Jose has a means. Because,*
*> Fact From Story*
*> Fact From Story*
*> Commonsense Knowledge*

*Output:*
*Jose has a means. Because,*
*> Jose has access to heroin. | Fact From Story*
*> Jose knows how much heroin is needed for an overdose. | Fact From Story*
*> Having access to heroin and knowing how much heroin is required to overdose implies you could have intentionally given the victim a dose of lethal heroin providing a means for murder. | Commonsense Knowledge*
*Here is another example.*

*Scenario:*

*Victim: Jared*
*Crime scene: Public park bench*
*Murder weapon: Heroin overdose*
*Suspect: Jose*
*Suspect's role in story: Drug user*
*Suspects motive: Public humiliation*

*Current Tree:*
*Jose is a murderer. | Deduced Root Conclusion*
*> Jose has a means. | Fact From Story*
*> Jose has a motive | Fact From Story*
*> Jose has an opportunity. | Fact From Story*

*Entailment Step to complete:*
*Paul and Alice are at a karaoke bar. Because,*
*> Alice moves the beer to the trash can. Because,*
*> > Complex Fact*

*Output:*
*Paul and Alice are at a karaoke bar. Because,*
*> Alice moves the beer to the trash can. Because,*
*> > Paul did not see the beer move to the trash can. | Complex Fact*

*Your Turn.*

*Scenario: A zoo requires caretakers for different animals and keepers for cleaning the exhibits.*

*Current Tree:*
*A zoo requires caretakers for different animals and keepers for cleaning the exhibits. | Deduced Root Conclusion*
*> Olivia is bad at exhibit cleaner. Because we find out in the story that, | Deduced Fact*
*> Olivia is bad at animal caretaker. Because we find out in the story that, | Deduced Fact*
*> Mia is bad at animal caretaker. Because we find out in the story that, | Deduced Fact*
*> Mia is good at exhibit cleaner. Because we find out in the story that, | Deduced Fact*
*> Alex is okay at exhibit cleaner. Because we find out in the story that, | Deduced Fact*
*> Alex is bad at animal caretaker. Because we find out in the story that, | Deduced Fact*
*> Alex and Mia work okay together. Because we find out in the story that, | Deduced Fact*
*> Olivia and Alex work badly together. Because we find out in the story that, | Deduced Fact*
*> Olivia and Mia work badly together. Because we find out in the story that, | Deduced Fact*

*Entailment Step to Complete:*
*A zoo requires caretakers for different animals and keepers for cleaning the exhibits. Because,*
*> Olivia is bad at exhibit cleaner. Because we find out in the story that,*
*> > Fact From Story*
*> > Fact From Story*
*> > Commonsense Knowledge*

*Output:*

Listing 27: Team Allocation story creation prompt for the Story Generation stage

*You will write a short story given the description of a scenario and a list of facts. You must include every fact in the story.*

*You should take the role of a manager or leader, someone who can assign people to skills.*

*Your job is to find the perfect assignment of each person to a single skill. You will not say what this perfect assignment is, that is left to the reader to decide.*

*Instead, you must write out the narrative as if it were a short story. Make this story coherent and flow nicely.*

*Most importantly, make this story interesting!*

*Start with an introduction.*
*Introduce each person:*
*– Olivia*
*– Alex*
*– Mia*

*And the tasks the manager has to assign them to (mention these by name in the beginning, i.e. "And they all had to be assigned to cleaning and sales" or something.):*
*– Animal Caretaker*
*– Exhibit Cleaner*

*Scenario description: A zoo requires caretakers for different animals and keepers for cleaning the exhibits.*
*Facts that you must include in your story:*
*– Alex admitted that he feels uncomfortable around animals larger than him.*
*– Alex and Mia used to cooperate well in the same high school club, often collaborating on fundraising initiatives.*
*– Alex expressed his discomfort around Olivia, mentioning how her avoidance makes him feel ostracized.*
*– Alex expressed in the past no desire to pursue furthering his knowledge of animals outside of pets.*
*– Alex has shown mild interest in keeping his surroundings neat, but he doesn't go out of his way to tidy up.*
*– Alex sometimes voluntarily helped with exhibit cleaning when he was a volunteer at a cat shelter.*
*– At lunch breaks, Alex and Mia engage in friendly conversations and share a similar sense of humor.*
*– Mia finds Olivia too passive and non–confrontational, which results in stewing resentment and lack of open communication.*
*– Mia gets nervous and excessively worried at loud noises.*
*– Mia insists on a spotless living environment, often spending free time in her own home cleaning and organizing.*

− *Mia mentioned a discomforting encounter with a roaring lion that left her shaken.*
− *Mia showed enthusiasm when she found out about the cleaning tasks at the zoo, expressing her belief that the exhibitors need to mirror the animals' natural habitats as closely as possible.*
− *Olivia avoids Alex during their shared shifts because of an old workplace disagreement.*
− *Olivia experienced sneezing fits and allergies during her custodial job at a school where cleaning was part of her task.*
− *Olivia has admitted her fear of wild animals due to a past traumatic experience with a chimp.*
− *Olivia has explicit disagreements with Mia's work habits, often criticizing her for cleaning the exhibits too thoroughly.*
− *Olivia mentioned her allergy to dust and pollen.*
− *Olivia often avoids the animal exhibits during her visits to the zoo.*

*Output:*

Listing 28: Team Allocation introduction rewrite for the Story Generation stage

*I am writing a story that is similar to a word problem where a manager has to assign the right worker to a task. Here's the full story:*

*In the bustling world of the Redwood Zoo, at the crux of the city's great jungle, the need for careful assigning of roles played a major role in the smooth function of the behemoth. As the manager, the delicate balance of where to assign Olivia, Alex, and Mia, to the roles of Animal Caretaker and Exhibit Cleaner was an intriguing puzzle to solve. Each with their unique temperament, skills, and admittedly, eccentricities, made the task all the more interesting.*

*Let's begin with Alex, the tall lad with bright eyes, whose history with the mighty beast of the animal kingdom, lacked a certain comfort. The lad, known to express an almost innate unease around animals larger than him, fell short of the prerequisites for an Animal Caretaker. His comfort zone extended to the four−legged companions in our homes, a sentiment I withheld from the petting zoo section of our park. Yet, his association and collaboration with Mia had seen quite the successes in their high school club's fundraising initiatives.*

*However, his relationship with the gentle Olivia was not as seamless. Alex often mentioned feeling ostracized due to Olivia's tendency to maintain her distance. This seemingly innocent avoidance stirred disquiet within our hushed ranks. And all this, stemming from a disagreement rooted in their previous shared workplaces. Unresolved perhaps, but a factor nonetheless.*

*Then, there was Mia, the determined bright spark, whose affinity for cleanliness would often bemuse us. She would spend her spare time in her immaculate home cleaning and reorganizing, while her enthusiasm for a spotless Exhibit could not be underestimated. However, her overly thorough methods would invariably result in clashes with Olivia, who criticized her for crossing some form of unspoken boundary.*

*Mia too had her phobias, the gravelly roars of the zoo's majestic lion had once left her shaken and worried. Loud noises had a similar effect leaving her in a state of nervous terror, much like that of the petite animals held within our barriers. Yet, she was all smiles and peasant conversation around Alex during lunch breaks, sharing a sense of humor that lightened the mood of our everyday grind.*

*Finally, subdued Olivia, a soul strangled with allergies, and a deep−seated fear for wild animals. An incident with a chimp in her past wove tales of nightmarish betrayal, enough to send her away from the animal exhibits during her zoo visits. Potent elements of dust and pollen resulted in uncontrolled sneezing fits, a remainder from her days at the school as a custodian, responsible for the cleanliness and maintenance.*

*Three souls; Animals to be cared for, Exhibits to be cleaned. Assigning them was always going to be an enigma for anyone navigating the zoological labyrinth. Love for animals, discomfort, alliances, conflicts; each factor extraordinarily crucial in shaping not just the overall productivity but also the personal growth of each of these individuals at the Redwood Zoo.*

*−−−*

*I want you to rewrite the introduction paragraph to introduce each person and the tasks. Do not assign people to a task (this is the question I want to ask the readers), just introduce them. Rewrite the introduction:*

*In the bustling world of the Redwood Zoo, at the crux of the city's great jungle, the need for careful assigning of roles played a major role in the smooth function of the behemoth. As the manager, the delicate balance of where to assign Olivia, Alex, and Mia, to the roles of Animal Caretaker and Exhibit Cleaner was an intriguing*

> *puzzle to solve. Each with their unique temperament, skills, and admittedly, eccentricities, made the task all the more interesting.*
>
> *Make sure it includes:*
>
> *Mentions to these peoples names (do not describe them)*
> *– Olivia*
> *– Alex*
> *– Mia*
>
> *And the tasks the manager has to assign them to (mention these by name in the beginning, i.e. "And they all had to be assigned to cleaning and sales" or something.)*
> *– Animal Caretaker*
> *– Exhibit Cleaner*
>
> *It should be short. No longer than the original introduction.*

In the Tree Template Construction stage, we build a matrix $M$ where the first two columns are scores for each person's abilities on a task, and the last three are their relationship scores. Once we have that matrix, we can generate facts for each person's skill level and relationship score.

For skill level facts, we look at each person's proficiency at a skill and set it either to "bad", "okay", or "good", then add to the fact set $F$, "[person] is [skill level label] at [skill]".

For relationship scores, we look at each unique pair's score and label it as either "badly", "okay", or "well". We can then add the fact, "[person 1] and [person 2] work [relationship label] together." to the fact set $F$.

Once we have all the facts in $F$ we can pass these to the Reasoning Tree Construction stage. Team Allocation required no validators in the construction of the reasoning tree nor the story to create valid contexts.

## H  IMPLEMENTATION DETAILS, API COST

### H.1  IMPLEMENTATION DETAILS: CREATING AND EVALUATING MuSR

**Creating MuSR**  GPT-4 is used in all creation prompts for creating MuSR. We do not change any parameters from the defaults when prompting GPT-4. Temperature and top-p are set to 1.0.

For the Murder Mystery domain, we set the maximum depth of the reasoning tree to 4. Only two suspects are allowed to be included in the generation process. To add diversity, we sample a victim's name, a murder weapon, a crime scene, suspect names, suspect motives, suspect roles in the story, and suspicious facts. All of these facts are used in the tree construction stage to guide the created deductions to include more diversity.

For Object Placements, we set the maximum depth of the reasoning tree to 3. Only three moves are allowed to be included in a story, and only two items which forces one item to be moved twice. The chance for a person to see a move is set to 33%. To include diversity, we sample scenarios where people need to move items around, then we sample their names, moves, and motivations for moving the item.

Finally, for Team Allocation, we set the maximum depth of reasoning to 2. We sample a scenario for diversity similar to Object Placements, then sample tasks and people's names.

**Evaluating MuSR**  During evaluation, we use the "gpt-4" and "gpt-3.5-turbo" chat endpoints. For Llama2 and Vicuna models, we use Hugging Face's implementations (Wolf et al., 2020) and use bitsandbytes quantization (Dettmers et al., 2022) when running inference.[2] No changes to the inference parameters were made when sampling from the models. For all local models, we ran them on on a machine with 4xNVIDIA RTX A6000 48GB cards.

---

[2]We did not see any significant performance increases in some of the smaller models when we did not use quantization.

Table 9: Estimates of how long it takes to run one of our dataset domains as well as cost. All prices and times are based off of GPT4 and the CoT+ 1-Shot prompt variant.

| | Murder Mystery | | Object Placements | | Team Allocation | |
|---|---|---|---|---|---|---|
| | Time | Cost | Time | Cost | Time | Cost |
| | 2:00:00 | $27 | 1:40:00 | $23 | 2:10:00 | $17 |

## H.2 API COSTS AND TIMES

Table 9 shows the average costs and times to run the evaluation of each domain using GPT4 with the CoT+ 1-Shot prompt variant.

## I EVALUATION PROMPTS AND OUTPUTS

### I.1 EVAL PROMPTS

**Murder Mystery** CoT+ evaluation prompt for Murder Mysteries.

Listing 29: Murder Mystery COT+ Prompt

{{*context*}}

*Who is the most likely murderer?*

*Pick one of the following choices:*
{{*answer choices*}}

*You must pick one option. Before selecting a choice, explain your reasoning step by step. The murderer needs to have a means (access to weapon), motive (reason to kill the victim), and opportunity (access to crime scene) in order to have killed the victim. Innocent suspects may have two of these proven, but not all three. An innocent suspect may be suspicious for some other reason, but they will not have all of motive, means, and opportunity established.*

*If you believe that both suspects have motive, means, and opportunity, you should make an educated guess and pick the one for whom these are best established. If you believe that neither suspect has all three established, then choose the suspect where these are most clearly established. Explain your reasoning step by step before you answer. Finally, the last thing you generate should be "ANSWER: (your answer here, including the choice number)"*

**Object Placement** CoT+ evaluation prompt for Object Placements.

Listing 30: Object Placements COT+ Prompt

{{*context*}}

*Based on this story, we want to identify where someone believes that a certain object is at the end of the story. In order to do that, you need to read the story and keep track of where they think the object is at each point. When an object is moved, the person may observe its new location if they saw it move.*

*To see where an object ends up, they must be able to see the location that it moves to and not be too distracted by what they are doing. If they do not observe the object moving, then they will still believe it to be in the last location where they observed it.*

*Which location is the most likely place Danny would look to find the earphones given the story?*

*Pick one of the following choices:*
{{*answer choices*}}

*You must pick one option. Explain your reasoning step by step before you answer. Finally, the last thing you generate should be "ANSWER: (your answer here, including the choice number)"*

**Team Allocation**    CoT+ evaluation prompt for Team Allocation.

Listing 31: Team Allocation COT+ Prompt

> {{*context*}}
>
> *Given the story, how would you uniquely allocate each person to make sure both tasks are accomplished efficiently?*
>
> *Pick one of the following choices:*
> {{*answer choices*}}
>
> *You must pick one option. The story should allow you to determine how good each person is at a skill. Roughly, each person is either great, acceptable, or bad at a task. We want to find an optimal assignment of people to tasks that uses their skills as well as possible. In addition, one task will have to have two people assigned to it. The effectiveness of their teamwork (great team, acceptable team, or bad team) also impacts the overall quality of the assignment.*
>
> *When two people need to work on a task and one is bad at it, they don't necessarily benefit from the other person being good, unless they work well together.*
>
> *With different strengths, weaknesses, and interpersonal dynamics at play, you should allocate your team to find the single assignment to ensure that the tasks overall are completed as effectively as possible.*
>
> *Explain your reasoning step by step before you answer. Finally, the last thing you generate should be " ANSWER: (your answer here, including the choice number)"*

Listing 32: Team Allocation PAL Prompt

> {{*context*}}
>
> *Given the story, how would you uniquely allocate each person to make sure both tasks are accomplished efficiently?*
>
> *Pick one of the following choices:*
> *1 – {{choices[0]}}*
> *2 – {{choices[1]}}*
> *3 – {{choices[2]}}*
>
> *You must pick one option. The story should allow you to determine how good each person is at a skill. Roughly, each person is either great, acceptable, or bad at a task. We want to find an optimal assignment of people to tasks that uses their skills as well as possible. In addition, one task will have to have two people assigned to it. The effectiveness of their teamwork (great team, acceptable team, or bad team) also impacts the overall quality of the assignment.*
>
> *When two people need to work on a task and one is bad at it, they don't necessarily benefit from the other person being good, unless they work well together.*
>
> *With different strengths, weaknesses, and interpersonal dynamics at play, you should allocate your team to find the single assignment to ensure that the tasks overall are completed as effectively as possible.*
>
> *Now let's write a python function to solve it. Our program will contain three steps:*
>
> *First, find out how good a person is at the two tasks. Assign it to a variable. Use 3 for great, 2 for acceptable, and 1 for bad.*
> *For example 'Alice_at_speaking = 2' means a person Alice is acceptable at the task of speaking. When you assign values to individual scores, show your supporting facts (excerpt from the question) in the comments.*
>
> *Second, find out the cooperation score for two people, use 3 for great, 2 for acceptable, and 1 for bad.*
> *For example 'Alice_with_Bob = 3' means Alice works great with Bob. When you assign values to cooperation scores, show your supporting facts (excerpt from the question) in the comments.*
>
> *Third, find the score of each option, and choose the best scored option. Concretely, the score of an option of assignment is the sum of all the scores, including individual scores and cooperation scores.*
> *Hint: you can use 'max([(option1_score, '1'),(option2_score, '2'),(option3_score, '3')])[1]' to find out the best option.*

> *Please write a python function 'assign_task' which returns the answer choice in your response.*

## I.2 INCORRECT EVALUATIONS FROM GPT-4

### Listing 33: Murder Mystery Reasoning Error

*In the cold echo of a deserted hockey rink, Otis was found dead from a grenade explosion, and now Detective Winston must decipher the clues, suspecting Ervin and Andrew are somehow involved.*

*Detective Winston rubbed his temples while the facts flickered across his computer screen. Otis, the victim, would most definitely be missed. His usual vibrant energy that filled the hockey rink was replaced by cold, solemn silence.*

*Some contradictions needed to be resolved. The crime scene at the hockey rink was empty, both at the time of the murder and throughout the investigation. Winston's only lead was Andrew, who was an old military pal of Otis. Winston heard that Andrew had recently switched his career to become a solo hiking guide. Indeed, a remarkable part of his solo hikes was his ability to help hikers they came across on the way overcome environmental dangers—a way of applying his skills acquired from his military service. It was an unusual practice in the industry but allowed due to his military expertise in bomb disposal—an odd career choice for a man of such skills.*

*Winston drove to the outskirts of the town, where Andrew lived in a remote cabin. Andrew's solo job didn't always welcome company, but that didn't seem to bother him. Rumor had it that he uses his military skills to help strangers on the trails overcome environmental dangers—detonating falling rocks, perhaps?*

*Once in the cabin, Winston noticed several hiking supplies meticulously arranged. He inspected them and found, hidden strategically among the normal hiking necessities—food, water, emergency kit—were several small explosives. Chills ran down his spine as he remembered a recent break—in at a military base.*

*"I see you have explosives mixed with your hiking guide supplies," Winston said, trying to evoke a reaction.*

*Andrew replied calmly, "Those are just for emergencies."*

*Listening to Andrew, Winston remembered that Otis had been friendly with him. Otis was always open with Andrew about his daily habits, even shared his calendar with him. Andrew was the only person who knew Otis would be alone at the hockey rink that evening.*

*On his way out, Winston remembered Andrew's frank words, "I don't like to hang out after work."*

*It seemed somewhat true, living far from town, working alone, and being prone to avoid after—work hangouts. Winston thought to himself: Andrew did like to be alone. It all seemed so strange, the grenade murder, the empty hockey rink, Andrew's distant demeanor. However, he couldn't draw any obvious conclusions just yet. He needed more.*

*Detective Winston sat on the chilly bleachers of the local hockey rink, a place as familiar to him as it was to Ervin, the mild—mannered Rabbi who had never missed a game. The rink was still splattered with telltale signs of the merciless violence that had taken place there. Otis had been killed during a game, a seemingly jovial event turned nightmarish by the unexpected detonation of a grenade.*

*Taking his eyes off the crime scene momentarily, he scanned his surroundings and spotted his next person of interest: Ervin, a massive hockey fan, and a staunch supporter of their local team. His presence at the game had been expected, as his dedicated attendance was part of the everyday routine. Some of the spectators had mentioned Ervin stepping out of the stadium briefly, interestingly around the same time Otis was killed.*

*Not missing his beat, Winston approached Rabbi Ervin, who was still in deep shock at the loss of a fellow spectator and friend, "Rabbi Ervin?"*

*The Rabbi looked up to meet the detective's prying eyes, "I am Ervin, indeed. And you are?"*

*"I am Detective Winston, investigating the recent unfortunate event."*

*Observing Ervin's face closely, Winston began: "The strangest thing happened a few days ago. Otis came to me and told me he was planning to go to the authorities. He knew something was off, dangerous even. I guess*

*we didn't react quickly enough. He mentioned your name, Ervin, and something about another life before the synagogue..."*

*Ervin was quiet, his gaze distant but thoughtful. "Yes, Otis did confront me. He saw me moving some funds around. He saw, Detective, he saw me committing the crime which I've regretted every day since," Ervin confessed.*

*Winston let the confession sink in before shifting the subject subtly. "I heard you were in the military?"*

*Ervin smiled, just barely, nodding with a touch of pride. "Yes, I am a former military officer. Had a special training in demolitions you see, before I found my calling in religion."*

*Winston returned his smile, "I suppose it's a bit of a leap from demolitions to the pulpit."*

*"Just a tad," Ervin's laughter is soft. "But the old life stays with you in ways you wouldn't expect. I've got a collection of war memorabilia back home. host some of my friends over sometimes, they marvel at these artifacts. I even got grenades in there. Nothing active, of course. They tend to catch the eye."*

*Winston excused himself, leaving Ervin to mourn at the crash of their sanctuary of play turned battleground.*

*Who is the most likely murderer?*

*Pick one of the following choices:*
*1 − Ervin*
*2 − Andrew*

*You must pick one option. Before selecting a choice, explain your reasoning step by step. The murderer needs to have a means (access to weapon), motive (reason to kill the victim), and opportunity (access to crime scene) in order to have killed the victim. Innocent suspects may have two of these proven, but not all three. An innocent suspect may be suspicious for some other reason, but they will not have all of motive, means, and opportunity established.*

*If you believe that both suspects have motive, means, and opportunity, you should make an educated guess pick the one for whom these are best established. If you believe that neither suspect has all three established, then choose the suspect where these are most clearly established. Explain your reasoning step by step before you answer. Finally, the last thing you generate should be "ANSWER: (your answer here, including the choice number)"*

*[[MODEL OUTPUT]]*

*Looking at Detective Winston's investigation into the characters of Ervin and Andrew, we can examine the means, motive, and opportunity for each.*

*First, Andrew, an ex−military guy and Otis's old friend, has a motive. His recent change to a hiking guide involves using his expertise in bomb disposal which demonstrates the means. Interestingly, the explosives noticed with the hiking supplies could very well be utilized for a crime like this. Additionally, his opportunity is clear, as he knew Otis' schedule, including his solitude in the hockey rink.*

*Ervin, the Rabbi and hockey fan, also has a motive, as his secret funds−moving activity was found out by Otis. Like Andrew, Ervin also has a military background and knowledge of demolitions, hinting at the means. His opportunity lies in the fact that he was present at the hockey rink during the game, where Otis was killed, and reportedly stepped out of the stadium briefly around the same timeframe.*

*However, Ervin's collection of war memorabilia including grenades were described as "nothing active." Unless this is a misdirection and he does possess live grenades, this could imply that his grenades wouldn't be functional for the act. Andrew, on the other hand, has been found with actual explosives that are implied to be active. We can also note that Andrew's solitary activity and lifestyle further suggest his ability to commit such a crime without leaving apparent witnesses.*

*Therefore, while both candidates present motive, means, and opportunity, it seems that Andrew's capacity to commit this crime is better substantiated. He has a clear opportunity knowing Otis's schedule, an explicit motive with his recent career change contrasted with his military prowess, and useful means evidenced by the explosives found among his hiking supplies.*

*ANSWER: 2 − Andrew*

*[[Correct Answer]]*

*1*

Listing 34: Team Allocation Reasoning Error

*Amidst the vibrant chaos of the Redwood Zoo, nestled in the heart of the city's sprawling jungle, the task of assigning roles was a crucial cog in the machinery of its operation. As the manager, the responsibility of allocating Olivia, Alex, and Mia to the positions of Animal Caretaker and Exhibit Cleaner presented an intriguing conundrum. Each individual, with their distinct personalities and skill sets, added a layer of complexity to this assignment puzzle.*

*Let's begin with Alex, the tall lad with bright eyes, whose history with the mighty beast of the animal kingdom, lacked a certain comfort. The lad, known to express an almost innate unease around animals larger than him, fell short of the prerequisites for an Animal Caretaker. His comfort zone extended to the four−legged companions in our homes, a sentiment I withheld from the petting zoo section of our park. Yet, his association and collaboration with Mia had seen quite the successes in their high school club's fundraising initiatives.*

*However, his relationship with the gentle Olivia was not as seamless. Alex often mentioned feeling ostracized due to Olivia's tendency to maintain her distance. This seemingly innocent avoidance stirred disquiet within our hushed ranks. And all this, stemming from a disagreement rooted in their previous shared workplaces. Unresolved perhaps, but a factor nonetheless.*

*Then, there was Mia, the determined bright spark, whose affinity for cleanliness would often bemuse us. She would spend her spare time in her immaculate home cleaning and reorganizing, while her enthusiasm for a spotless Exhibit could not be underestimated. However, her overly thorough methods would invariably result in clashes with Olivia, who criticized her for crossing some form of unspoken boundary.*

*Mia too had her phobias, the gravelly roars of the zoo's majestic lion had once left her shaken and worried. Loud noises had a similar effect leaving her in a state of nervous terror, much like that of the petite animals held within our barriers. Yet, she was all smiles and peasant conversation around Alex during lunch breaks, sharing a sense of humor that lightened the mood of our everyday grind.*

*Finally, subdued Olivia, a soul strangled with allergies, and a deep−seated fear for wild animals. An incident with a chimp in her past wove tales of nightmarish betrayal, enough to send her away from the animal exhibits during her zoo visits. Potent elements of dust and pollen resulted in uncontrolled sneezing fits, a remainder from her days at the school as a custodian, responsible for the cleanliness and maintenance.*

*Three souls; Animals to be cared for, Exhibits to be cleaned. Assigning them was always going to be an enigma for anyone navigating the zoological labyrinth. Love for animals, discomfort, alliances, conflicts; each factor extraordinarily crucial in shaping not just the overall productivity but also the personal growth of each of these individuals at the Redwood Zoo.*

*Given the story, how would you uniquely allocate each person to make sure both tasks are accomplished efficiently?*

*Pick one of the following choices:*
*1 − Animal Caretaker: Alex, Exhibit Cleaner: Mia and Olivia*
*2 − Animal Caretaker: Olivia, Exhibit Cleaner: Alex and Mia*
*3 − Animal Caretaker: Mia, Exhibit Cleaner: Alex and Olivia*

*You must pick one option. The story should allow you to determine how good each person is at a skill. Roughly, each person is either great, acceptable, or bad at a task. We want to find an optimal assignment of people to tasks that uses their skills as well as possible. In addition, one task will have to have two people assigned to it. The effectiveness of their teamwork (great team, acceptable team, or bad team) also impacts the overall quality of the assignment.*

*When two people need to work on a task and one is bad at it, they don't necessarily benefit from the other person being good, unless they work well together.*

*With different strengths, weaknesses, and interpersonal dynamics at play, you should allocate your team to find the single assignment to ensure that the tasks overall are completed as effectively as possible.*

*Explain your reasoning step by step before you answer. Finally, the last thing you generate should be "
ANSWER: (your answer here, including the choice number)"*

*[[MODEL OUTPUT]]*

*Reading through the story, we learn a lot about each individual's strengths, weaknesses, fears, and
relationships.*

*Starting with Alex, we find that he has discomfort around large animals, which indicates he may not be suited
for the role of Animal Caretaker. However, Alex has a good working relationship with Mia from their previous
fundraising activities. However, his relationship with Olivia is strained.*

*Next, we have Mia. She loves cleanliness and would likely be ideal for an Exhibit Cleaner role. Nonetheless,
she doesn't get along well with Olivia and has a deep fear of large, loud animals. Her bond with Alex is
strong, as they share humorous exchanges during their breaks.*

*Finally, Olivia is mentioned. Her allergies and fear of wild animals make her a poor fit for the Animal
Caretaker role. She also seems to clash with Mia over cleaning methods. While her relationship with Alex is
not explicitly described as negative, Alex has stated that he feels left out by Olivia, suggesting a strained
relationship.*

*Given these insights, let's look at the options:*

*Option 1 − Wouldn't work well because Olivia and Mia do not have a good relationship, so them working
together on cleaning could be problematic. Also, Alex is not comfortable around large animals, making him a
poor fit for the Animal Caretaker role.*

*Option 2 − May not work as efficiently because Olivia has problems with allergies, which could be triggered
by cleaning dust and pollen. Her fear of wild animals also makes her a poor choice for the Animal Caretaker.
Again, Alex and Mia have a good relationship, but both are not comfortable with large animals.*

*Option 3 − appears to be the best choice here. Mia, with her love for cleanliness, would be excellent as an
Exhibit Cleaner, while the strained relationship between Olivia and Mia might not be too much of an issue if
they work on different exhibits. Alex could bring his excellent teamwork and communication with Mia to the
Exhibit Cleaning team, helping to mitigate Olivia's involvedness. Mia's fear of animals and Alex's discomfort
around them won't be an issue as they would not have to interact closely with the animals in their roles as
Exhibit Cleaners.*

*So, after considering everyone's skills, comfort levels, and interpersonal dynamics, the answer would be:*

*ANSWER: 3 − Animal Caretaker: Mia, Exhibit Cleaner: Alex and Olivia.*

*[[Correct Answer]]*

*2*

Listing 35: Object Placements Reasoning Error

*In the heart of the bustling studio, Ricky, Emma, and Danny readied themselves for a day of creating magic.
Ricky, holding the helm as the gifted singer−songwriter, was poised for perfection, his precious notebook of
lyrics awaiting its call to duty on the producer's desk. Emma, their dutiful and talented producer, was just as
eager to breathe life into Ricky's lyrics. She was cognizant of the notebook's place at her desk, awaiting the
melodies they would cultivate together.*

*Across the room, Danny, the diligent studio assistant, was doing his due diligence, keeping the earphones
nestled in the recording booth. His aim was to ensure an optimized and meticulous environment for recording,
a testament to his commitment to their shared mission. They were all aware of the arrangement − the
notebook on the producer's desk, the earphones in the recording booth. Their shared consciousness of these
items only intensified the anticipation; they were eager to turn the contents of a weathered notebook into a
world−class album.*

*Ricky, with his weathered notebook of potent lyrics in hand, gently places it onto the piano. An air of creativity and anticipation lingers in the room, everyone aware that this was the first instrumental step in the creation of their masterpiece. In sync with the palpable creative energy, Ricky was engrossed in perfecting the rhythm of his song, preparing himself for an intense day ahead. Not too far away, Emma was sincerely engrossed in her role of musically steering the session. She was focussed on Ricky's progress, her eyes constantly monitoring him and her mind alive with ideas to enhance the music.*

*Meanwhile, Danny was diligently covering every corner of the studio. He was making his rounds, ensuring that the studio was prim and proper for Ricky's crucial session. As part of his tasks, he passed by Ricky several times, always careful not to interrupt the artist's flow.*

*Emma, engrossed in her thoughts, deftly moves the earphones to the producer's desk. She is preparing to tweak the sound settings, pre−empting Ricky's need for perfect audio in his performance. Diverting from his rounds, Danny found himself in the midst of a stirring conversation with a visiting sound engineer. Knowledge flowed between them, illuminating the studio's atmosphere, the engineer's insight bringing a new perspective into Danny's role. Ricky, ensconced in his own world, was in deep discussion with the blank page before him. The daunting silence of the empty studio buzzed with his focus, as he honed his lyrics to perfection in a space separate from the producer's. The visitor, oblivious to the careful choreography of the studio session, stood blocking Danny's general overview of the studio space.*

*Delicately lifting Ricky's notebook, Danny orchestrates its move to the producer's desk. At the desk, he glimpses a pair of earphones indirectly drawing his attention amidst his routine of tidying up. Emma, from the isolated interior of a sound−proofed booth, lent her ears diligently to already recorded tracks, pouring over them for any room for improvement. Being lost in the music was her way of paying homage to her craft − an unspoken ritual she followed each time she embarked on a music production journey. The entirety of her focus was consumed by the musical notes and rhythm filtering through the studio speakers.*

*Concurrently, Ricky was absorbed in the act of playing his guitar. His fingers navigated deftly over the strings , lost in an intimate dance with the instrument. As he played, the melodic strums reverberated throughout the studio, filling it with an infectious pulse that hinted at the birth of yet another musical masterpiece. Despite the flurry of activity around him, Ricky was lost in a world of his own, operating on a singular vision of delivering his best performance.*

*In the meantime, Danny was continuing his cautious management of the studio, ensuring that everything fell into place for the optimum recording session. His watchful eyes were scanning every corner, taking stock of the minor details that could impact the session. However, the design of the studio didn't allow for an unrestricted view into all the corners. The sound booth, where Emma was engrossed in her work, was out of his visual range. The seclusion provided by the booth, although crucial for immersive work, also acted as a barrier for Danny's comprehensive vigilance.*

*As the day progressed, the studio was entwined in a concerted symphony of dedication and workmanship, the trio, each engrossed in their pursuit, working together to create the best version of Ricky's impending album. As the final note of the day rang through the studio, each person revelled in the satisfaction of another day done right, another step closer towards the realization of Ricky's artistic vision.*

*Within the dynamic dance of the day's events, the relationships of the trio sang a compelling tune. Each individual played their crucial part in the creation of the impending masterpiece − Ricky with his raw talent, Emma with her passion for perfection, and Danny with his meticulous eye for detail. And as the lights faded on another day of creation, they could sense the beginning of an important chapter in their artistry, a silence collecting the scattered notes of the day, signing off on another critical step in the journey of Ricky's upcoming album.*

*Based on this story, we want to identify where someone believes that a certain object is at the end of the story. In order to do that, you need to read the story and keep track of where they think the object is at each point. When an object is moved, the person may observe its new location if they saw it move.*

*To see where an object ends up, they must be able to see the location that it moves to and not be too distracted by what they are doing. If they do not observe the object moving, then they will still believe it to be in the last location where they observed it.*

*Which location is the most likely place Danny would look to find the earphones given the story?*

*Pick one of the following choices:*
*1 − piano*

*2 − producer's desk*
*3 − recording booth*

*You must pick one option. Explain your reasoning step by step before you answer. Finally, the last thing you generate should be "ANSWER: (your answer here, including the choice number)"*

*[[MODEL OUTPUT]]*

*In the story, Danny prepared the recording booth by placing the earphones there. He was then occupied in ensuring other studio duties and discussing with a visiting sound engineer. He did not see Emma move the earphones from the recording booth to the producer's desk, nor was there any cue indicating he was aware of this change. The story highlights that the sound booth is out of Danny's visual range, confirming that he didn't witness the movement. Hence, given the last observed place and lack of communication about the change of place, the most likely location where Danny would look to find the earphones would be the recording booth.*

*ANSWER: 3 − recording booth*

*[[Correct Answer]]*

*2*

