# OpenReview forum: "MuSR: Testing the Limits of Chain-of-thought with Multistep Soft Reasoning"
_ICLR.cc/2024/Conference — ICLR 2024 spotlight_

### Official Review · Reviewer_WJFH · 2023-10-19

**Soundness:** 3 good
**Presentation:** 4 excellent
**Contribution:** 3 good
**Rating:** 5
**Confidence:** 4

**Summary:**

The paper contributed two things. (1) The paper generated 750 test examples in total across three domains. The generation process is grounded in reasoning trees (example in Figure 1). (2) LLMs are run on the benchmark, with different chain-of-thought prompting approaches and some neurosymbolic approaches.

For (1), the three domains are murder mystery stories, object placement, team assignment (involving commonsense, social, physical, theory-of-mind reasoning).
- LLMs are used to generate all the data. Figure 1 gives an example of the data generation process of the murder mystery task.
In general, the generation process is in three steps: domain injection (obtaining initial gold facts), reasoning tree construction (expanding the gold facts), and story generation (generating narrative based on the tree).

For (2), Table 5 provides results on GPT-4, GPT-3.5, two Llama-based models, and three Vicuna-based models. Human performance is high (but far from perfect). The machine performance is quite low (except for GPT-4 which is also far from perfect). A few chain-of-thought variants are explored.

**Strengths:**

The domains are relatively novel.

It’s great that this paper provides an approach to generating arbitrary complex/long stories. If we increase the complexity, at some point it’s possible LLMs won’t be able to solve the problems (but the stories may also get less realistic).

It'll be especially great if the authors use this strategy to generate arbitrary long-context stories so that we can have infinite training/testing data on long-doc QA benchmarks.

**Weaknesses:**

Have humans checked whether the stories are coherent, and reference answers are correct?
- I’m a bit concerned because the human results are far from perfect. Is the result due to poor crowdsourcing incentive structures / crowdworker filtering processes? Or are they good annotators but just being careless? Or are the references actually incorrect (or other problems in the story)?
- Are there confusing / incoherent parts in the story (perhaps inspired by some of the story generation evaluation metrics)?

The benchmark is synthetic (which is fine). It only covers three domains where there are lots of people or objects so it may be hard for LLMs to keep track of them. But:
- There have been so many benchmarks recently (Open Assistant, advanced reasoning benchmark, https://arxiv.org/abs/2307.02477, etc.). How do you expect this particular benchmark to be used?
- We can generate arbitrarily complex synthetic problems. Given that the data is synthetic, how important is it for LLMs to be able to solve these problems (over other benchmarks)? I would be more convinced if this technique can be used to generate arbitrary story and ask arbitrary questions.

Lack of citation on tree construction and incorrect citation.
- The authors’ Table 1 included PrOntoQA which explores logical reasoning, in particular, deductive reasoning (ICLR 2023; https://arxiv.org/abs/2210.01240), but the citation after PrOntoQA is actually ProtoQA (EMNLP 2020, which is quite different!).
- I think PrOntoQA (https://arxiv.org/abs/2210.01240) and PrOntoQA-OOD (https://arxiv.org/abs/2305.15269) are relevant, given that those reasoning papers generated ontology tree first, and then generated the examples from the tree. There may be many other papers that have relevant ideas.
- Some of the story generation ideas are also extremely relevant. For example, https://arxiv.org/abs/2210.06774; check other work by those authors for more examples.

One thing that seems missing (apologies if I missed it) is to count the number of responses which can not be clearly parsed for each method – those responses do not necessarily mean that the model isn’t able to solve the problem. Is the non-parseable responses zero?


Minor: The Table 1 framing needs to be better. For example, if Sophia knows about the shooting, and Sophia has a grudge toward Emily because Emily stole Sophia’s fortune, then the reference says Sophia is the murderer. In the real world this killer identification is not that simple. We can only take a guess based on the existing evidence. We don’t want LLM to be certain that Sophia is the killer. (The prompts in the appendix seems okay.)

**Questions:**

Did the authors try the “let’s think step by step” zero-shot prompt?

Did authors try few-shot prompting (more than 1 shot)

What are the advantages over existing long-document question answering benchmarks? For example, I can see that your generated data will be new, so there won’t be the issue of data leakage (so that LLMs might pretrain on them). You can also make the question arbitrarily long.

---

> ### Author Response · Authors · 2023-11-18
> **Response to Reviewer 4s Questions and Feedback**
>
> Thank you for your review!  We appreciate that you share similar visions for MuSR in self-distillation contexts as well as some of the more nuanced advantages of MuSR like no dataset leakage in the pretraining data.
>
> ### “I’m a bit concerned because the human results are far from perfect. [Why?]”
>
> MuSR was annotated with a 96% human accuracy across all datasets. We consider this close to perfect. Reiterating from section 5.1 in the human evaluation subsection, our analysis showed that a majority of misannotations are due to inattention due to the task requiring careful reading.
>
> It is possible that a failure in the reasoning tree construction or narrative translation process can introduce inconsistencies.  However, this is very rare.  One instance we found of this was in the Object Placements domain:
>
> “[...] the headphones found their resting place on the equipment rack thanks to Mark. […]. Austin […] shifts his attention from the console, rising from his chair […]”
>
> This led the annotators to believe that Austin did not see Mark move the headphones, but in the reasoning tree Austin is said to have seen it because he is near Mark rearranging some equipment. (Note that our high consensus evaluation numbers show that a panel of humans generally *do* agree on who observed what in this domain.)
>
> ### “How do you expect this particular benchmark to be used?”
>
> Reiterating from our conclusion, we expect our dataset to test the limits of current reasoning systems that require vast amounts of commonsense, deductive reasoning, and “system 2” deliberation (decision making).  To our knowledge, no dataset effectively combines all three of these components, yet we consider them vital for real world reasoning (for example, reasoning about real murder mysteries).  Furthermore, the reasoning trees in MuSR allow for more in-depth analysis of intermediate reasoning in both prompt-based and neurosymbolic methods that have previously not been possible.
>
> ### “Given that the data is synthetic, how important is it for LLMs to be able to solve these problems (over other benchmarks)?”
>
> Although the dataset is synthetic, we believe the problems themselves are realistic by virtue of being grounded in real-world reasoning scenarios. The text is not at the level of a great human writer, but it presents the necessary information for the reasoning problem in a coherent narrative. Most importantly, we believe that this kind of long-form narrative reasoning provides an important counterpoint to benchmarks like test questions (e.g., LSAT questions, math questions) which are artificial by design and optimized to test humans’ reasoning capabilities.
>
> I would be more convinced if this technique can be used to generate arbitrary story and ask arbitrary questions.
>
> We believe that MuSR is as close as you can get here with existing methods. Sections 3 and 4 detail how once a basic template is engineered, any number of stories can be generated and with minor tweaks you can ask various questions.  To the best of our knowledge, it’s as close to any work that has ever come to allow for an unguided creation of a narrative QA dataset. In our experience, less scaffolded versions of the dataset creation (e.g., generate a story and then ask questions) lead to either easy questions, simple narratives, or inconsistencies between the story and the generated questions’ answers.
>
> ### “count the number of responses which can not be clearly parsed for each method [...] Is the non-parseable responses zero?”
>
> GPT and Llama 2 70B very rarely produce unparsable answers.  Vicuna 13B produced unparsable answers 7.8% of the time.  Smaller models (7B-scale models) can produce high rates of unparsable answers (either no murderer or the instructions were not followed).  When this happens, we treat the model as having guessed randomly.
>
> ### “The Table 1 framing needs to be better. […] In the real world this killer identification is not that simple.”
>
> We assume this is referring to Figure 1. First, we note that we don’t include the example due to the length of the mysteries in MuSR, but complete examples can be seen in the appendix. The context is proving a motive, not the murderer.
>
> ### “Did the authors try the “let’s think step by step” zero-shot prompt?”
>
> Yes, see table 7 “CoT” row, which is a fairly minimal chain-of-thought prompt similar in spirit to “let’s think step-by-step”.
>
> ### “What are the advantages over existing long-document question answering benchmarks?”
>
> See Table 1 and our conclusion.   To reiterate: our text is not templated, we require multiple steps of reasoning coupled with commonsense, we have intermediate data structures proving the correct answer for each question, and there is no rule-based solution for the examples we create in MuSR.
>
> Furthermore, we can create an arbitrary number of new examples, new domains are easily created (hours with a single engineer), an example is cheap to create (cost of the API call), and there is no dataset leakage.

---

### Official Review · Reviewer_NkVA · 2023-10-30

**Soundness:** 4 excellent
**Presentation:** 4 excellent
**Contribution:** 4 excellent
**Rating:** 8
**Confidence:** 3

**Summary:**

This paper proposes a dataset named MuSR for evaluating language models on multistep soft reasoning. The proposed dataset is created through a novel neurosymbolic synthetic-to-natural generation algorithm using GPT-4, and the data instances contain free text narratives corresponding to real-world domains of reasoning. The dataset is used to evaluate LLMs including GPT-4, GPT-3.5, Llama2, and Vicuna.

**Strengths:**

The proposed dataset provides new challenges to current large language models. The neurosymbolic synthetic-to-natural generation algorithm provides an innovative method for obtaining challenging reasoning data. This paper also conducts comprehensive experiments to demonstrate the properties of the dataset. And the benchmarking experiments that cover extensive analyses.

**Weaknesses:**

Please see the questions listed below.

**Questions:**

Q1: The dataset size is small according to Table 2. Does the dataset size affect the evaluation of model performances?

Q2: In Section 3.2, when recursively calling the GPT-4 to produce the reasoning trees, how to deal with the possible logical or factual inconsistency between the statements/facts?

Q3: What does “CMCR” in the title of Section 5.2 refer to?

---

> ### Author Response · Authors · 2023-11-18
> **Response to Reviewer 3s Questions and Feedback**
>
> Thanks for your review and feedback!
>
> ### “The dataset size is small according to Table 2. Does the dataset size affect the evaluation of model performances?”
>
> Increasing the size of the dataset would lower the variance of model accuracy numbers, but we do not believe it would change the core conclusion of this paper. Furthermore, given the expense of running LLMs on long narratives, particularly for researchers who want to explore neurosymbolic reasoning methods like decomposed prompting that require several prompts per example, we felt that the size of this benchmark balanced affordability of running on our dataset with high enough statistical power. Because human eval is averaging 96% and the models are underperforming by a decent margin, we believe that this gap will remain and is still useful for researchers to build on.
>
> ### “In Section 3.2, when recursively calling the GPT-4 to produce the reasoning trees, how to deal with the possible logical or factual inconsistency between the statements/facts?”
>
> Section 3.2 and appendix C.1 describe our validation algorithms in detail.  These validators aim to ensure that the facts being generated in the reasoning tree are consistent both with themselves and the generated narratives. Ultimately, the human performance reflects that humans agree on the answers, which is an extrinsic measure of this kind of factual consistency.
>
> ### “What does “CMCR” in the title of Section 5.2 refer to?”
>
> This is an erroneous reference to an earlier name of the dataset. It is fixed now.

---

### Official Review · Reviewer_cJvj · 2023-11-01

**Soundness:** 2 fair
**Presentation:** 2 fair
**Contribution:** 2 fair
**Rating:** 6
**Confidence:** 3

**Summary:**

This paper proposed a new dataset, called MuSR, for evaluating LLMs in multi-step reasoning. The authors argued that existing datasets either are not complicated enough (e.g., requiring limited steps of reasoning) or don't have sophisticated natural language descriptions. MuSR instead is a synthetic dataset in three domains, i.e., murder mysteries, object placement, and team assignment, whose questions often involve multi-step reasoning over the provided textual descriptions. Experiments showed that state-of-the-art LLMs such as GPT-4 can achieve decent performance but still underperform humans by a large gap. Furthermore, customized, domain-specific CoT prompting, or prompting with 1 shot of examples, helps.

**Strengths:**

The paper contributes a new dataset for evaluating LLMs in multi-step reasoning tasks, particularly when the reasoning has to be grounded in natural texts. As the dataset is synthetic, it has the potential to scale up to an arbitrary size (though I have a concern; see Weakness 1) and can be refreshed in the future. In experiments, the dataset was shown to be challenging for LLMs.

**Weaknesses:**

1. It's unclear to me how much human engineering is needed in the dataset construction process. The major confusion comes from the "scenario-specific facts" generation. Looking at the prompt in Appendix C.1, at the end of page 30, it seems that human engineering is needed to provide the scenario setup, e.g., the following. If my understanding is correct, then the dataset is not "scalable" (which the authors claimed is a main advantage over human-annotated ones).
```
Scenario: Victim: Tessa
Crime Scene: kitchen
Murder Weapon: poisonous gas
Suspect: Penelope
Role in story: Tarot Reader
The suspect’s motive: To protect a secret
```

2. The dataset creation ablation needs more justification. The authors suggested low GPT-4 performance (Acc) from the ablated baselines as a good indicator of the effectiveness of each creation component, but MuSR outperforms the two ablations in Object Placements, which seems to imply that the creation components are not useful for Object Placements. Reverse observations were seen in the Murder domain, and no comparison was done for Team Allocation.

3. The current experiment exploration seems a bit shallow, as the discussions are mostly about comparing numbers achieved by different LLMs or prompting methods. It's suggested that the authors could provide some qualitative analyses and elaborate on the bottlenecks for LLMs to perform well in such tasks.
In addition, adding 1 shot of example showed to improve GPT-4 CoT+ substantially. A conjecture is that adding more examples could improve the performance further, eventually to be close to the human performance. In this case, can this dataset still be considered as challenging one?

**Questions:**

1. Could the authors clarify the human engineering in the dataset construction?
2. Could the authors provide more justification for the creation ablation analysis?
3. Could the authors clarify the dataset challenge under few(>1)-shot settings?

---

> ### Author Response · Authors · 2023-11-18
> **Response to Reviewer 2s Feedback and Questions**
>
> Thank you for your review.  We are encouraged that you have found our dataset scalable while still being grounded in natural text. To that point, your question:
>
> ### “how much human engineering is needed in the dataset construction process”
>
> The facts you list from the prompt in “Scenario: Victim: Tessa…” are described in Appendix D.1 in more depth. This template is constructed through engineering, but the values are sampled from LLMs to provide diverse scenario facts. That is, once a domain (like a murder mystery) is set, we prompted GPT-4 for “100 murder weapons” or “100 crime scenes” and sample from those lists.
>
> Therefore, the extent of engineering for each domain is designing the templates themselves and the prompts. The actual engineering for each specific example is zero, as these templates are filled automatically from our generated lists.
>
> We have clarified this in the new version of Section 3.
>
>
> ### “The authors suggested low GPT-4 performance (Acc) from the ablated baselines as a good indicator of the effectiveness of each creation component”
>
> Actually, the point in Section 5.1 is a bit more nuanced. “Although our goal is to make a challenging dataset, in the context of this table, low GPT-4 performance usually means that the examples are ambiguous or unsolvable.”  We argue that low model accuracy, high human accuracy, *and* other criteria like length and fact recall are required to have a good dataset. Thus changes in performance on that ablated dataset could improve due to consistent reasoning be applied in the narrative (more solvable questions) or the performance could decrease due to more complex reasoning be adhered to (narrative is more faithful to the reasoning tree).
>
> Looking at only one of those columns does not lead to a complete picture.
>
> ### “no comparison was done for Team Allocation. It's suggested that the authors could provide some qualitative analyses and elaborate on the bottlenecks for LLMs to perform well in such tasks.”
>
> We mention in Appendix C.3 that Team Allocation does not require validators: “Team Allocation required no validators in the construction of the reasoning tree nor the story to create valid contexts.”  This is why 2 out of 5 ablations are left blank in Table 4 for this domain. We further clarified this in the new version of the paper.
>
>
> ### “The current experiment exploration seems a bit shallow, as the discussions are mostly about comparing numbers achieved by different LLMs or prompting methods.”
>
> We would like to point out that our goal is to benchmark existing methods on this dataset, most of which do consist of variants of chain-of-thought prompting. Few other neurosymbolic methods can work broadly here (see 5.2). Finally, we can expand our qualitative analysis in a future version, but largely left this in the appendix due to space constraints.

---

> > ### Comment · Reviewer_cJvj · 2023-11-21
> > **Thanks for the clarification!**
> >
> > The response has addressed my concern about the dataset construction, and reading other reviewers' comments is also helpful. I will increase my score accordingly. However, I'm still hoping to see more qualitative analysis and get more insight into how LLMs can be promising (or not) to solve such complicated tasks in the future.

---

### Official Review · Reviewer_WiLx · 2023-11-05

**Soundness:** 3 good
**Presentation:** 3 good
**Contribution:** 3 good
**Rating:** 8
**Confidence:** 4

**Summary:**

This paper introduces a new dataset and mechanism for generating complex and challenging reasoning problems that test the limits of SotA LLMs. The resulting examples are designed to be long and contain elements of both (realistic) natural language and deductive reasoning. The method involves building a reasoning tree skeleton of facts and rules that are turned into narratives (through "chaptering")  for which particular questions are elicited. Examples in 3 different domains are generated and released (though the method can be applied to others) and an evaluation is preformed on a few SotA LM's like GPT-4, showing that there is still a gap to be crossed in the ability of LM's to do reasoning.

**Strengths:**

First of all, I find the direction of this work very important. There is a sore need for datasets and evaluations like the one generated here, as previous reasoning datasets, that focus on simpler forms of reasoning or expressed in very synthetic unrealistic language, get saturated as being too easy for SotA models. The current direction of Reasoning with LLMs has consequently taken a turn towards focusing on domains where we still have access to challenging datasets like Mathematics, which while important, is not representative of the real-world "messy" general reasoning that LLM's need to demonstrate to demonstrate next-level capabilities. The dataset and method presented here, while flawed in some ways, is to my knowledge the best so far at this.

I wish to call out the originality of the method as well, since it requires a lot more than the standard dataset generation workflow of decide domain -> collect some seed data -> do a lot of crowd-sourced collection-> refine. This requires a careful design of a 'synthetic' data pipeline, through key novel components like the reasoning tree and 'chaptering',  that nonetheless does seem to generate many truly realistic real-world scenarios (again at least in comparison to previous datasets) that challenge GPT-4 and its competitors.

The results show a reasonable (though not huge) gap between the best performing pure LLM and the claimed ceiling (i.e. the best performing human), which represents the opportunity for models to improve their reasoning capacity.

The paper is reasonably clear on what it is doing, though some details had me confused, which I enumerate below. I appreciate the thorough comparison in features to recent datasets.

**Weaknesses:**

By far the biggest concern I have, which is derived from working on attempts at similar dataset creation, is that there is a point where despite designing the reasoning chains to be deductively entailed from premises, somewhere in the translation to ambiguous natural language and the accumulation of steps, reasonable humans start to disagree on whether a particular conclusion follows or not, at least with certainty. This can lead to there possibly being a sizable number of cases in the dataset where in fact a single gold answer cannot (even in principle) be attributed to the problem.  I am wondering if the authors have considered this issue and how they may say their dataset construction procedure guards against it. Would they claim the best performing or majority human annotator label be considered a lower bound on the precentage of problems for which a definite true answer exists?

I am also slightly concerned about the self-referential use of GPT-4 in constructing the dataset. I think the authors have put in a lot of work to minimize the problem, since they use it only for specific steps after the bulk of the reasoning chain has been fleshed out by the reasoning tree algorithm and the chaptering approach introduces tight constraints so that it seems that the main effect of GPT-4 is 'simply' to turn the abstract narrative into natural language. However recently, there have been concerns raised about the practice of the field as a whole relying on LM's for every stage of generation,evaluation and so on (see [1]) so it would be good to discuss this. indeed the jump in performance from LLama 2 to GPT-4 is large enough to make me a bit suspicious that the use of GPT-4 in the dataset construction is biasing these results in a subtle manner. Would it have been helpful to have used a mixture of LMs to generate the text for the narratives?

A minor point: It seems the second part of the benchmarking is not as complete as may be hoped for. I do appreciate the attempt to explore neuro-symbolic approaches, and the authors used a fair number (5), but why was each only applied to a limited number of domains (were there limitations to each that prevented this?). Also, LAMBADA [2] is a recent general purpose approach to neurosymbolic reasoning that seems like it would be very helpful in all 3 domains (since it performs backward reasoning which is generally more effective on complex reasoning problems) that the authors could have tried.

[1] https://arxiv.org/abs/2307.01850
[2] https://arxiv.org/abs/2212.13894

Overall none of these concerns seem fatal to me. I think this paper makes a contribution that can unblock further progress in the reasoning field and should be accepted unless Ive missed something.

**Questions:**

1. I take objection to the title only talking about CoT; while obvious a big breakthrough in LM reasoning it isn't the only game in town :)

2. Table 2: suggest adding citations to the papers for each dataset in the table next to the dataset names.

3. Could the stories on 5minutemystery.com themselves be used for the dataset?

4. page 3 last line: sort of misleading claim. GPT-4 is used in generating the dataset, even if not the "reasoning chain" in an abstract sense.

5. you have not defined Prompt(T \ s)

6. there is no hard deductive  that a murderer "must* have  motive/opportunity/means to commit a murder, it is a sort of common-sense notion. So apart from the Cot+ setting where you explicitly call that out in the prompt, how is the system to know it has to look for this exclusively?

7.  sec 4.3: "strength of a relationship" is a vague term. I have to say that a lot of the writing from sec 3.2-4.3 could use a little more explanation and polish; i could only understanding them after looking at appendix.

---

> ### Author Response · Authors · 2023-11-18
> **Response to Reviewer 1s Feedback and Questions**
>
> Thank you for your feedback!  We are happy to hear you agree with us that this is an important direction for creating challenging reasoning datasets and that the creation process is more sustainable than the traditional method of creating benchmarks.
>
> ### "[...] somewhere in the translation to ambiguous natural language and the accumulation of steps, reasonable humans start to disagree on whether a particular conclusion follows or not [...]  I am wondering if the authors have considered this issue [...]"
>
> We have, and this is a difficult issue to fully address. We describe validation algorithms in section 3.2 and go into specific details about entailment validation in Appendix C.1:
>
> “[...] we also implemented a story validator during the narrative generation. The story validator prompts GPT-4 to check the entailment of each scenario-specific fact S(T) given the story. If any fact is not entailed by the story, we prompt GPT-4 to rewrite the story and ensure that fact is entailed.”
>
> The human accuracy also serves as an end-to-end measure of correctness of each example, giving us a combination of automated stepwise validation and human end-to-end validation. These are precision-focused measures that show that the ground truth reasoning is sound. However, we acknowledge that human evaluation of each step would be beneficial. Furthermore, we agree it would be useful to have another eval focused on whether alternative reasoning paths are possible.
>
> ### "recently, there have been concerns raised about the practice of the field as a whole relying on LM's for every stage of generation,evaluation and so on (see [1]) so it would be good to discuss this."
>
> Reiterating a bit from section 2 “Why a synthetic benchmark”, because the human eval is at 96% on average, we believe that language models should be able to solve MuSR regardless of how it was constructed. The structured nature of the creation process means that GPT-4 is not quite as privileged as if we simply generated instances from GPT-4 directly and asked GPT-4 to solve them, but we agree it still might find the text more “in-distribution” than other language models do.
>
> ### "Would it have been helpful to have used a mixture of LMs to generate the text for the narratives?"
>
> Maybe! This would be an exciting experiment to try out. We used GPT-4 because it generated the highest-quality narratives.
>
> ### "I do appreciate the attempt to explore neuro-symbolic approaches [...] but why was each only applied to a limited number of domains (were there limitations to each that prevented this?)."
>
> Yes. Each neuro-symbolic method is best fit for the domain we tested on.  Expanding them to work on other domains would require engineering a new algorithm which wasn’t the point of the evaluation.
>
> For example, SymbolicToM wouldn’t work on Murder Mysteries since the facts for motive/means/opportunity could span multiple paragraphs that are unrelated. Thus, no graph could be constructed.  Similarly, deconstructed prompting would require modification for object placement and team allocation as they require some form of decision-making after recovering higher-level facts.
>
> ### "LAMBADA [2] is a recent general purpose approach to neurosymbolic reasoning that seems like it would be very helpful in all 3 domains"
>
> Unfortunately, LAMBADA does not allow for natural narratives as input; it needs structured input as rules & facts. An extended version of this algorithm would be exciting to try in future work.
>
> ### "Could the stories on 5minutemystery.com themselves be used for the dataset?"
>
> We plan to explore evaluation on this dataset in future work. Our goal for this work was to explore a construction procedure that could generalize across multiple domains, and examples from 5minutemystery don’t have underlying reasoning traces that would allow them to deeply integrate with our method.
>
> ### "page 3 last line: sort of misleading claim. GPT-4 is used in generating the dataset, even if not the "reasoning chain" in an abstract sense."
>
> We are saying here that GPT-4 creates the dataset but the reasoning behind the correct answer is not GPT-4 specific (meaning that other models and humans can still solve them).  This is validated by our human experiments as well.
>
> ### "there is no hard deductive that a murderer "must* have motive/opportunity/means to commit a murder [...] So apart from the Cot+ setting where you explicitly call that out in the prompt, how is the system to know it has to look for this exclusively?"
>
> We acknowledge that the model may not have the precise task definition by default. We view CoT+ as the most suitable baseline, as this also matches the instructions given to annotators. We’ve clarified this in the new version of the paper.
>
> ### "I have to say that a lot of the writing from sec 3.2-4.3 could use a little more explanation and polish; I could only understanding them after looking at appendix."
>
> We agree. We have rewritten these sections to provide clarification.

---

> > ### Comment · Reviewer_WiLx · 2023-11-21
> > **thanks**
> >
> > Thank you for your responses. I have nothing substantial to add, but 2 requests:
> >
> > re: "page 3 last line: sort of misleading claim. GPT-4 is used in generating the dataset, even if not the "reasoning chain" in an abstract sense."
> >
> >   I understand what you're trying to say, but please do rewrite so it is not misunderstood.
> >
> > Also there was another more relevant work (than the autophagous one i shared before) that showed that auto-raters based on the same family of LLMs tended to prefer the outputs of their sibling models vs outputs of others (e.g. GPT-4 auto-raters prefer GPT-3 responses over LLama etc). That is very relevant to point out as a caveat, and I really do encourage you to consider creating a version of the benchmark that uses a combination of LLMs for the generation (even if it does "worse" in some absolute sense). TBH, it would make me more likely to use it and consider it reliable.

---

### Author Response · Authors · 2023-11-18
**General Response to Reviewers (Tweaks and new few-shot experiments!)**

## Summary

Thanks to all the reviewers for their thoughtful comments and feedback! We responded to individual comments below, but comment here on our new draft and a new experiment we ran.

## New draft

We have uploaded a new version of the paper with the following changes:
Few-shot experiments (requested by Reviewer cJvj and Reviewer WJFH). See below for a summary
Rewritten section 3 and 4 and Figure 2 to make the process more clear and clarify the human engineering involved (particularly in section 3.2-4.3)
Minor framing changes and fixes suggested by reviewers.

## New experiment: 3-shot prompting

We ran few-shot prompting experiments with GPT-4 and GPT-3.5, increasing the number of shots from 1 (reported in the previous draft) to 3 examples per domain. 3 examples reached the context size limit of GPT-4.

For GPT-4, the results are mixed.  Very small performance deltas were observed for the three domains (murder mysteries: -1.2%; object placements: -0.8%; team allocation: +1.2%). The additional shots do not have a meaningful impact on GPT-4’s performance, likely because it can already imitate the form of the reasoning given in a single shot and the additional examples do not actually help teach it broadly how to reason in these domains.

GPT-3.5 saw similar trends except for team allocation, where it showed a large increase (28.4%) but is still below GPT-4 and Human performance. We analyzed the behavior in this setting, and we believe this is due to the in-context examples helping the model understand the last part of the team allocation problem where you must assign people to each skill then sum the skills and relationships scores to find the best assignment.

With 1-shot GPT3.5 accidentally adds terms for multiple relationships when really only 1 relationship should be accounted for (who is actually working together)

*1-shot:*
```
…
1 - Animal Caretaker: Alex (1) + Exhibit Cleaner: Mia (3) and Olivia (3) + Relationship(Alex, Mia) (3) + Relationship(Alex, Olivia) (1) + Relationship(Mia, Olivia) (2) = 13
2 - Animal Caretaker: Olivia (1) + Exhibit Cleaner: Alex (3) and Mia (3) + Relationship(Alex, Olivia) (1) + Relationship(Mia, Olivia) (2) = 10
3 - Animal Caretaker: Mia (1) + Exhibit Cleaner: Alex (3) and Olivia (3) + Relationship(Alex, Mia) (3) + Relationship(Mia, Olivia) (2) = 12
…
```

With multiple shots, we see that the model understands how to perform this calculation:

*Few-shot:*
```
…
1 - Animal Caretaker: Alex (1) + Exhibit Cleaner: Mia (3) and Olivia (1) + Relationship(Mia, Olivia) (2) = 7
2 - Animal Caretaker: Olivia (3) + Exhibit Cleaner: Alex (2) and Mia (3) + Relationship(Alex, Mia) (2) = 10
3 - Animal Caretaker: Mia (1) + Exhibit Cleaner: Alex (2) and Olivia (1) + Relationship(Alex, Olivia) (1) = 5
…
```

---

### Meta-Review · Area_Chair_ruCH · 2023-12-06

**Metareview:**

This paper introduces MuSR, a new dataset for evaluating language models on multistep soft reasoning tasks specified in a natural language narrative. Their method introduces a scalable annotation pipeline allowing them to generate natural mystery stories from initially symbolically-defined settings. The resulting datasets contains examples designed to be long and contain elements of both (realistic) natural language and deductive reasoning. All reviewers agreed this was a strong paper, highlighting the method's originality, the dataset's difficulty, and the clarity of the paper. I agree, and think it should be accepted.

**Justification For Why Not Higher Score:**

There have been prior works on converting symbolic tree structures to natural language reasoning scenarios.

**Justification For Why Not Lower Score:**

See meta-review.

---

### Decision · Program_Chairs · 2024-01-16

Accept (spotlight)